# SOCIAL BAYESIAN OPTIMIZATION FOR BUILDING SOCIAL-INFLUENCE-FREE CONSENSUS

## ABSTRACT

We introduce *Social Bayesian Optimization* (SBO), a query-efficient algorithm for consensus-building in collective decision-making. In contrast to single-agent scenarios, collective decision-making encompasses group dynamics that may distort agents' preference feedback, thereby impeding their capacity to achieve a social-influence-free consensus—the most preferable decision based on the aggregated latent agent utilities. We demonstrate that under standard rationality assumptions, reaching social-influence-free consensus using noisy feedback alone is impossible. To address this, SBO employs a dual voting system: cheap but noisy public votes (e.g., show of hands in a meeting), and more accurate, though expensive, private votes (e.g., one-to-one interview). We model social influence using an unknown social graph and leverage the dual voting system to efficiently learn this graph. Our findings show that social graph estimation converges faster than the black-box estimation of agents' utilities, allowing us to reduce reliance on costly private votes early in the process. This enables efficient consensus-building primarily through noisy public votes, which are debiased based on the estimated social graph to infer social-influence-free feedback. We validate the effectiveness of SBO across multiple real-world applications, including thermal comfort optimization, team building, travel destination discussion, and strategic collaboration in energy trading.

## 1 INTRODUCTION

This paper presents a sample-efficient algorithm to achieve consensus $x^\star$ in an online setting, within a social group of agents, by solving the following optimization problem:

$$x^\star \in \arg\max_{x \in \mathcal{X}} \mathcal{A}[u(x,:)], \quad u(x,:) = \{u(x,i)\}_{i \in V}, \qquad (1)$$

where $V$ represents a social group of $n$ agents, i.e., $|V| = n$, $\mathcal{X}$ is a set of options that is a bounded subset of $\mathbb{R}^d$, and $\mathcal{A}$ is an aggregation function that produces the *social utility*. For each agent $i \in V$ and option $x \in \mathcal{X}$, $u(x,i)$ represents the utility of option $x$ for agent $i$, and $u(x,:) = \{u(x,i)\}_{i \in V}$ represents the set of utilities for all agents given option $x$. We assume throughout that $u : \mathcal{X} \times V \to \mathbb{R}$ is a black-box utility function that rationalizes the collective preferences of the social group $V$. We iteratively collect votes from agents to minimize regret to reach $x^\star$ within the given budget.

Efficient consensus-building is essential for collective decision-making but is notoriously difficult, underpinned by various theoretical impossibility results (Patty & Penn, 2019; Muandet, 2023). Three key challenges contribute to this difficulty. First, modeling human preferences is inherently complex. Although utility theory (Fishburn, 1968) offers a framework, individuals often struggle to introspect and report their preferences accurately (Kahneman & Tversky, 1979), so preferences are inferred from feedback, such as pairwise comparisons and rankings (Fürnkranz & Hüllermeier, 2010). Second, even with access to each agent's utility, constructing a social utility that fairly represents all agents is impossible under mild rationality constraints (Arrow, 1950). Third and most importantly for our settings, social dynamics, i.e. interactions between agents, can distort preference feedback. For instance, the bandwagon effect (Simon, 1954; Farjam, 2021) occurs when agents' preferences are swayed by influential individuals, resulting in biased or influenced feedback, i.e., feedback that does not reflect their actual underlying utilities. This can result in *groupthink*, a failure mode in collective decision-making where the group reaches a influenced consensus, and Section 5 introduces four real-world examples.

Preferential Bayesian optimization (PBO; González et al. 2017) provides a potentially sample-efficient framework for solving (1). However, their approach fails to address the second and third challenges mentioned above. Their framework focuses on single-agent scenarios or assumes homogeneity among agents, averaging preferences without addressing individual differences and social dynamics (Xu et al., 2024b). The influence of social factors is non-trivial, as it distorts agents' opinions, resulting in *influenced* votes. Ideally, the consensus-building procedure should be robust to social influence. However, under mild rationality constraints, we demonstrate that it is impossible to design a social aggregation method that consistently produces a consensus robust to social dynamics—a phenomenon we term the *impossibility of groupthink-proof consensus*. To circumvent this issue, we propose a dual voting system consisting of a cheap but noisy **public vote** (e.g., show of hands in a meeting) and a **private vote** (e.g., one-to-one interview) that is free from social influence but costly to obtain. We formalize the social influence effect through an unknown social graph and utilize the dual feedback mechanism to perform graph learning (Dong et al., 2019). Our results show that the social graph estimation converges faster than black-box utility estimations, allowing us to reduce reliance on private feedback early in the process (Theorem 3.12). This approach facilitates efficient consensus-building, primarily relying on noisy public feedback while accounting for social dynamics (see Figure 1 for an illustration of our procedure).

**Contributions.** (1) **Novel setting**: We formulate a new class of optimization problems to facilitate collective decision-making under social influence among heterogeneous agents. (2) **Algorithm**: We introduce a Social Bayesian Optimization (SBO) algorithm that enables vote-efficient consensus-building robust to social influence. (3) **Theoretical**: We prove an impossibility result highlighting the necessity of a dual voting system. We also establish two sublinear convergence rates for SBO: one for cumulative regret, ensuring SBO achieves groupthink-proofness and no-regret property, and one for social-influence-free votes, allowing the algorithm to reduce reliance on costly private votes as the

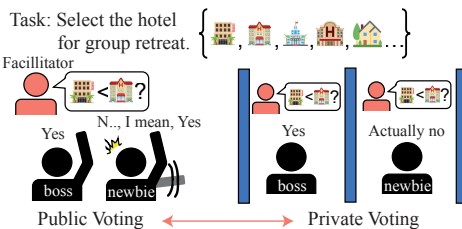

Figure 1: A dual voting: the difference in public and private votes can identify the social influence. Once identified, social-influence-free consensus can be estimated only from noisy public votes, thereby reducing the total cost.

model gains sufficient confidence. (4) **Real-world contributions**: We demonstrate SBO's fast convergence than baselines in four practical scenarios.

## 2 Social Bayesian Optimization

Unlike standard Bayesian optimization (BO), where direct evaluation of the black-box objective is possible, some objectives—like human utility—are latent and difficult to introspect. To solve (1), we rely on preference feedback, or *votes*, where each agent expresses preferences between pairs of options $[x, x']$. Following standard preference modeling, agent $i$ prefers option $x_1$ over $x_2$, denoted $x_1 \succ_{u(\cdot,i)} x_2$, if and only if $u(x_1, i) > u(x_2, i)$, where $\succ_{u(\cdot,i)}$ represents agent $i$'s preference relation. Our optimization procedure thus involves estimating $n$ utility functions $\{u(\cdot, i)\}_{i \in V}$ from their *votes*. When the context is clear, this set also denotes the vector of utilities queried at $x$. Pairwise comparisons are widely used in human-in-the-loop systems (Koyama et al., 2020; Li et al., 2021), as people tend to evaluate relative differences better than absolute magnitudes (Kahneman & Tversky, 2013). In our setting, we make two key assumptions.

**Assumption 2.1** (**Facilitator**). *There exists a single facilitator (or social planner) who facilitates the decision-making process, and decides the aggregation rule $\mathcal{A}$.*

**Assumption 2.2** (**Pairwise feedback**). *Given a pair of options $(x_t, x'_t)$ at time step $t$, there exists an oracle that returns a preference signal $\mathbf{1}^{(i)}_{x \succ x'}$ from the $i$-th agent where $\mathbf{1}^{(i)}_{x \succ x'} := 1$ if $x$ is preferred and zero if $x'$ is preferred. The feedback $\mathbf{1}^{(i)}_{x \succ x'}$ from the oracle follows the Bernoulli distribution with $\mathbb{P}(\mathbf{1}^{(i)}_{x \succ x'} = 1) = p^{(i)}_{x \succ x'} = \sigma(u(x, i) - u(x', i))$, where $\sigma(z) = (1 + \exp(-z))^{-1}$.*

Assumption 2.2 is widely accepted Bradly-Terry model (Bradley & Terry, 1952).

### 2.1 Aggregation Functions

The crucial part of problem (1) is $\mathcal{A}$ that aggregates agents' utilities into a social utility.

**Definition 2.3** (**Aggregation function**). *The aggregation function $\mathcal{A}$ (also known as a social choice function or social welfare function) combines individual utilities into a single utility via a positive linear combination, i.e., $\mathcal{S}(x) := \mathcal{A}[u(x,:)] = \mathbf{w}^\top u(x,:)$, where $\mathcal{S}(x)$ represents the social utility and $\mathbf{w} \in \mathbb{R}^n_{\geq 0}$ depends on $u(x,:)$. $\mathcal{A}$ is provided* a priori *by the facilitator and is independent of both the option $x$ and the time step $t$, meaning it is homogeneous and stationary.*

Harsanyi (1955) demonstrated that a positive linear combination of individual utilities is the only aggregation rule that satisfies both the von Neumann-Morgenstern (VNM) axioms (Von Neumann & Morgenstern, 1947) and Bayes optimality (Brown, 1981). Many popular aggregation function, such as the utilitarian rule, i.e. $\mathcal{A}[u(x,:)] = 1/n \sum_{i \in V} u(x,i)$ and the egalitarian rule, i.e., $\mathcal{A}[u(x,:)] = \min_{i \in V} u(x,i)$ are positive linear combinations (c.f. Appendix C.1). Given the range of candidate methods, we adopt the generalized Gini social-evaluation welfare function (GSF; Weymark (1981); Sim et al. (2021)) as it can interpolate between utilitarian and egalitarian approaches:

$$\mathcal{A}[u_t(x,:)] := \mathbf{w}^\top \phi(u_t(x,:)) \quad \text{s.t.} \quad w_i := \rho^{i-1}/\mathbf{w}^\top \mathbf{1}, \ w_i > 0, \ 0 < \rho \leq 1, \tag{2}$$

where $\mathbf{w} := (w_i)_{i \in V}$ is a weight vector, $\mathbf{1}$ is the one-vector, and $\phi$ is a sorting function that arranges the elements of the input vector in ascending order and returns the sorted vector.

**Proposition 2.4** (Proposition 1 in Sim et al. (2021)). *GSF in Eq. (2) satisfies monotonicity and the Pigou-Dalton principle (PDP; Pigou (1912); Dalton (1920)) on fairness. Moreover, when*

(a) *$\rho = 1$, $\mathcal{A}$ is utilitarian such that it satisfies the PDP in the weak sense;*

(b) *$0 < \rho < 1$, $\mathcal{A}$ satisfies the PDP in the strong sense;*

(c) *$\rho \to 0$, then $w_i/w_1 \to 0$ for $i = 2, \ldots, n$, $\mathcal{A}$ converges to egalitarian.*

See Appendix C.4 for the proof and the details. The key result is that the GSF interpolates between two popular aggregation rules through a single real parameter, $\rho$, while adhering to the fairness principle of PDP and maintaining monotonicity for regret analysis. Although $\rho$ must be defined a priori, we assume the facilitator will make this definition (Assumption 2.1). We argue that selecting a single parameter is far simpler than choosing an arbitrary aggregation function, and GSF is intuitive: $\rho$ controls the balance between prioritizing the group average and the worst-off agent. Nonetheless, our algorithm, described later, is applicable to any aggregation function that is a positive linear combination and monotonic, encompassing a wide range of popular aggregation rules.

## 2.2 Modelling Social Influence

While various methods exist to model social interactions among agents, we chose a graph convolutional approach. Let $V$ represent the set of nodes (agents) and $E$ the set of weighted directed edges, forming the social influence graph $G = (V, E)$, with $A$ as the corresponding adjacency matrix.

**Definition 2.5** (**Social influence**). *Given influence graph $G$, social-influence-free utility $u$, and agent $i \in V$, the corrupted utilities of agent $i$, $v(\cdot, i)$ can be expressed as $h(u(\cdot, i), \{u(\cdot, j) \mid j \in N_G(i)\})$, for some generic function $h$ that specifies how the signals interact, and $N_G(i)$ is the (in)-neighbour of $i$ in $G$, defined as $\{j : A_{ij} \neq 0\}$.*

We make the following assumption on the utility functions $u(\cdot, :)$ and $v(\cdot, :)$.

**Assumption 2.6** (**Bounded norm**). *For each $i \in V$, let $\mathcal{H}_{k_i}$ be a reproducing kernel Hilbert space (RKHS) endowed with a symmetric, positive-semidefinite kernel function $k_i : \mathbb{R}^d \times \mathbb{R}^d \to \mathbb{R}$. We assume that $v(\cdot, i) \in \mathcal{H}_{k_i}$ and $\|v(\cdot, i)\|_{k_i} \leq L_v$, where $\|\cdot\|_{k_i}$ is the norm induced by the inner product in the corresponding RKHS $\mathcal{H}_{k_i}$, $k_i(x, x') \leq 1$, $x, x' \in \mathcal{X}$, and $k_i(x, x')$ is continuous on $\mathbb{R}^d \times \mathbb{R}^d$. We denote the set $\mathcal{B}^{v_i} := \{\tilde{v}(\cdot, i) \in \mathcal{H}_{k_i} \mid \|\tilde{v}(\cdot, i)\|_{k_i} \leq L_v\}$, and $\mathcal{B}^v := [\mathcal{B}^{v_1}, \cdots, \mathcal{B}^{v_n}]$. This assumption also applies to $u(\cdot, :)$, of which bound is isotropic; $L_u, \forall i \in V$.*

Assumption 2.6 require that the utility functions $u, v$ are regular in the sense that they have bounded norms in the corresponding RKHS. These assumptions are common in the BO literature. Under Definition. 2.5, it is natural to consider the graph convolution operation as the function $h$,

$$v^\top(\cdot, :) = A u^\top(\cdot, :), \ \text{s.t.} \ v(\cdot, i)^\top = A_{ii} u^\top(\cdot, i) + \sum_{j \in N_G(i)} A_{ij} u^\top(\cdot, j), \ \sum_{j=1}^n A_{ij} = 1, \ A_{ij} > 0. \tag{3}$$

**Lemma 2.7** (**Graph properties**). *Under Eq. (3) and Assumption 2.6, $u, v \in [-L_v, L_v]$ have the same bound ($L_v = L_u$), thereby being comparable. Moreover, if $A$ is invertible, the matrix $A$ has the Euclidean norm of inverse matrix bounded by $1 \leq \|A^{-1}\| \leq n$, and $A$ is identifiable from the observed data pairs $(v(x_\tau, :), u(x_\tau, :))_{\tau \in [T]}$ if $(u(x_\tau, :))_{\tau \in [T]}$ is full rank.*

See Appendix B.1 for the proof. See also Appendix B.3 for the extension to other graph structures.

## 3 MITIGATING UNDUE SOCIAL INFLUENCE

As discussed, the main challenge in achieving consensus is social influences that hinder the facilitator from accessing the true utilities. Consequently, the facilitator can only observe the feedback from distorted utilities, leading to a consensus that misrepresents the agents' actual preferences, i.e., groupthink. To overcome this, we propose in Section 3.2 a dual voting mechanism consisting of a cost-effective but noisy public vote and a costly private vote that is free from social influence.

### 3.1 IMPOSSIBILITY THEOREM

Before introducing our dual voting mechanism, we first illustrate the difficulty of consensus building based solely on distorted utilities through the impossibility theorem. To this end, we define groupthink-proofness as desirable property of any aggregation function.

**Definition 3.1** (**Groupthink-proof**). *An aggregation function $\mathcal{A}$ is groupthink-proof if, for any social-influence graph $G$, $\arg\max_{x \in \mathcal{X}} \mathcal{A}[u(x,:)] = \arg\max_{x \in \mathcal{X}} \mathcal{A}[v(x,:)]$.*

Intuitively, the aggregation function $\mathcal{A}$ is groupthink-proof if its consensus is preserved under any social influence graph. For example, any aggregation function is groupthink-proof if $v(\cdot,:) = u(\cdot,:)$, i.e., no social influence. In addition, we define the triviality of the social consensus $x^*$ as

**Definition 3.2.** *(Trivial social consensus) $x^*$ is the trivial social consensus if for all $i \in |V|$, $x^* = \arg\max_x u(x,i)$*

Non-triviality of social consensus excludes the situations where all agents unanimously agree on the best option. The following theorem states that, in the absence of a trivial consensus, no aggregation function when applied on the distorted utilities satisfies groupthink-proofness.

**Theorem 3.3** (**Impossibility of groupthink-proof aggregation**). *Under Definitions 2.3, 3.1, 3.2, there exists no aggregation rule $\mathcal{A}$ satisfying groupthink-proof in the absence of a trivial consensus.*

Theorem 3.3 implies that if the agents are not unanimously in consensus on the best option with respect to their true utilities, then the facilitator cannot identify an aggregation function that is groupthink-proof. Appendix C.2 provides the detailed proof where the core idea of the proof is to assume that there exists a groupthink-proof aggregation rule $\mathcal{A}$ in the absence of a trivial social consensus and then show that consensus obtained by this aggregation rule needs to satisfy triviality if the aggregation rule is groupthink proof w.r.t a selected subset of influence graphs. Thus resulting in a contradiction. To show the direct implication of our impossibility results we show that a non-trivial set of consensus—the Pareto front of distorted utilities—is not groupthink-proof, see Appendix C.3 for more details.

### 3.2 DUAL VOTING MECHANISM WITH APPROXIMATED SOCIAL GRAPH

We now introduce the procedure for learning the social graph $A$ and the surrogate models for $u$ and $v$. While preferential Gaussian processes (GPs) (Chu & Ghahramani, 2005) are a common choice, combining a Gaussian prior with a Bernoulli likelihood poses theoretical and computational challenges. Hence, we opted for the likelihood ratio model (Owen, 1990; Emmenegger et al., 2024), which estimates only the worst-case prediction interval, rather than the full predictive distribution. This approach aligns well with optimistic algorithms like UCB (Srinivas et al., 2010), which only require confidence intervals, and it provides theoretical guarantees even in cases where the GP prior and likelihood are non-conjugate such as preference modeling.

Let $D_{\mathcal{Q}_t^u} := (x_\tau, x_\tau', \mathbf{1}_\tau^{(i)})_{\tau \in \mathcal{Q}_t^u, i \in V}$ be the **private** votes, $D_{\mathcal{Q}_t^v} := (x_\tau, x_\tau', \mathbf{1}_\tau^{(i)})_{\tau \in \mathcal{Q}_t^v, i \in V}$ be the **public** votes, $\mathbf{1}_\tau^{(i)} \in \{0,1\}$ be the realization of the Bernoulli random variable $\mathbf{1}_{x \succ x'}^{(i)}$, $[t] := \{1,\ldots,t\}$, $\mathcal{Q}_t^v := \{\tau \in [t-1] \mid \text{if } v \text{ is queried in step } \tau\}$ be public queries, $\mathcal{Q}_t^u$ be private queries $(t \geq |\mathcal{Q}_t^u|, t \geq |\mathcal{Q}_t^v|)$. We use capitals, e.g., $X_{\mathcal{Q}_t^v}$, for the set $(x_\tau)_{\tau \in \mathcal{Q}_t^v}$. See Appendix A.1 for the summarised table of notations.

**Assumption 3.4** (**Public and private votes**). *While private votes reflect the social-influence-free utility $u$, public votes reflect the (possibly) influenced utility $v$. The query costs satisfy the relationship $\lambda_u \gg \lambda_v$.*

Assumption 3.4 relaxes the typical strong assumption in PBO that votes always reflect the social-influence-free utility. The cost suggests $|\mathcal{Q}_t^u| \ll |\mathcal{Q}_t^v|$ is more cost-effective. This setting is so-called

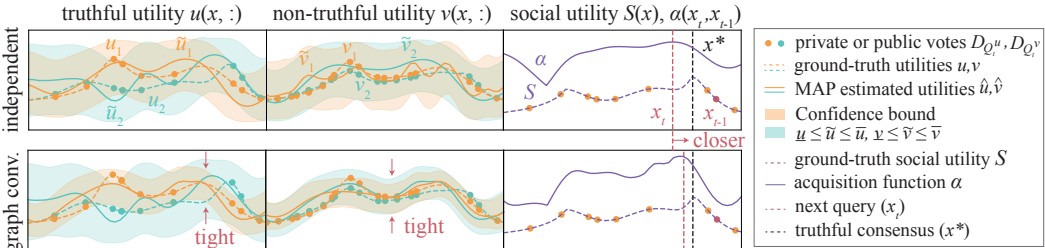

Figure 2: The dotted and solid lines represent the ground truth $u, v$ and estimated utilities $\tilde{u}, \tilde{v}$, respectively, with the shaded area indicating the confidence interval. Dots mark the queried points, where $|\mathcal{Q}_t^u| = 10$ and $|\mathcal{Q}_t^v| = 20$. Since utility values are not directly observable, the dots are for visual guidance. The acquisition function $\alpha(\cdot, x_{t-1})$ represents the upper confidence bound of improvement from the previous query $x_{t-1}$, with its argmax determining the next query $x_t$ (red vertical line). While the independent model assumes $\tilde{u} \in \mathcal{B}^u$ and $\tilde{v} \in \mathcal{B}^v$, the graph convolution model enforces a linear constraint $\tilde{u}, \tilde{v} \in \mathcal{B}^{u,A,v}$. This yields a tighter predictive interval, resulting in a next query closer to the true consensus $x^\star$ (black vertical line).

*heterotopic* data if each vote is to be associated with a (partially) different set of pairs (Wackernagel, 2003).

**Graph prior.** We introduce the regularised row-wise Dirichlet prior for graph $A$:

$$p(A) = \frac{1}{Z} \exp(-\xi \|A\|_F^2) \prod_{i \in V} \text{Dirichlet}(A_i; \kappa_i), \ \text{s.t.} \ \min_{i,j \in V} A_{ij} > \delta_A, \tag{4}$$

where $A_i$ is the $i$-th row, $\kappa_i > 1$ is the concentration parameter vector, $\delta_A > 0$ is a small positive constant, ensuring strict positivity, $\|\cdot\|_F$ denotes the Frobenius norm, $\xi$ is a small positive constant, and $Z$ is the normalising constant. The exponential term is a Tikhonov regularization that encourages invertibility, and the Dirichlet distribution assures $\sum_{j=1}^n A_{ij} = 1$.

**Likelihood modelling.** Under Assumptions 2.2, 3.4 and Definition 2.5, we define the likelihood as below. See Appendix D.1 for the derivation.

**Corollary 3.5 (Bradley-Terry model).** *Given dataset $D_{\mathcal{Q}_t^u}$ and corresponding utility function $u$, the log-likelihood (LL) for an estimate function $\hat{u} \in \mathcal{H}_{k_i}$ is given by Bradley & Terry (1952):*

$$\ell_t(\hat{u}(\cdot, i) \mid D_{\mathcal{Q}_t^u}^{(i)}) = \sum_{\tau \in \mathcal{Q}_t^u} \left[ \hat{u}(x_\tau, i) \boldsymbol{I}_\tau^{(i)} + \hat{u}(x'_\tau, i)(1 - \boldsymbol{I}_\tau^{(i)}) \right] - \sum_{\tau \in \mathcal{Q}_t^u} \log \left[ \exp(\hat{u}(x_\tau, i)) + \exp(\hat{u}(x'_\tau, i)) \right].$$

**Bayesian modelling.** The joint LL of $D_{\mathcal{Q}_t^v}, D_{\mathcal{Q}_t^u}$ are $\ell_t(\hat{u}, \hat{A}, \hat{v} \mid D_{\mathcal{Q}_t^v}, D_{\mathcal{Q}_t^u}) := \ell_t(\hat{v} \mid D_{\mathcal{Q}_t^v}) + \ell_t(\hat{u} \mid D_{\mathcal{Q}_t^u})$. For the prior, by range preservation Lemma 2.7, we can set the uniform prior $p(u) = p(v) = \mathcal{U}(u; -L_v, L_v)$, and Eq. (4) for $p(A)$. Then, the (unnormalised) log posterior becomes: $\mathcal{L}_t(\hat{u}, \hat{A}, \hat{v}) := \ell_t(\hat{u}, \hat{A}, \hat{v} \mid D_{\mathcal{Q}_t^v}, D_{\mathcal{Q}_t^u}) + \log p(u) + \log p(v) + \log p(A)$, and $\hat{A}$ can be estimated via linear constraint $\hat{u} = A\hat{v}$ when solving maximisation problem. Maximum a posteriori (MAP) offers Bayesian point estimation for $u$, $A$, and $v$; $\hat{\mathcal{L}}_t := \mathcal{L}_t(\hat{u}_t, \hat{A}_t, \hat{v}_t) = \max_{\tilde{u}, \tilde{A}, \tilde{v} \in \mathcal{B}^{u,A,v}} \mathcal{L}_t(\tilde{u}, \tilde{A}, \tilde{v})$.

**Optimistic MAP.** We apply the optimistic MLE approaches (Liu et al., 2023; Emmenegger et al., 2024; Xu et al., 2024b) to MAP objective to quantify uncertainty using a confidence set. This enables us to derive theoretical confidence bounds, and accurately compute the upper bound as efficient optimisation problem, even for non-conjugate case such as preference likelihood.

**Lemma 3.6 (MAP-based confidence set).** *For all $\delta > 0$, we have*

$$\mathcal{B}_t^v = \{\tilde{v} \in \mathcal{B}^v \mid \ell_t(\tilde{v} \mid D_{\mathcal{Q}_t^v}) \geq \hat{\ell}_t^v - \beta_t^v\}, \ \mathcal{B}_t^u = \{\tilde{u} \in \mathcal{B}^u \mid \ell_t(\tilde{u} \mid D_{\mathcal{Q}_t^u}) \geq \hat{\ell}_t^u - \beta_t^u\},$$

$$\mathcal{B}^{u,A,v} = \{\tilde{u} \in \mathcal{B}^u, \tilde{v} \in \mathcal{B}^v, \tilde{v} = \tilde{A}\tilde{u}\}, \ \mathcal{B}_{t+1}^{u,A,v} := \left\{ \tilde{u}, \tilde{A}, \tilde{v} \in \mathcal{B}^{u,A,v} \mid \mathcal{L}_t(\tilde{u}, \tilde{A}, \tilde{v}) \geq \hat{\mathcal{L}}_t - \beta_t \right\},$$

*with $\mathbb{P}\left( u, A, v \in \mathcal{B}_{t+1}^{u,A,v}, \forall t \geq 1 \right) \geq 1 - \delta$, where $\hat{\ell}_t^v, \hat{\ell}_t^u$ are the MLE of LLs, $\beta_t^u, \beta_t^v$ are selected as in Xu et al. (2024b), $|\mathcal{Q}_t^{uv}| := |\mathcal{Q}_t^u| + |\mathcal{Q}_t^v|$, and $\beta_t = n\epsilon C_L^v |\mathcal{Q}_t^{uv}| + \sqrt{64n^2 L_v^2 |\mathcal{Q}_t^{uv}| \log \frac{\pi^2 |\mathcal{Q}_t^{uv}|^2 \mathcal{N}(\mathcal{B}^v, \epsilon, \|\cdot\|_\infty)}{6\delta}}$.*

The proof and notations are in Appendix D.3. See Fig. 2 for intuition. As introduced in Assumption 2.6, while the function $\tilde{u}(\cdot, i)$ was originally in a broader set of RKHS functions $\tilde{u}(\cdot, i) \in \mathcal{B}^u$, it is now in a smaller set defined as $\tilde{u} \in \mathcal{B}_{t+1}^u$ conditioned on the preference feedback. Intuitively, with limited data, the MAP may be imperfect. Hence, it is reasonable to suppose that $\mathcal{B}_{t+1}^u$, bounded by MAP values 'slightly worse' than the MAP, contains the ground truth with high probability.

---

**Algorithm 1** Social Bayesian Optimisation (SBO)

---

1: **Input**: decay rate $0 \leq q \leq 1$
2: Set $\mathcal{Q}_0^u = \emptyset$, $\mathcal{Q}_0^v = \emptyset$, $\mathcal{B}_1^{u,A,v} = \mathcal{B}^{u,A,v}$, and draw the initial point $x_0 \in \mathcal{X}$.
3: **for** $t \in [T]$ **do**
4:     Solve $x_t = \arg\max_{x \in \mathcal{X}} \alpha(x, x_t')$ s.t. $x_t' = x_{t-1}$     ▷ Maximising the acquisition function
5:     Query public vote on the pair $(x_t, x_t')$, and obtain $\mathbf{1}_t^v := \{\mathbf{1}_t^{(i)}\}_{i=1}^n$.
6:     Update $D_{\mathcal{Q}_t^v} = D_{\mathcal{Q}_t^v} \cup (x_t, x_t', \mathbf{1}_t^v)$ and confidence set $\mathcal{B}_{t+1}^{u,A,v}$.
7:     **if** $w_t^u(x_t, x_t') \geq \max\left\{\frac{1}{t^q}, \ w_t^v(x_t, x_t')\right\}$ **then**     ▷ Stopping criterion
8:         Query private vote on the pair $(x_t, x_t')$, and obtain $\mathbf{1}_t^u := \{\mathbf{1}_t^{(i)}\}_{i=1}^n$.
9:         Update $D_{\mathcal{Q}_t^u} = D_{\mathcal{Q}_t^u} \cup (x_t, x_t', \mathbf{1}_t^u)$ and confidence set $\mathcal{B}_{t+1}^{u,A,v}$.

---

*Remark* 3.7 (**Confidence bound**). By Lemma 3.6, we define the pointwise confidence bound for $u_i \in \mathcal{H}_{k_i}$ as $\underline{u}_t(x, i) \leq u(x, i) \leq \bar{u}_t(x, i)$ where $\underline{u}_t(x, i) := \inf_{\tilde{u}_i \in \mathcal{B}_t^{u_i}} \tilde{u}(x, i)$, $\bar{u}_t(x, i) := \sup_{\tilde{u}_i \in \mathcal{B}_t^{u_i}} \tilde{u}(x, i)$. Similarly, the confidence bound for $\mathcal{S}(x)$ is $[\mathcal{A}[\underline{u}_t(x, :)], \mathcal{A}[\bar{u}_t(x, :)]]$ by the monotonicity of GSF (see Proposition 2.4).

*Remark* 3.8 (**Prediction via optimisation**). Given a prediction point $x$, the upper confidence bound (UCB) $\bar{u}_t(x, :)$ can be estimated via finite-dimensional optimisation; see Appendix E for details.

### 3.3 Proposed Algorithm

Algorithm 1 summarises our SBO algorithm. Line 4 finds the next vote $(x_t, x_t')$ by maximising the acquisition function. Line 7 outlines the stopping criterion, which halts the expensive private queries when the graph $A$ is accurately estimated. In Algorithm 1, the condition $|\mathcal{Q}_t^v| = t$ and $|\mathcal{Q}_t^v| \geq |\mathcal{Q}_t^u|$ are satisfied. The parameter $q$ is the decay rate which balances the trade-off between private votes querying cost and consensus convergence rate.

**Acquisition function.** We propose an optimistic algorithm, akin to GP-UCB (Srinivas et al., 2010):
$$\alpha(x, x') = \max_{\tilde{u} \in \mathcal{B}_t^u} \mathcal{A}[\tilde{u}(x, :)] - \mathcal{A}[\tilde{u}(x', :)]. \tag{5}$$

Intuitively, this acquisition function operates similarly to the expected improvement approach. The second term represents the best observed points so far, while the overall maximization identifies the highest potential improvement from a given point.

**Stopping criterion.** We introduce the projection weight function $w_t$ as the uncertainty criteria. We can then estimate the uncertainty of $\tilde{u}$ and $\tilde{v}$ when projecting to the comparison pairs $(x_t, x_t')$,
$$w_t^v(x_t, x_t') = \sup_{\tilde{v}, \tilde{v}' \in \mathcal{B}_t^v} \|\tilde{v}(x_t, :) - \tilde{v}(x_t', :) - (\tilde{v}'(x_t, :) - \tilde{v}'(x_t', :))\|, \tag{6}$$

and same notation rule applies to $w_t^u$. $u$ is more uncertain than $v$ due to the smaller sample size and the uncertainty in estimating $A$. When $w_t^u$ becomes smaller, $A$ is confidently estimated at the point $(x_t, x_t')$ thus no more $u$ queries are needed. The decay rate $q$ controls the threshold to satisfy this stopping condition, and we recommend $q = 1/2$ (detailed later in Section 3.4).

**Efficient computation.** So far, we have introduced several optimization problems (MAP, Prob (5), Remark 3.8, Line 4 in Alg. 1, and Probs. (6)). However, the functions $\tilde{u}$ and $\tilde{v}$ exist in an infinite-dimensional space. Fortunately, by applying the representer theorem (Schölkopf et al., 2001) and utilizing the RKHS property, we can kernelize these problems into tractable, finite-dimensional optimization problems. For example, MAP and Prob. (5) become $n(t + n)$ and $n(t + n + 1)$-dimensional optimization problems, respectively. The convexity of the kernelized problems allows for scalable solutions, although the computational cost scales as $\mathcal{O}(n(t + n))$. Notably, this cost is still more efficient than the multi-task GP, which scales as $\mathcal{O}(n^3 t^3)$ and involves non-convex optimization (Bonilla et al., 2007). See Appendix E for details on the kernelized problems.

### 3.4 Theoretical Analyses

**Cumulative regret and cost.** We define two performance metric: cumulative regret $R_T := \sum_{t=1}^T (\mathcal{A}[u(x^*, :)] - \mathcal{A}[u(x_t, :)])$ and cumulative private query $|\mathcal{Q}_T^u|$. Under Assumption 3.4 that $\lambda_u \gg \lambda_v$, $|\mathcal{Q}_T^u|$ dominates the cost $C_T := \lambda_u |\mathcal{Q}_T^u| + \lambda_v T$.

**Graph properties.** The regret bound is affected by the following properties of adjacency matrix $A$:

    (a) **Given**: We know $A$, otherwise $A$ is unknown a priori and we need to estimate it.
    (b) **Invertible**: There exists $A^{-1}$, otherwise $A$ is singular.
    (c) **Identifiable**: $A$ is identifiable from the votes, otherwise dual voting cannot identify true $A$.

**Theorem 3.9** (**Regret bound**). *Under Assumptions 2.2 to 2.6, Algorithm 1 satisfies,*

| *Voting scheme and Assumptions* | *cumulative regret* $R_T$ | *sample complexity of private votes* $|\mathcal{Q}_t^u|$ |
|---|---|---|
| *Oracle (public only)* (a)(b)(c) | $\mathcal{O}\left(nL_{\mathcal{A}}\sqrt{\beta_T^v\gamma_{\mathcal{Q}_T}^{vv'}T}\right)$ | $0$ |
| *SBO (dual voting)* (c) | $\mathcal{O}\left(L_{\mathcal{A}}T^{1-\frac{q}{4}} + L_{\mathcal{A}}\sqrt{(\beta_T^u\gamma_T^{uu'} + \beta_T^v\gamma_T^{vv'})T}\right)$ | $\mathcal{O}\left(T^q\left(\gamma_T^{vv'}\right)^2\log\frac{T\mathcal{N}(\mathcal{B}^u,1/T,\|\cdot\|_\infty)}{\delta}\right)$ |
| *None (private only)* - | $\mathcal{O}\left(L_{\mathcal{A}}\sqrt{\beta_T^u\gamma_{\mathcal{Q}_T}^{uu'}T}\right)$ | $T$ |

*with probability at least $1-\delta$, where $L_{\mathcal{A}} := \sqrt{n}\|\boldsymbol{w}\|$, $\gamma$ is maximum information gain, and $\gamma_{\mathcal{Q}_T}^{vv'}, \gamma_{\mathcal{Q}_T}^{uu'}$ are $\gamma$ for the corresponding kernel, $0 \leq q \leq 1$ is a user-defined parameter.*

*Remark* 3.10 (**Groupthink-proof**). Any identifiable $A$ has sublinear convergence to true consensus $x^\star$. Without (c) (='None'), we end up querying private votes only ($|\mathcal{Q}_T^u| = T$).

*Remark* 3.11 (**Trade-off in** $q$). Larger $q$ can make $R_T$ converge faster, yet requires more $|\mathcal{Q}_T^u|$.

Appendix F provides the proof and more details. Furthermore, by incorporating maximum information gain bounds (Srinivas et al., 2012; Vakili et al., 2021) and covering number bounds (Wu, 2017; Xu et al., 2024a; Bull, 2011; Zhou, 2002), we apply Theorem 3.9 to derive the kernel-specific bounds in Table 1, omitting $T$-independent constants. The convergence rate clarifies several aspects of Algorithm 1. While 'None' achieves the tightest regret bound $R_T$, its sample complexity $|Q_t^u|$ is the worst, scaling linearly with $T$. In contrast, SBO achieves a sublinear sample complexity bound, but the $R_T$ has an additional $T^{1-q/4}$ term. This is reasonable, as later rounds must infer consensus from corrupted queries. We can view $q$ as a necessary compromise for the graph estimation error, supported by the impossibility theorem. Importantly, within the typical conditions ($T \leq 1000, d \geq 2$, RBF kernel), $q$ dependent term is always smaller than non-dependent ones in $R_T$, suggesting a similar convergence rate in practice. The oracle setting has an additional $n$ factor compared to the 'None' case because the norm of the inverse of the graph adjacency matrix, $\|A^{-1}\|$, amplifies the noise of $v$ when it is transformed onto $u$. $\beta_T^u$ and $\beta_T^u$ reflect the utility function estimation error (see Lemma 3.6). The aggregation function $\mathcal{A}$ affects $R_T$, with egalitarian being the slowest ($L_{\mathcal{A}} \to \sqrt{n}$) and utilitarian the fastest ($L_{\mathcal{A}} = 1$). $R_T$ is sublinear to the number of agents. i.e., $\sqrt{n}$. The dimension $d$ scales similarly to standard BO, techniques like additive kernels (Kandasamy et al., 2015) can improve dimensional scalability. Furthermore, by applying the minimum excess risk (MER; Xu & Raginsky (2022)), our optimistic MAP has the following asymptotic convergence rate:

**Theorem 3.12** (**Graph identification**). *Asymptotic convergence rates of the estimation errors are*

$$\|\tilde{A}_t - A\| \leq \mathcal{O}\left(2^{-1/2}|\mathcal{Q}_t^u|^{-1/2}\right), \quad |\tilde{u}_t(x_t) - \tilde{u}_t(x_t') - (u(x_t) - u(x_t'))| \leq \mathcal{O}\left(L_{k,\mathcal{Q}_t^u}^{1/4}|\mathcal{Q}_t^u|^{-1/4}\right),$$

*for $\max_{i \in V} \kappa_i = 1 + \delta_A^2/n^2 - 2\xi\delta_A^2$, $\kappa_i > 1$, $\xi < 1/2n^2$ and $\tilde{A}_t, \tilde{u}_t \in \mathcal{B}_t^{u,A,v}$ defined in Lemma 3.6.*

Appendix G provides the proof and more details. Theorem 3.12 shows the graph identification error converges faster than pointwise utility estimation error in the worst-case scenarios. Intuitively, the estimation error for the linear model ($A$) converges faster than for the black-box non-linear functions $(u, v)$. Thus, we can stop querying private votes $|\mathcal{Q}_t^u|$ once we reach sufficient confidence. This results also suggests $q = 1/2$ can be optimal as the cumulative error of $\sum_{i \in [T]} t^{-1/2} \leq \mathcal{O}(T^{1/2})$ to match the rate of $|\mathcal{Q}_t^u|$. This convergence rate relies on the strong convexity of MAP objective (c.f.,

Table 1: Kernel-specific bounds for main algorithm where $\nu > d/4(3 + d + \sqrt{d^2 + 14d + 17}) = \Theta(d^2)$.

| Metric | Linear | RBF | Matérn |
|---|---|---|---|
| $R_T$ | $\mathcal{O}\left(T^{1-\frac{q}{4}} + T^{\frac{3}{4}}(\log T)^{\frac{3}{4}}\right)$ | $\mathcal{O}\left(T^{1-\frac{q}{4}} + T^{\frac{3}{4}}(\log T)^{\frac{3}{4}(d+1)}\right)$ | $\mathcal{O}\left(T^{1-\frac{q}{4}} + T^{\frac{d}{4\nu}+\frac{d(d+1)}{4\nu+2d(d+1)}}(\log T)^{\frac{3}{4}}\right)$ |
| $|\mathcal{Q}_T^u|$ | $\mathcal{O}\left(T^q(\log T)^3\right)$ | $\mathcal{O}\left(T^q(\log T)^{3(d+1)}\right)$ | $\mathcal{O}\left(T^{q+\frac{d}{\nu}\frac{2d(d+1)}{2\nu+d(d+1)}}(\log T)^3\right)$ |

MLE is convex yet not strongly convex). As such, Theorem 3.12 also highlights the benefits of the MAP extension, beyond just improving numerical stability through regularizers.

## 4 RELATED WORKS

**BO for black-box games.** In the BO community, the game-theoretic approach has been researched as the *application* to multi-objective BO (MOBO) (Hernández-Lobato et al., 2016; Daulton et al., 2020). Direct consideration of the multi-agentic scenario is limited, which typically assumes the specific aggregation rule (Nash equilibrium; Al-Dujaili et al. (2018); Picheny et al. (2019); Han et al. (2024), Kalai-Smorodinsky solution; Binois et al. (2020)) or Chebyshev scalarisation function (Astudillo et al., 2024), which requires discrete domain to ensure the existence of solution. Ours is the *first-of-its-kind* principled work that addresses the influenced votes over continuous space.

**Preferential BO.** Preferential BO is a single-agentic preference maximisation algorithms (González et al., 2017; Astudillo et al., 2023; Xu et al., 2024b), extended to diverse scenarios; choice data (Benavoli et al., 2023a), top-$k$ ranking (Nguyen et al., 2021), preference over objectives on MOBO (Abdolshah et al., 2019; Ozaki et al., 2024), human-AI collaboration (Adachi et al., 2024b). Our work is the first to study the multi-agentic social influence, and is orthogonal to these works.

**Other BO.** Multitask BO (Kandasamy et al., 2016) addresses scenarios where cheap but lower-fidelity information is available and leverages this information to identify promising regions for exploration. However, when the low-fidelity information is unreliable, these algorithms may converge much more slowly than the original BO algorithm (Mikkola et al., 2023). In contrast, our SBO framework does not use public votes as constraints but rather as supplementary data to help identify the underlying model, i.e., the social influence graph $A$. While cost-aware BO (Lee et al., 2020) deals with location-dependent cost functions, public votes in our model are not location-dependent. Misspecified BO (Bogunovic & Krause, 2021; Berkenkamp et al., 2019), which addresses cases where the surrogate model is misspecified, primarily focuses on Gaussian process hyperparameters and is not directly applicable to the likelihood-ratio model.

## 5 EXPERIMENTS

Since our problem setting—optimization with preference feedback under social influence—is novel, we benchmark our proposed algorithm against simpler versions of our method by systematically removing one design choice at a time. An RBF kernel is used by default unless otherwise specified, and for each optimization iteration, the inputs are rescaled to the unit cude $[0, 1]^d$. The initial dataset consist of 5 randomly sampled pairs $(x_t, x_t')$ from the domain $\mathcal{X}$ with labels generated according to Assumption 2.2. A $(x_t, x_t')$ is determined by $\arg\max_{x,x' \in \mathcal{X}} \alpha(x, x')$ in each iteration then votes are queried. All experiments were repeated 10 times under different seeds and initial datasets. Hyperparameters such as kernel lengthscales, the norm bound $L_v$, and the confidence bound $\beta_t$, were tuned online at each iteration (c.f. Appendix I for details). Optimization problems are solved using the interior-point nonlinear optimizer IPOPT (Wächter & Biegler, 2006), interfaced via the symbolic framework CasADi (Andersson et al., 2019). Code repository[1] are provided for reproducibility. Models are implemented in GPyTorch (Gardner et al., 2018), and experiments are conducted using a laptop PC[2]. Computational time is discussed in Appendix J.4. Along with cumulative regret and query counts, we also report simple regret, defined as $\text{SR}_t := \min_{\tau \in [T]} (\mathcal{S}(x^\star) - \mathcal{S}(x_\tau))$.

**Baseline setup.** We benchmark our algorithm against four baseline models, each based on specific assumptions about the aggregation function and influence graph, as considered in Theorem 3.9: **(a) Oracle**: $A$ is known, thus only $v$ is queried; **(b) Single Agent**: Ignores agent heterogeneity, performing single-agent preference maximization (similar to standard preferential BO); **(c) Independence**: Assumes $u$ and $v$ are completely independent; **(d) Optimistic MLE**: Assumes $v = Au$ without a prior on $A$; and **(e) Optimistic MAP (ours)**: Assumes $v = Au$ with the prior $p(A)$ in Eq. (4).

**Robustness to aggregate function.** We first demonstrate the robustness of our algorithm against different aggregation functions under a fixed influencer-follower graph, $A = \left( \begin{smallmatrix} 0.9 & 0.1 \\ 0.6 & 0.4 \end{smallmatrix} \right)$, where the first agent (influencer) prioritizes their own utility 9 times more than the second's, while the second (follower) values the influencer's utility 1.5 times more than their own. Overall, the egalitarian case

---

[1] `https://anonymous.4open.science/r/socialBO-1EED/`
[2] MacBook Pro 2019, 2.4 GHz 8-Core Intel Core i9, 64 GB 2667 MHz DDR4

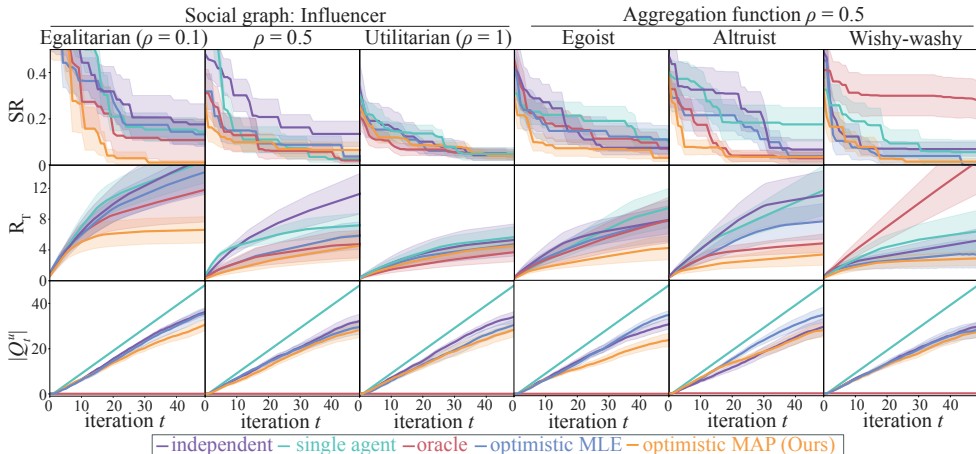

Figure 3: Robustness analysis was conducted using the function shown in Fig. 2. The lines and shaded areas represent the mean $\pm 1$ standard error. The cumulative regret $R_T$ reaches a plateau, confirming the no-regret property, while the cumulative queries $|\mathcal{Q}_t^u|$ demonstrate sublinear convergence.

($\rho = 0.1$) shows the slowest $R_T$ convergence, while the utilitarian case ($\rho = 1$) is the fastest. The cumulative queries $|\mathcal{Q}_t^u|$ remain unaffected, supporting Theorem 3.9. The egalitarian case displays the most diverse results among the baselines, as this rule prioritizes the worst-off agent—in this case, the follower—making accurate prediction of the follower's utility key to faster convergence. Interestingly, our optimistic MAP model outperforms the oracle model. While the oracle model does not use private votes ($|\mathcal{Q}_t^u|$ stays at zero), our model accesses both public and private votes, utilizing more data and improving early-stage predictions. In contrast, the utilitarian rule focuses on the average utility among agents, making follower utility prediction less critical for convergence.

**Robustness to influence graph.** Under a fixed aggregation rule ($\rho = 0.5$), three cases were considered: egoist $A = \begin{pmatrix} 1-10^{-10} & 10^{-10} \\ 0.3 & 0.7 \end{pmatrix}$, which strongly prioritizes self-utility; altruist $A = \begin{pmatrix} 10^{-10} & 1-10^{-10} \\ 0.3 & 0,7 \end{pmatrix}$, which selflessly prioritizes others' utility; and wishy-washy $A = \begin{pmatrix} 0.5 & 0.5 \\ 0.5 & 0.5 \end{pmatrix}$, where agents are indecisive. Notably, the oracle performs the worst in the wishy-washy case due to the singularity of the matrix, which makes it impossible to identify $u = A^{-1}v$ through inversion. In the altruist case, where a selfless agent causes the public votes to be unanimously influenced, the scenario ironically becomes one of the most challenging for the single-agent baseline, which assumes homogeneous agents. In contrast, our method remains unaffected by the structure of graph $A$, as it does not rely on invertibility or homogeneity assumptions.

**Additional experiments.** Due to page limitations, we defer additional experimental results to Appendix J. The computational time, provided in Appendix J.4, shows that the overhead of our method is comparable to that of simpler baselines. Furthermore, we detail a GP-based variant of our algorithm using convolutional kernel in Appendix H.1. We did not adopt this as the main algorithm because the non-conjugate posterior makes the theoretical analysis challenging. While this GP-based approach is heuristic, it allows for efficient approximation methods, such as the Laplace approximation, making it a simpler and faster alternative via convolutional kernel.

**Real-world tasks.** We introduce four new real-world collective-decision tasks to test our algorithm. For all tasks, we assume that agents prefer not to disclose their social-influence-free votes to other agents in public voting, but are willing to share them with the facilitator, provided the results remain closed.

**1. Thermal comfort:** Three office workers (influencer, follower, altruist) wishes to optimize the thermal condition (set temperature and air speed) under different garments and activity conditions, leading to varying thermal insulation and metabolic generation. The facilitator sets aggregation rule as egalitarian because they considers uncomfortable thermal conditions deteriorates productivity and health. We calculate each utility function under the given conditions using a simulator (Tartarini & Schiavon, 2020), following the ASHRAE industrial standard.

**2. TeamOpt:** Four office workers (a leader, a follower, and two indecisive members) hold a meeting to select team members for the next project, which includes eight workers in total, including themselves. The leader prioritizes skill set diversity to enhance productivity, while the others fo-

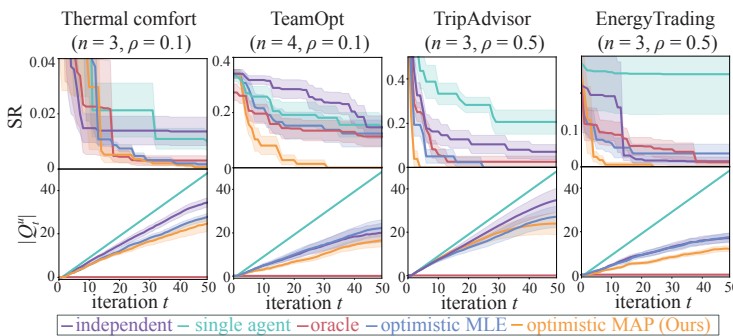

Figure 4: Real-world experiments with varying number of agents $n$ and aggregation rule $\rho$

cus on inter-member compatibility: one considers average compatibility, another seeks to maximize the worst-case scenario, and the last prefers a balanced team with minimal variance in compatibility. The facilitator applies an egalitarian rule, ensuring that the worst-off member is at least satisfied with the team. Each potential team forms a graph with an adjacency matrix representing skills and compatibility, and summary statistics define each utility function. The inter-graph kernel was computed using the graph diffusion kernel (Zhi et al., 2023).

**3. TripAdvisor:** Three colleagues (two influencers and a follower) are deciding on a hotel for their upcoming group retreat, using the TripAdvisor website. One prefers a luxurious hotel with the highest ranking, another seeks a budget-friendly option with reasonable reviews, and the third prioritizes the hotel with the highest overall review score. The facilitator (group leader) chooses $\rho = 0.5$ to balance both egalitarian and utilitarian aspects. We used the TripAdvisor New Zealand Hotel dataset (Rahman, 2023).

**4. EnergyTrading:** Three firms (a large corporation, a niche startup, and a joint venture) form a strategic collaboration to enhance their energy trading business. Their profits rely on a machine learning model that predicts day-ahead market prices, which in turn depends on a demand dataset. Due to the scarcity of such data, they jointly invest in a market research project. However, since the demand data is spatiotemporal, they must decide on the optimal location for data collection. Each firm's utility function is based on the information gain for the selected location, and their internal datasets lead to heterogeneous utilities. The facilitator (project leader) adopts $\rho = 0.5$ to balance egalitarian and utilitarian considerations. We used the UK-wide energy demand dataset (Grünewald & Diakonova, 2020; Grünewald & Diakonova, 2019).

See Appendix J for further details on experimental conditions. Fig. 4 summarizes the results. Our optimistic MAP model consistently identifies better or comparable solutions compared to other baselines. Additionally, the cumulative queries $|\mathcal{Q}_t^u|$ remained the smallest among the three adaptive query baselines. Further comparisons with multi-fidelity BO are available in Appendix J.3.

## 6 CONCLUSION AND LIMITATIONS

We explored the impact of social influence, a common but under-researched cognitive bias in collective decision-making. To bypass the impossibility of groupthink-proof aggregation, we proposed a dual voting mechanism and developed a learning algorithm for social influence graphs using optimistic MAP, which accelerated social-influence-free consensus-building across six synthetic and four real-world tasks. Our results generalize to any positive linear combination of aggregation functions and apply to both provable likelihood-ratio models and popular GP-based approaches.

While our algorithm is the *first of its kind* with a general theoretical guarantee in the social-influence setting, it shares limitations common to optimistic algorithms like GP-UCB (Srinivas et al., 2010), particularly in high-dimensional problems. Additionally, our current framework does not support batch settings, and the aggregation function must be specified in advance, assuming stationarity and homogeneity. To extend to heterogeneous and dynamic cases, the probabilistic choice function approach (Benavoli et al., 2023b) presents a promising avenue for future research. While we framed social influence as a bias to eliminate, positive social influence—such as debiasing confusion or using nudge theory (Thaler & Sunstein, 2008) to guide behavior—can also be beneficial. Since our algorithm is symmetric to $u, v$, this inverse approach is possible and a promising direction for future research, as discussed further in Appendix H.2.

## 7    ETHICS STATEMENT

Our experiments do not involve human subjects. We study social influence in general, focusing on the spontaneous aspects of human society, such as cognitive biases that arise from societal influence, leading to biased votes. We do not investigate issues related to harassment or politically incorrect behavior. Even if someone attempts to misuse the method, our approach is designed to mitigate such effects, never to act adversarially. By aiming to protect the worst-off agent, who may be a selfless altruist, our research upholds ethical standards.

## 8    REPRODUCIBILITY STATEMENT

Our code is open-sourced at `https://anonymous.4open.science/r/socialBO-1EED/` for reproducibility. The experimental details are delineated in Appendix J.

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

# Part I

# Appendix

## Table of Contents

## A Preliminary

### A.1 Table of Notations

Table 2: Notations and Descriptions (Part I)

| Category | Symbol | Description | Reference |
|---|---|---|---|
| Domain | $x$ | Option | Eq. (1) |
| | $x^*$ | Consensus (global optimum) | Eq. (1) |
| | $x_t$ | Queried option at $t$-th step | Eq. (1) |
| | $\mathcal{X} \in \mathbb{R}^d$ | (continuous) domain | Eq. (1) |
| | $d$ | Number of dimensions | Eq. (1) |
| Utility | $n$ | Number of agents | Eq. (1) |
| | $V$ | Set of $n$ agents | Eq. (1) |
| | $i$ | Index of agents | Eq. (1) |
| | $u(x, i)$ | Truthful utility for the $i$-th agent. | Eq. (1) |
| | $v(x, i)$ | Non-truthful utility for the $i$-th agent. | Eq. (1) |
| | $u(x, :)$ | Utilities of all agents | Eq. (1) |
| | $\succ_{u(\cdot, i)}$ | Preference of agent $i$ induced by utility $u(\cdot, i)$ | Assumption 2.2 |
| | $\succ_{\mathcal{S}}$ | Preference of group induced by social utility $\mathcal{S}$ | Definition 3.2 |
| | $p(u), p(v)$ | Prior over utilities (uniform prior) | Section 3.2 |
| | $\hat{u}, \hat{v}$ | MAP estimate of utilities. | Lemma 3.6 |
| | $v(x, i)$ | Non-truthful utility for the $i$-th agent. | Eq. (1) |
| | $u(x, :)$ | Utilities of all agents | Eq. (1) |
| | $\tilde{u}, \tilde{v}$ | Utility function sample from confidence set. | Assumption 2.6 |
| | $\hat{u}_t, \hat{v}_t$ | MAP estimated utility function at $t$-th step. | Assumption 2.6 |
| | $\underline{u}_t, \underline{v}_t$ | The lower confidence bound of utility. | Remark 3.7 |
| | $\overline{u}_t, \overline{v}_t$ | The upper confidence bound of utility. | Remark 3.7 |
| Likelihood | $\sigma$ | Sigmoid function | Assumption 2.2 |
| | $\ell_t(u), \ell_t(v)$ | Log Likelihood (LL) function | Corollary 3.5 |
| | $\hat{\ell}_t^u, \hat{\ell}_t^v$ | MLE estimate of LL values | Lemma 3.6 |
| | $\mathcal{L}_t(u, A, v)$ | Unnormalized negative log posterior | Section 3.2 |
| Aggregation function | $\mathcal{A}$ | Aggregation function | Definition 2.3 |
| | $\mathcal{S}$ | Social utility | Definition 2.3 |
| | $\mathbf{w}$ | Weight function of $\mathcal{A}$ | Proposition C.4 |
| | $\rho$ | GSF interpolation parameter | Proposition C.4 |
| | $\phi$ | Sorting function | Proposition C.4 |
| RKHS | $k_i(x, x')$ | Kernel of $i$-th agent's utility | Assumption 2.6 |
| | $\mathcal{H}_{k_i}$ | RKHS corresponding to the kernel $k_i$ | Assumption 2.6 |
| | $\|\cdot\|_{k_i}$ | Norm induced by inner product in RKHS $\mathcal{H}_{k_i}$ | Assumption 2.6 |
| | $L^v$ | Isotropic norm bound of $\mathcal{H}_{k_i}$ | Assumption 2.6 |
| | $\mathcal{B}^{v_i}, \mathcal{B}^{u_i}$ | Set of utility functions of $i$-th agent | Assumption 2.6 |
| | $\mathcal{B}^v, \mathcal{B}^u$ | Superset of utility functions of all agent | Assumption 2.6 |
| | $\mathcal{B}_t^v, \mathcal{B}_t^u$ | Superset of MLE-estimated utility functions at $t$-th step | Lemma 3.6 |
| | $\mathcal{B}_t^{u, A, v}$ | Superset of MAP-estimated utility functions at $t$-th step | Lemma 3.6 |
| | $\gamma_t^u, \gamma_t^v, \gamma_t^{uu}, \gamma_t^{vv}$ | Maximum information gain for corresponding kernels | Lemma 3.6 |
| | $L_{k,t}$ | Kernel specific term | Theorem 3.12 |
| | $\nu$ | Matérn kernel smoothness paramter | Table 1 |
| Confidence set | $\mathcal{F}_t$ | Filtration at the step $t$ | Lemma 3.6 |
| | $\delta$ | Probability that $\mathcal{B}_t^{u, A, v}$ does not contain $u, A, v$ | Lemma 3.6 |
| | $\epsilon$ | Radius of the function space ball $L_{k_i}$ | Lemma 3.6 |
| | $\mathcal{N}(\mathcal{B}^u, \epsilon, \|\cdot\|_\infty)$ | Covering number of the set $\mathcal{B}^u$ | Lemma 3.6 |
| | $\beta_t^u, \beta_t^v$ | MLE-based confidence set bound parameter | Lemma 3.6 |
| | $\beta_t$ | MAP-based confidence set bound parameter | Lemma 3.6 |

Table 3: Notations and Descriptions Part II

| Category | Symbol | Description | Reference |
|---|---|---|---|
| Graph | $E$ | The set of weighted directed edges. | Definition 2.5 |
| | $G = (V, E)$ | The social-influence graph | Definition 2.5 |
| | $A$ | The adjacency matrix of $G$. | Definition 2.5 |
| | $N_G(i)$ | The (in)-neighbour of $i$ in $G$. | Definition 2.5 |
| | $p(A)$ | Graph prior | Eq. (4) |
| | $\xi$ | Tiknohov parameter (invertibility regularizer) | Eq. (4) |
| | $\kappa_i$ | Dirichlet concentration parameter of $i$-th row | Eq. (4) |
| | $\delta_A$ | The smallest element of $A$ | Eq. (4) |
| | $Z$ | Normalising constant of prior $p(A)$ | Eq. (4) |
| | $\tilde{A}$ | graph sample from confidence set. | Assumption 2.6 |
| | $\hat{A}_t$ | MAP estimated graph. | Assumption 2.6 |
| Queries | $t$ | The step of iteration | Assumption 3.4 |
| | $T$ | The running horizon | Assumption 3.4 |
| | $\mathcal{Q}_t^v$ | public queries | Assumption 3.4 |
| | $\mathcal{Q}_t^u$ | private queries | Assumption 3.4 |
| | $\mathcal{Q}_t^{uv}$ | combined set of private and public queries | Lemma 3.6 |
| | $D_{\mathcal{Q}_t^v}$ | Public vote | Assumption 3.4 |
| | $D_{\mathcal{Q}_t^u}$ | Private vote | Assumption 3.4 |
| | $\lambda_u, \lambda_v$ | Query costs of private and public votes | Assumption 3.4 |
| Algorithm | $\alpha$ | Acquisition function | Eq. (5) |
| | $w_t^u, w_t^v$ | Projection weight function | Eq. (6) |
| | $q$ | Decay rate | Algorithm 1 |
| | $R_T$ | Cumulative regret | Theorem 3.9 |
| | SR | Simple regret | Section 5 |
| | $L_{\mathcal{A}}$ | Regret constant from aggregate function | Theorem 3.9 |

## A.2 HYPERPARMATER LIST

Table 4: The complete list of hyperparameters and their settings.

| hyperparameters | initial value | data-driven optimisation? | tuning method |
|---|---|---|---|
| kernel hyperparamters | GPyTorch default | ✓ | the method in Appendix I.1 |
| $\gamma_T^u, \gamma_T^v$ in Theorem 3.9 | – | ✓ | algorithm using Hong et al. (2023) |
| $\delta_A$ in Eq. 4 | 0.01 | fixed | – |
| $\xi$ in Eq. 4 | $1/2\delta_A^2 n^2$ | fixed | – |
| $\kappa_i$ in Eq. 4 | $1 + 1/n^2(2\delta^2 - 1)$ | fixed | – |
| $L_v$ in Assumption 2.6 | 1.5 | ✓ | the method in Appendix I.1 |
| $\beta_t, \beta_t^u, \beta_t^v$ in Lemma 3.6 | 0.5 | ✓ | the method in Appendix I.1 |
| $q$ in line. 7 in Alg. 1 | 0.5 | fixed | – |

## A.3 DEFINITIONS

At first, we formally define the definitions we introduced:

**Definition A.1** (**Invertibility**). *The graph matrix $A$ is invertible (also known as nonsingular or nondegenerate) if there exists a square matrix $B$ such that*

$$AB = BA = I, \tag{7}$$

*and we denote such $B$ as $A^{-1}$.*

**Definition A.2** (**Identifiability**). *Let $\mathcal{M} = \{\mathcal{M}_A : v(x, :) = Au(x, :) \mid A \in \mathbb{R}^{n \times n}\}$ be a statistical model of social influence. $\mathcal{M}$ is identifiable if the mapping $A \mapsto \mathcal{M}_A$ is one-to-one:*

$$\mathcal{M}_{A_1} = \mathcal{M}_{A_2} \Rightarrow A_1 = A_2, \quad \forall A_1, A_2 \in \mathbb{R}^{n \times n}. \tag{8}$$

In an unidentifiable case, the above relationship is not one-to-one. Such cases can be found. For instance, if $v_i = u_j = u, \forall i, j$, then matrix $A$ can be any matrix with $\sum_j A_{ij} = 1$. This assumption excludes such cases.

**Definition A.3** (**Strong convexity**). *A function $h(x)$ is said to be strongly convex with parameter $m > 0$ if, for all matrices $A, B \in \mathbb{R}^{n \times n}$, the following inequality holds:*

$$h(B) \geq h(A) + \nabla h(A)^\top (B - A) + \frac{m}{2} \|B - A\|^2, \tag{9}$$

*where $\nabla h(A)$ denotes the gradient of $h$ at $A$.*

**Definition A.4** (**Monotonicity**). *For any $t \in [T], i \in V, x, x' \in \mathcal{X}$, a social utility $\mathcal{S}$ is monotonic, $\mathcal{S}(x) > \mathcal{S}(x')$ if:*

$$(u(x, i) > u(x', i)) \wedge (\forall j \in V \backslash \{i\} \ u(x, j) = u(x', j)).$$

Intuitively, a social utility $\mathcal{S}(x)$ should increase if any individual utility $u_t(x, i)$ improves. Monotonicity guarantees the maximising the social utility leads to the improvement of individual utilities.

**Definition A.5** (**Pigou-Dalton principle (PDP)**). *An social utility $\mathcal{S}$ satisfies $\mathcal{S}(x) > \mathcal{S}(x')$ if:*

*1. **In a strong sense**, $u(x, :) \succ u(x', :)$ for $x \in \mathcal{X}$ whenever there exist $i, j \in V$ such that (a) $\forall k \in V \backslash \{i, j\} \ u(x, k) = u(x', k)$, (b) $u(x, i) + u(x, j) = u(x', i) + u(x', j)$, and (c) $|u(x', i) - u(x', j)| > |u(x, i) - u(x, j)|$.*
*2. **In a weak sense**, we only require that $u(x, :) \succeq u(x', :)$.*

Intuitively, given a utility vector $u(x, :)$, if an agent with a higher utility transfers $\leq 1/2$ of its excess utility to another worse-off agent, the aggregate function $\mathcal{A}$ should prefer the transferred utility over the original for the fairness.

# B  GRAPH PROPERTIES

## B.1  PROOF OF LEMMA 2.7

We begin by introducing useful lemmas.

**Lemma B.1** (**Utility bound**). $\forall u \in \mathcal{B}^u, x \in \mathcal{X}, i \in V$, *the utility is bounded by $u(x, i) \in [-L_u, L_u]$.*

*Proof.*

$$\begin{aligned}
|u(x)| =& |\langle u, k(x, \cdot) \rangle|, & (10) \\
\leq & \|u\| \|k(x, \cdot)\|, & \text{(Cauchy-Schwarz)} \\
\leq & L_u \sqrt{k(x, \cdot)}, & \text{(Assumption 2.6)} \\
\leq & L_u. & \text{(Assumption 2.6)}
\end{aligned}$$

$\square$

**Lemma B.2** (**Range preservation**). *Under Eq. (3) and Assumption 2.6, both $u, v$ are bounded by the same range $u, v \in [-L_v, L_v]$.*

*Proof.* Given the conditions $A_{ij} \geq 0$ and $\sum_{j=1}^n A_{ij} = 1$, each $v(\cdot, i) = \sum_{j=1}^n A_{ij} u(\cdot, j)$ is a convex combination of the $u(\cdot, j)$. The convex combination ensures that,

$$\min_j u(\cdot, j) \leq v(\cdot, i) \leq \max_j u(\cdot, j) \tag{11}$$

By Lemma B.1, $u(\cdot, i) \in [-L_u, L_u]$. Therefore, $v(\cdot, i) \in [-L_u, L_u]$. $\square$

**Lemma B.3** (**Matrix norm bound**). *Under Eq. (3), if the graph matrix $A$ is invertible, the Euclidean norm of the inverse matrix satisfies*

$$1 \leq \|A^{-1}\| \leq n. \tag{12}$$

*Proof.* **Lower bound.** Let $\lambda_1 \geq \lambda_2 \geq \cdots \geq \lambda_n \geq 0$ be the eigenvalues of $A$. The Euclidean norm $\|A^{-1}\|$ is the largest singular value of $A^{-1}$, which, for symmetric matrices, is the largest eigenvalue of $A^{-1}$, therefore $\|A^{-1}\| = 1/\lambda_n$. The condition $\sum_{j=1}^{n} A_{ij} = 1$ is known as row-stochastic matrix. The Perron-Frobenius theorem (Meyer, 2023) ensures that such a matrix has a unique largest eigenvalue. The row-stochastic condition can be understood as $Ae = e$, where $e := [1, \cdots, 1]^\top$ is the column vector, implying $e$ is an eigenvector of $A$ with eigenvalue 1. Therefore, $\lambda_1 = 1$ holds, thereby $\lambda_n \leq 1$. As such, the lower bound: $\|A^{-1}\| \geq 1/\lambda_n = 1$ is obtained.

**Upper bound.** We consider the operator norm induced by the Euclidean norm $\|\cdot\|_2$, which is defined as:

$$\|A^{-1}\| = \sup_{\|x\|_2 = 1} = \|A^{-1}x\|_2. \tag{13}$$

To find an upper bound, it is helpful to use properties of induced norms. Specifically, for any matrix $M$, we have:

$$\|M\|_2 \leq \|M\|_1, \tag{14}$$
$$\|M\|_2 \leq \|M\|_\infty, \tag{15}$$

where $\|M\|_1$ is the maximum absolute column sum, and $\|M\|_\infty$ is the maximum absolute row sum. Since $A$ is row-stochastic matrix, $\|A\|_\infty = 1$.

We can consider $B = A^{-1} = (B_{ij})_{i,j \in V}$:

$$\|B\|_1 = \max_{j \in V} \sum_{i \in V} |b_{ij}|. \tag{16}$$

Given that $A$ has positive entries and is invertible, the inverse $B$ will have entries that can be bounded based on $n$. Specifically, by leveraging properties of positive matrices and norm inequalities, it can be shown that:

$$\|B\|_1 \leq n. \tag{17}$$

Consequently,

$$\|B\|_2 \leq \|B\|_1 \leq n. \tag{18}$$

$\square$

**Lemma B.4** (Identifiability). *Under Eq. (3), given the full rank observation dataset $(u(x_\tau, :))_{\tau \in [T]}$ from data pairs $(v(x_\tau, :), u(x_\tau, :))_{\tau \in [T]}$, $A$ is identifiable from such dataset.*

*Proof.* Positivity constraint in Eq. (3) eliminates the possibility of multiple solutions differing by sign or magnitude in a way that violates positivity. The full rank assumption ensures each row $A_{i:}$ has a unique solution. Thus, identifiability holds regardless of the invertibility of $A$. $\square$

**Main proof.**

*Proof.* Under Assumption 2.6 and Eq. (3), range preservation is proven in Lemma B.2.

If we add another condition that matrix $A$ is invertible, Lemma B.3 shows its Euclidean norm is bounded by $1 \leq \|A^{-1}\| \leq n$.

Alternatively, if we assume the full rank condition for the observed pairs of $(u, v)$, Lemma B.4 shows $A$ is identifiable from observed datasets. $\square$

## B.2    Graph prior properties

**Lemma B.5** (Strongly convex prior). *The prior in Eq. (4) satisfies:*

    *(a) **positive matrix**: Every entry $A_{ij} > 0$.*
    *(b) **row-stochastic matrix**: Each row sums to 1, i.e., $\sum_{j=1}^{n} A_{ij} = 1$ for all $i$.*
    *(c) **strong convexity**: $-\log p(A)$ is strongly convex with regard to $A$.*

*Proof.* **(a) positive matrix.** By definition of Dirichlet distribution, all row-wise samples are nonzero. $\alpha_i > 1$ discourages sampled matrix $A$ from selecting the boundary, i.e., $A_{ij} = 0$. To exclude the small chance of zero entry, the constraint $A_{ij} > \delta_A$ and $\delta_A > 0$ directly ensures the strict positivity.

**(b) row-stochastic matrix.** By definition of Dirichlet distribution, all row-wise samples are row-stochastic.

**(d) Strong convexity.** The negative log prior is

$$-\log \mathbb{P}(A) = \sum_{i,j \in [n]} A_{ij}^2 - \sum_{i=1}^{n} \log \mathrm{Dirichlet}(a_i; \kappa_i). \tag{19}$$

The sum of strongly convex functions is strongly convex. Therefore, if all terms are strongly convex, we can say $\log \mathbb{P}(A)$ is strongly convex with regard to $A$. We can show the strong convexity if its Hessian $\nabla^2 f(x)$ satisfies:

$$\nabla^2 f(x) \succeq mI. \tag{20}$$

For the Tikhonov term, we have simple sum of squared elements, thereby its Hessian is $2\lambda I$, thus strongly convex.

For the Dirichlet term, we have

$$-\log \mathrm{Dirichlet}(a_i; \kappa_i) = -\log \left( \frac{1}{\mathbb{B}(\alpha_i)} \prod_{j=1}^{n} A_{ij}^{\kappa_{ij}-1} \right), \tag{21}$$

$$= -\log \mathbb{B}(\kappa_i) - \sum_{j=1}^{n} (\kappa_{ij} - 1) \log A_{ij}, \tag{22}$$

where $\mathbb{B}(\kappa_i)$ is the multivariate Beta function, which is a constant with respect to $A$. For each $A_{ij}$, the Hessian entry is $(\kappa_{ij}-1)/A_{ij}^2$. As $\kappa_{ij} > 1$ and $A_{ij} > 0$ for $\forall i, j \in V$, therefore the Hessian is strictly positive, thereby strongly convex. $\qquad\square$

### B.3 OTHER POSSIBLE GRAPH STRUCTURES

**Non-linear graph** For non-linear cases, we can employ a graph convolutional kernel network approach [1]. Using the kernel trick and Nyström method, this approach can transform non-linear functions into effectively linear forms. Popular graph neural network models can also be seen as special cases of [1]. Once the model is linearized, our approach can be applied directly. The linearized graph has $m$ components, where $m$ represents the number of Nyström model centroids. By applying the Cauchy-Schwarz inequality, our bound in Theorem 3.12 becomes looser by a factor of $\sqrt{m}$. However, since this is a constant, the order of the asymptotic convergence rate remains the same at $-1/2$. The Nyström method provides an eigendecomposition-based approximation of the non-linear network, where $m$ reflects the complexity of the non-linear social graph. This adjustment is reasonable, as a more complex ground-truth social graph would naturally lead to a slower convergence.

**Peer-pressure model** In a peer-pressure scenario, we assume a setting where a minority of agents may shift their votes toward the majority. This minority or majority could vary based on the option $x$, creating a heterogeneous setting. If the setting is homogeneous, our algorithm can model peer pressure directly. For a heterogeneous setting, as noted in L534-536, we can incorporate diversity using a probabilistic choice function (Benavoli et al., 2023b). While regret bounds for this model are not yet established in the literature—given that even linear graphs are novel in this context—the submodularity of the probabilistic choice function suggests sublinear convergence.

**Hierarchical influence** For hierarchical influence, we are unsure of its relevance in settings where all participants vote in the same room. This scenario may arise when influence propagates slowly among voters, as in presidential voting. Our target tasks, however, are in an online setting where

voting is iterative and occurs at a relatively faster pace than in presidential elections (see Section 5). Still, if it does occur, a grey-box Bayesian optimization approach [2] could be applied, where the hierarchical graph structure is known but the specific attributions remain unknown. Under these assumptions, we can demonstrate the same convergence rate as in Theorem 3.9 with high probability, although this analysis is beyond the scope of the current paper.

## C  AGGREGATION FUNCTION

### C.1  POPULAR AGGREGATION FUNCTIONS

We generalize the aggregation function as $v(x, i) = \sum_{i \in [n]} w(x, i) u(x, i)$. Then, we will show the popular aggregation rule can be expressed as $w(x, i) \geq 0$.

**Utilitarian aggregation**

$$w(x, i) = \frac{1}{n} \tag{23}$$

**Egalitarian aggregation**

$$w(x, i) = \begin{cases} 1 & \text{if } i = \arg\min_{i \in [n]} u(x, i), \\ 0 & \text{otherwise;} \end{cases} \tag{24}$$

$$w(x, i) = \begin{cases} 1 & \text{if } i = \arg\min_{i \in [n]} u(x, i), \\ 0 & \text{otherwise;} \end{cases} \tag{25}$$

**Chebyshev scalarisation function**

$$w(x, i) = \begin{cases} 1 & \text{if } i = \arg\min_{i \in [n]} \frac{u(x, i)}{w_i}, \\ 0 & \text{otherwise;} \end{cases} \tag{26}$$

We can understand this function is similar to egalitarian aggregation.

### C.2  PROOF OF IMPOSSIBILITY THEOREM

*Proof.* We prove the impossibility of groupthink-proofness for any aggregation rule $\mathcal{A}$ in the absence of a trivial social consensus.

**Definition C.1.** *(Trivial social consensus) $x^*_{trivial}$ is the trivial social consensus if for all $i \in |V|$, $x^*_{trivial} = \arg\max_x u(x, i)$*

Let us assume that there exists a groupthink-proof aggregation function $\mathcal{A}$ and no trivial social consensus. With respect to (Defn. 3.1), any aggregation rule $\mathcal{A}$ is groupthink-proof if for any social graph $G$

$$\arg\max_x \mathcal{A}[u(x, :)] = \arg\max_x \mathcal{A}[v(x, :)]$$

Consider a subset of all possible graphs, $G_{dictatorial} = \{G_i | \forall i, j, k \in |V| \, A_{jk} = \mathbb{I}\{i = k\}\}$. Intuitively, $G_{dictatorial}$ is the collection of social influence graphs where one agent forces everyone else to take their utility. Since $\mathcal{A}$ is groupthink proof for any $G$, it must also be groupthink proof w.r.t subset $G_{dictatorial}$.

Let us denote the social consensus of the groupthink-proof aggregation rule as

$$x^* = \arg\max_x \mathcal{A}[u(x, :)]$$

Let us iteratively compute the social consensuses obtained by aggregation of non-truthful utilities under $G_{dictatorial}$, i.e. for all $G_i \in G_{dictatorial}$,

$$\arg\max_x \mathcal{A}[v_i(x,:)] = \arg\max_x \mathcal{A}[A_i u(x,:)]$$
$$= \arg\max_x \mathcal{A}[\mathbf{1}u(x,i)]$$
$$= \arg\max_x u(x,i)$$

For $\mathcal{A}$ to be groupthink-proof for each $G_i \in G_{dictatorial}$

$$\arg\max_x \mathcal{A}[u(x,:)] = \arg\max_x \mathcal{A}[v_i(x,:)]$$
$$(\Rightarrow) \text{ for all } i \in |V| \quad x^* = \arg\max_x u(x,i)$$

This means $x^*$ is a trivial social consensus, thus resulting in a contradiction. Therefore, no aggregation rule is groupthink proof in the absence of a trivial social consensus. $\qquad\square$

The old proof is below for reference:

*Proof.* To prove impossibility, we show that for every aggregation rule $\mathcal{A}$ there exists a graph $G$ and utility profile $u(x,:)$ such that groupthink-proofness (Defn. 3.1) is not satisfied. To show this, we construct a counter-example for any aggregation rule $\mathcal{A}$.

Consider the $\succeq_{S_u}$ be a preference relation defined on $\mathcal{X}$ with respect to $S_u(x) := \mathcal{A}[u(x,:)]$. Formally, $\forall x, x' \in \mathcal{X}, x \succeq_{S_u} x'$ iff $\mathcal{A}[u(x,:)] \geq \mathcal{A}[u(x',:)]$ Since $\mathcal{A}$ ensures non-dictatorship, it implies that $\forall x, x' \in \mathcal{X}$

$$\nexists i \text{ (dictator) such that} \quad x \succeq_{u_i} x' \iff x \succeq_{S_u} x' \tag{27}$$

where $u_i = u(\cdot, i)$ is the utility of the agent $i$.

**Case 1: $|\mathbf{V}| = 1$**

Given $|V| = 1$, there exists a single agent with truthful utility $u$, since it cannot be socially influenced

$$\mathcal{A}[v(x,:)] = \mathcal{A}[u(x,:)]$$
$$= u(x,i) \quad \text{(Def 2.3)}$$

Thus all aggregation rules are dictatorships in this case where the single agent acts as a dictator.

**Case 2: $|\mathbf{V}| > 1$ and $|\mathcal{X}| = 2$**

Given $|\mathcal{X}| = 2$. Lets consider $\mathcal{X} = \{x, x'\}$ then for all agents $\{u(\cdot, i)\}_{i \in V}, (x \succeq_{u_i} x') \vee (x' \succeq_{u_i} x)$ is true. Now when we consider the social utility $S_u$ since it is a complete preference $(x \succeq_{S_u} x') \vee (x' \succeq_{S_u} x)$ is also true which implies there exists an agent $i$ such that $x \succeq_{u_i} x' \iff x \succeq_{S_u} x'$. Hence all aggregation rules act as dictatorships in this case.

**Case 3: $|\mathbf{V}| > 1$ and $\mathcal{X} > 2$**

To show the counter example, we consider $u(x,:)$ such that for $i \neq j$, $\arg\max_{x \in \mathcal{X}} u(x,i) \neq \arg\max_{x \in \mathcal{X}} u(x,j)$. We can construct a social influence graph $G$ with transition matrix $A$ such that $A_{jk} = \delta_{i=k}$ where $i$ is the index of agent which influences everyone to take their utility.

Then the final aggregated non-truthful utility

$$\mathcal{A}[v(x,:)] = \mathcal{A}[Au(x,:)]$$
$$= \mathcal{A}[\mathbf{1}u(x,i)]$$
$$= u(x,i) \quad \text{(Def 2.3)}$$

Thus, $S_u$ cannot be the same as the aggregation of non-truthful utilities as it will lead to dictatorship. And given that for all $i, j$, $\arg\max_x u(x,i) \neq \arg\max_x u(x,j)$, there are two possible cases

**Case 3.1:** $\arg\max_x S_u(x) \neq \arg\max_x u(x,i)$ for all $i$. In such a case, groupthink-proofness is impossible by construction for graphs $G$ where an agent influences everyone to take their utility.

**Case 3.2:** $\exists i$ such that $\arg\max_x \mathcal{S}_u(x) = \arg\max_x u(x, i)$. We consider a graph $G$ such that agent $j \neq i$ influences everyone to take their utility.

Thus we show for every aggregation function $\mathcal{A}$ there exists a graph $G$ and a utility profile $u(x. :)$ such that groupthink-proofness is not satisfied. $\qquad\square$

### C.3 EXAMPLE: PARETO FRONTS ARE NOT GROUPTHINK-PROOF

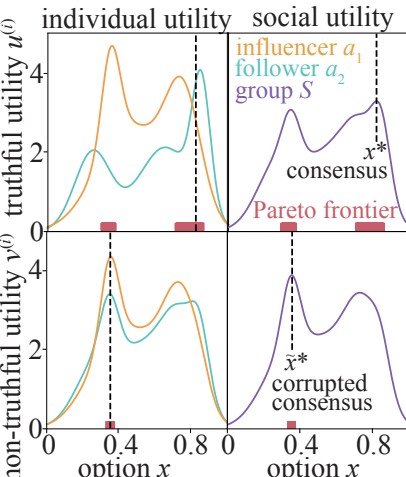

Figure 5: Pareto frontier corrupted by the social influence can exclude the true consensus.

A common approach to overcome this challenge in multi-objective optimization is to estimate the Pareto frontier, assuming it includes the consensus point $x^\star$, as a 'trade-off' solution. We present a simple illustration where this assumption fails due to social influence. Consider two agents: the influencer ($a_1$) and the follower ($a_2$)[3]. Here, we assume that $v$ is given, and $\mathcal{A}$ is utilitarian (c.f. Section 2.1), yet we do not have the access to $u$ nor $A$. We set the ground truth as $A = \left(\begin{smallmatrix} 0.9 & 0.1 \\ 0.6 & 0.4 \end{smallmatrix}\right)$. This indicates that $a_1$ prioritizes their own utility 9 times more than $a_2$'s, while $a_2$ values $a_1's$ utility 1.5 times more than their own. As shown in Figure 5, the non-truthful utilities $v$ become nearly identical. While the truthful consensus is at $x^* = 0.82$, the non-truthful one is at $\tilde{x}^* = 0.38$. Furthermore, the non-truthful Pareto frontier does not contain the truthful consensus $x^*$. This example demonstrates that even the Pareto frontier, a conservative consensus approach, fails to address the issue of non-truthful feedback.

### C.4 PROOF OF PROPOSITION 2.4

*Proof.* This proof is adapted from Proposition 1 in Sim et al. (2021)) for our cases.

For any $x, x' \in \mathcal{X}$, let $u^*(x, :) := \phi(u(x, :))$ be the utility vectors obtained after sorting elements of $u(x, :)$ in ascending order, and $w_1 > w_2 > \cdots > w_n > 0$ be the weight function.

**Proof of monotonicity.** Let the position of $u(x, i)$ in $u^*(x, :)$ be $i_x$, i.e., $u^*(x, i_x) = u(x, i)$. Given $\forall k \in V \setminus \{i\}, u(x, k) = u(x', k)$ and $u(x, i) > u(x', i)$, we must have $i_x \geq i_{x'}$. Furthermore, (i) for $k \in [0, i_{x'})$ and $k \in (i_x, n], u^*(x, k) = u^*(x', k)$ and (ii) if $i_x > i_{x'}$, then for $k \in [i_{x'}, i_x)$,

---

[3]Note that the decision-maker is not the influencer but the facilitator. The facilitator seeks to elicit the truthful social utility $\mathcal{S}$, but the influencer distorts the entire voting process.

$u^*(x', k+1) = u^*(x, k).$

$\mathcal{S}(x) - \mathcal{S}(x')$

$$= \sum_{k=1}^{n} w_k u^*(x, k) - \sum_{k=1}^{n} w_k u^*(x', k), \qquad \text{(Definition of GSF in Eq. 2)}$$

$$= w_{i_x} u^*(x, i_x) + \sum_{k=i_x}^{i_x-1} w_{i_x} u^*(x, k) - \sum_{k=i_{x'}+1}^{i_x} w_{i_x} u^*(x', k) - w_{i_{x'}} u^*(x', i_{x'}), \quad \text{(Assumption (i))}$$

$$= w_{i_x} u^*(x, i_x) - w_{i_{x'}} u^*(x', i_{x'}) + \sum_{k=i_{x'}+1}^{i_x} (w_{k-1} - w_k) u^*(x', k), \qquad \text{(Assumption (ii))}$$

$$\geq w_{i_x} u^*(x, i_x) - w_{i_{x'}} u^*(x', i_{x'}) + u^*(x', i_{x'}) \sum_{k=i_{x'}+1}^{i_x} (w_{k-1} - w_k),$$

$$\text{(Using sorting properties: } \forall k > i_{x'}, \ u^*(x', k) \geq u^*(x', i_{x'}) \text{ and } w_{k-1} - w > 0)$$

$$= w_{i_x} u^*(x, i_x) - w_{i_{x'}} u^*(x', i_{x'}) + u^*(x', i_{x'})(w_{i_{x'}} - w_{i_x}), \qquad \text{(telescoping series)}$$

$$= w_{i_x}(u^*(x, i_x) - u^*(x', i_{x'})),$$

$$= w_{i_x}(u(x, i) - u(x', i)),$$

$$> 0. \qquad (u(x, i) > u(x', i) \text{ and } w_i > 0)$$

**Proof of PDP.** Let $l_x$ be the index of $\min(u^*(x, i), u^*(x, j))$ and $h_x$ be the index of $\max(u^*(x, i), u^*(x, j))$. We will see the following two useful facts:

(i) We must have $l_{x'} \leq l_x < h_x \leq h_{x'}$. We will see the validity of this condition by considering the following contradicting assumption; $l_{x'} > l_x$. Because of the strong PDP condition (a) and the fact that $l_{x'}$ index a minimum, it would mean $\min(u(x', i), u(x', j)) > \min(u(x, i), u(x, j))$. By the strong PDP condition (b), we would also have $\max(u(x', i), u(x', j)) < \max(u(x, i), u(x, j))$. As such, we would have $|u(x', i) - u(x', j)| < |u(x, i) - u(x, j)|$, which contradicts the strong PDP condition (c).

(ii) GSF can be decomposed as:

$$\mathcal{S}(x') = \sum_{k=1}^{l_{x'}-1} w_k u^*(x', k) + \mathcal{S}_{l_{x'}-1:h_{x'}}(x') + \sum_{k=h_{x'}+1}^{n} w_k u^*(x', k),$$

$$\mathcal{S}(x) = \sum_{k=1}^{l_{x'}-1} w_k u^*(x, k) + \mathcal{S}_{l_{x'}-1:h_{x'}}(x) + \sum_{k=h_{x'}+1}^{n} w_k u^*(x, k),$$

where

$$\mathcal{S}_{l_{x'}-1:h_{x'}}(x') = \ w_{l_{x'}} u^*(x', l_{x'}) \qquad\qquad + \sum_{k=l_{x'}+1}^{h_{x'}-1} w_k u^*(x', k) \qquad\qquad + w_{h_{x'}} u^*(x', h_{x'}),$$

$$\mathcal{S}_{l_{x'}-1:h_{x'}}(x) = \sum_{k=l_{x'}}^{l_x-1} w_k u^*(x, k) + w_{l_x} u^*(x, l_x) \ + \sum_{k=l_x+1}^{h_x-1} w_k u^*(x, k) + w_{h_x} u^*(x, h_x) \ + \sum_{k=h_x+1}^{h_{x'}} w_k u^*(x, k)$$

(iii) Here, by combining the strong PDP condition (a) and the condition (i), we have $u^*(x, k) = u^*(x', k)$ for $k \in [1, l_{x'}-1] \cup [h_{x'}+1, n] \cup [l_x+1, h_x+1]$. Thus, we have;

$$\sum_{k=1}^{l_{x'}-1} w_k u^*(x', k) = \sum_{k=1}^{l_{x'}-1} w_k u^*(x, k),$$

$$\sum_{k=h_{x'}+1}^{n} w_k u^*(x', k) = \sum_{k=h_{x'}+1}^{n} w_k u^*(x, k),$$

$$\sum_{k=l_x+1}^{h_x-1} w_k u^*(x', k) = \sum_{k=l_x+1}^{h_x-1} w_k u^*(x, k),$$

then,

$$\mathcal{S}(x') - \mathcal{S}(x) = \mathcal{S}_{l_{x'}-1:h_{x'}}(x') - \mathcal{S}_{l_{x'}-1:h_{x'}}(x).$$

Based on the PDP conditions (a)(b)(c) and the facts (i)(ii)(iii), we have,

$$
\begin{aligned}
&\mathcal{S}(x') - \mathcal{S}(x) \\
&= \sum_{k=l_{x'}}^{l_x-1} w_k u^*(x,k) + w_{l_x} u^*(x,l_x) + w_{h_x} u^*(x,h_x) + \sum_{h_x+1}^{h_{x'}} w_k u^*(x,k) \\
&\quad - \left( w_{l_{x'}} u^*(x',l_{x'}) + \sum_{l_{x'}+1}^{l_x} w_k u^*(x',k) + \sum_{k=h_x}^{h_{x'}-1} w_k u^*(x',k) + w_{h_{x'}} u^*(x',h_{x'}) \right),
\end{aligned}
$$

(Using (ii)(iii))

$$
\begin{aligned}
&= \sum_{k=l_{x'}}^{l_x-1} w_k u^*(x,k) + w_{l_x} u^*(x,l_x) + w_{h_x} u^*(x,h_x) + \sum_{h_x+1}^{h_{x'}} w_k u^*(x,k) \\
&\quad - \left( w_{l_{x'}} u^*(x',l_{x'}) + \sum_{l_{x'}+1}^{l_x} w_k u^*(x,k-1) + \sum_{k=h_x}^{h_{x'}-1} w_k u^*(x,k+1) + w_{h_{x'}} u^*(x',h_{x'}) \right),
\end{aligned}
$$

Here we used the ranking structure: decrement for $k \in (l_{x'}, l_x], u^*(x',k) = u^*(x,k-1)$ and increment for $k \in (h_x, h_{x'}], u^*(x',k) = u^*(x,k+1)$. Then, by regrouping the related terms, we have,

$$
\begin{aligned}
&= \sum_{k=l_{x'}}^{l_x-1} (w_k - w_{k+1}) u^*(x,k) + (w_{l_x} u^*(x,l_x) - w_{l_{x'}} u^*(x',l_{x'})) \\
&\quad + (w_{h_x} u^*(x,h_x) - w_{h_{x'}} u^*(x',h_{x'})) + \sum_{h_x+1}^{h_{x'}} (w_k - w_{k-1}) u^*(x,k), \\
&\geq \sum_{k=l_{x'}}^{l_x-1} (w_k - w_{k+1}) u^*(x',l_{x'}) + (w_{l_x} u^*(x,l_x) - w_{l_{x'}} u^*(x',l_{x'})) \\
&\quad + (w_{h_x} u^*(x,h_x) - w_{h_{x'}} u^*(x',h_{x'})) + \sum_{h_x+1}^{h_{x'}} (w_k - w_{k-1}) u^*(x',h_{x'}),
\end{aligned}
$$

because $w_k - w_{k-1} > 0$ and $u^*(x,k) = u^*(x',k+1) \geq u^*(x',l_{x'})$ for $k = l_{x'}, \cdots, l_x - 1$, and $(w_k - w_{k-1})$ is negative, and $u^*(x,k) = u^*(x',k-1) \leq u^*(x,h_{x'})$ for $k = h_x + 1, \cdots, h_{x'}$. Then, using the telescoping series,

$$
\begin{aligned}
&= (w_{l_{x'}} - w_{l_x}) u^*(x',l_{x'}) + (w_{l_x} u^*(x,l_x) - w_{l_{x'}} u^*(x',l_{x'})) \\
&\quad + (w_{h_x} u^*(x,h_x) - w_{h_{x'}} u^*(x',h_{x'})) + (w_{h_{x'}} - w_{h_x}) u^*(x',h_{x'}), \\
&= -w_{l_x} u^*(x',l_{x'}) + w_{l_x} u^*(x,l_x) + w_{h_x} u^*(x,h_x) - w_{h_x} u^*(x',h_{x'}), \\
&= -w_{l_x} u^*(x',l_{x'}) + w_{l_x} \left( u^*(x',l_{x'}) + u^*(x',h_{x'}) - u^*(x,h_x) \right) + w_{h_x} u^*(x,h_x) - w_{h_x} u^*(x',h_{x'}),
\end{aligned}
$$

(PDP condition (b))

$$
\begin{aligned}
&= (w_{l_x} + w_{h_x}) \left( u^*(x',h_{x'}) - u^*(x,h_x) \right), \\
&> 0.
\end{aligned}
$$

$\quad\quad$ ($l_x < h_x$ and $u^*(x',h_{x'}) > u^*(x,h_x)$), i.e., $\max(u(x',i),u(x',j)) > \max(u(x,i),u(x,j))$.)

Therefore, $\mathcal{S}(x') - \mathcal{S}(x) > 0$. $\qquad\qquad\qquad\qquad\qquad\qquad\qquad\qquad\qquad\square$

# D PREFERENCE LIKELIHOOD

## D.1 PROOF OF COROLLARY 3.5

*Proof.* We begin by the original Bradley-Terry model definition:

$$\mathbb{P}(\mathbf{1}_{x \succ x'}^{(i)} = 1) = \frac{\exp(u(x, i))}{\exp(u(x, i)) + \exp(u(x', i))}. \tag{28}$$

Using $\mathbb{P}(\mathbf{1}_{x \succ x'}^{(i)} = 0) = 1 - \mathbb{P}(\mathbf{1}_{x \succ x'}^{(i)} = 1)$, we introduce the likelihood function for a comparison oracle:

$$p_{\hat{u}^{(i)}}(x_\tau, x'_\tau, \mathbf{1}_\tau^{(i)}) := \mathbf{1}_\tau^{(i)} S\left(\hat{u}(x_\tau, :) - \hat{u}(x'_\tau, :)\right) + \left(1 - \mathbf{1}_\tau^{(i)}\right)\left[1 - S\left(\hat{u}(x_\tau, :) - \hat{u}(x'_\tau, :)\right)\right]. \tag{29}$$

We can then derive the likelihood function of a fixed function $\hat{u}$ over the observed dataset, $D_{\mathcal{Q}_t^u}^{(i)}$,

$$\mathbb{P}_{\hat{u}^{(i)}}(D_{\mathcal{Q}_t^u}^{(i)}) := \prod_{\tau \in \mathcal{Q}_t^u} p_{\hat{u}^{(i)}}(x_\tau, x'_\tau, \mathbf{1}_\tau^{(i)}).$$

Consequently, the log-likelihood (LL) function becomes:

$$\ell_t(\hat{u}(\cdot, i)) = \log \mathbb{P}_{\hat{u}^{(i)}}(D_{\mathcal{Q}_t^u}^{(i)}), \tag{30}$$

$$= \log \prod_{\tau \in \mathcal{Q}_t^u} p_{\hat{u}^{(i)}}(x_\tau, x'_\tau, \mathbf{1}_\tau^{(i)}), \tag{31}$$

$$= \sum_{\tau \in \mathcal{Q}_t^u} \log p_{\hat{u}^{(i)}}(x_\tau, x'_\tau, \mathbf{1}_\tau^{(i)}), \tag{32}$$

$$= \sum_{\tau \in \mathcal{Q}_t^u} \log \left[\frac{\exp(\hat{u}(x_\tau, i))\mathbf{1}_\tau^{(i)} + \exp(\hat{u}(x'_\tau, i))(1 - \mathbf{1}_\tau^{(i)})}{\exp(\hat{u}(x_\tau, i)) + \exp(\hat{u}(x'_\tau, i))}\right], \tag{33}$$

$$= \sum_{\tau \in \mathcal{Q}_t^u} \left[\hat{u}(x_\tau, i)\mathbf{1}_\tau^{(i)} + \hat{u}(x'_\tau, i)(1 - \mathbf{1}_\tau^{(i)})\right] - \sum_{\tau \in \mathcal{Q}_t^u} \log\left[\exp(\hat{u}(x_\tau, i)) + \exp(\hat{u}(x'_\tau, i))\right].$$

$$\tag{34}$$

$\square$

## D.2 JOINT LIKELIHOOD.

**Multi-agent case** We can easily extend to all agent cases:

$$\ell_t(\hat{u} \mid \mathcal{Q}_t^u) \tag{35}$$

$$= \sum_{i \in V} \ell_t(\hat{u}(\cdot, i)), \tag{36}$$

$$= \sum_{i \in V} \sum_{\tau \in \mathcal{Q}_t^u} \left[\hat{u}(x_\tau, i)\mathbf{1}_\tau^{(i)} + \hat{u}(x'_\tau, i)(1 - \mathbf{1}_\tau^{(i)})\right] - \sum_{i \in V} \sum_{\tau \in \mathcal{Q}_t^u} \log\left[\exp(\hat{u}(x_\tau, i)) + \exp(\hat{u}(x'_\tau, i))\right],$$

$$\tag{37}$$

$$= \sum_{\tau \in \mathcal{Q}_t^u} \left[\hat{u}(x_\tau, :)\mathbf{1}_\tau^u + \hat{u}(x'_\tau, :)(1 - \mathbf{1}_\tau^u)\right] - \sum_{i \in V} \sum_{\tau \in \mathcal{Q}_t^u} \log\left[\exp(\hat{u}(x_\tau, i)) + \exp(\hat{u}(x'_\tau, i))\right]. \tag{38}$$

**Non-truthful case** Similar to Eq. (29), we can define the likelihood function for $v$,

$$\ell_t(\hat{v} \mid \mathcal{Q}_t^v) \tag{39}$$

$$= \sum_{i \in V} \ell_t(\hat{v}(\cdot, i)), \tag{40}$$

$$= \sum_{\tau \in [t]} \left[\hat{v}(x_\tau, :)\mathbf{1}_\tau^v + \hat{v}(x'_\tau, :)(1 - \mathbf{1}_\tau^v)\right] - \sum_{i \in V} \sum_{\tau \in [t]} \log\left[\exp(\hat{v}(x_\tau, i)) + \exp(\hat{v}(x'_\tau, i))\right]. \tag{41}$$

**Joint log likelihood**    The joint likelihood is simply the sum of each likelihood:

$$
\begin{aligned}
&\mathcal{L}_t(\hat{u}, \hat{v}) \\
=&\ell_t(\hat{u} \mid \mathcal{Q}_t^u) + \ell_t(\hat{u}, \hat{v} \mid \mathcal{Q}_t^v), \\
=&\sum_{\tau \in \mathcal{Q}_t^u} [\hat{u}(x_\tau, :)\mathbf{1}_\tau^u + \hat{u}(x_\tau', :)(1 - \mathbf{1}_\tau^u)] - \sum_{i \in V} \sum_{\tau \in \mathcal{Q}_t^u} \log\left[\exp(\hat{u}(x_\tau, i)) + \exp(\hat{u}(x_\tau', i))\right] \\
&+ \sum_{\tau \in [t]} [\hat{v}(x_\tau, :)\mathbf{1}_\tau^v + \hat{v}(x_\tau', :)(1 - \mathbf{1}_\tau^v)] - \sum_{i \in V} \sum_{\tau \in [t]} \log\left[\exp(\hat{v}(x_\tau, i)) + \exp(\hat{v}(x_\tau', i))\right]. \quad (42)
\end{aligned}
$$

**MAP estimation**    We can further extend the above joint log likelihood to MAP estimation by adding $\log p(A)$:

$$
\begin{aligned}
&\mathcal{L}_t^{\mathrm{MAP}}(\hat{u}, \hat{A}, \hat{v}) \\
=&\mathcal{L}_t(\hat{u}, \hat{v}) + \log p(A), \\
\propto&\mathcal{L}_t(\hat{u}, \hat{v}) + \sum_{i \in V} \log \mathrm{Dirichlet}(A_i, \alpha_i) - \sum_{i,j \in [n]} A_{ij}^2, \quad\quad\quad (43) \\
\propto&\mathcal{L}_t(\hat{u}, \hat{v}) + \sum_{j \in V} (\alpha_{ij} - 1)\log A_{ij} - \sum_{i,j \in [n]} A_{ij}^2, \quad\quad (\text{Remove constant term}) \\
=&\sum_{\tau \in \mathcal{Q}_t^u} [\hat{u}(x_\tau, :)\mathbf{1}_\tau^u + \hat{u}(x_\tau', :)(1 - \mathbf{1}_\tau^u)] - \sum_{i \in V} \sum_{\tau \in \mathcal{Q}_t^u} \log\left[\exp(\hat{u}(x_\tau, i)) + \exp(\hat{u}(x_\tau', i))\right] \\
&+ \sum_{\tau \in [t]} [\hat{v}(x_\tau, :)\mathbf{1}_\tau^v + \hat{v}(x_\tau', :)(1 - \mathbf{1}_\tau^v)] - \sum_{i \in V} \sum_{\tau \in [t]} \log\left[\exp(\hat{v}(x_\tau, i)) + \exp(\hat{v}(x_\tau', i))\right] \\
&+ \sum_{j \in V} (\alpha_{ij} - 1)\log A_{ij} - \sum_{i,j \in [n]} A_{ij}^2. \quad\quad\quad (44)
\end{aligned}
$$

D.3    PROOF OF LEMMA 3.6

To prepare for the proof of the lemma, we first introduce the following two preliminary lemmas adapted from the Lemma C.2 and C.3 in Xu et al. (2024b).

**Lemma D.1.** *For any fixed $\hat{v}^{(i)} := \hat{v}(\cdot, i)$ that is independent of $(x_\tau, x_\tau', \mathbf{1}_\tau^{(i)})_{\tau \in |\mathcal{Q}_t^v|}$, we have, with probability at least $1 - \delta$, $\forall t \geq 1, \forall i \in V$,*

$$
\log \mathbb{P}_{\hat{v}^{(i)}}\left((x_\tau, x_\tau', \mathbf{1}_\tau^{(i)})_{\tau \in |\mathcal{Q}_t^v|}\right) - \log \mathbb{P}_{v^{(i)}}\left((x_\tau, x_\tau', \mathbf{1}_\tau^{(i)})_{\tau \in |\mathcal{Q}_t^v|}\right) \leq \sqrt{32|\mathcal{Q}_t^v|L_v^2 \log \frac{\pi^2 |\mathcal{Q}_t^v|^2}{6\delta}}, \tag{45}
$$

*where $v^{(i)}$ is the ground truth function.*

**Lemma D.2.** *There exists an independent constant $C_L^v > 0$, such that, $\forall \epsilon > 0, \forall v_1^{(i)}, v_2^{(i)} \in \mathcal{B}^u$ that satisfies $\|v_1^{(i)} - v_2^{(i)}\|_\infty \leq \epsilon$, we have,*

$$
\log \mathbb{P}_{v_1^{(i)}}\left((x_\tau, x_\tau', \mathbf{1}_\tau^{(i)})_{\tau \in |\mathcal{Q}_t^v|}\right) - \log \mathbb{P}_{v_2^{(i)}}\left((x_\tau, x_\tau', \mathbf{1}_\tau^{(i)})_{\tau \in |\mathcal{Q}_t^v|}\right) \leq C_L^v \epsilon |\mathcal{Q}_t^v|, \tag{46}
$$

*where $C_L^v := 1 + \frac{2}{1 + e^{-2L_v}}$.*

**Main proof.**    We use $\mathcal{N}(\mathcal{B}^v, \epsilon, \|\cdot\|_\infty)$ to denote the covering number of the set $\mathcal{B}^v$, with $(v_j^{(i),\epsilon})_{j=1}^{\mathcal{N}(\mathcal{B}^v, \epsilon, \|\cdot\|_\infty)}$ be a set of $\epsilon$-covering for the set $\mathcal{B}^v$. Reset the '$\delta$' in Lemma D.1 as $\delta / \mathcal{N}(\mathcal{B}^v, \epsilon, \|\cdot\|_\infty)$ and applying the probability union bound, we have, with probability at least $1 - \delta, \forall v_j^{(i),\epsilon}, t \geq 1, i \in V$,

$$
\log \mathbb{P}_{v_j^{(i),\epsilon}}\left((x_\tau, x_\tau', \mathbf{1}_\tau^{(i)})_{\tau \in |\mathcal{Q}_t^v|}\right) - \log \mathbb{P}_{v^{(i)}}\left((x_\tau, x_\tau', \mathbf{1}_\tau^{(i)})_{\tau \in |\mathcal{Q}_t^v|}\right) \leq \sqrt{32|\mathcal{Q}_t^v|L_v^2 \log \frac{\pi^2 |\mathcal{Q}_t^v|^2}{6\delta}}. \tag{47}
$$

By the definition of $\epsilon$-covering, there exists $k \in [\mathcal{N}(\mathcal{B}^v, \epsilon, \|\cdot\|_\infty)]$, such that,

$$\|\hat{v} - v_k^\epsilon\|_\infty \leq \epsilon, \tag{48}$$

Hence, with probability at least $1 - \delta$,

$$
\begin{aligned}
&\log \mathbb{P}_{\hat{v}^{(i)}}\left((x_\tau, x'_\tau, \mathbf{1}_\tau^{(i)})_{\tau \in |\mathcal{Q}_t^v|}\right) - \log \mathbb{P}_{v^{(i)}}\left((x_\tau, x'_\tau, \mathbf{1}_\tau^{(i)})_{\tau \in |\mathcal{Q}_t^v|}\right) \\
&= \log \mathbb{P}_{\hat{v}^{(i)}}\left((x_\tau, x'_\tau, \mathbf{1}_\tau^{(i)})_{\tau \in |\mathcal{Q}_t^v|}\right) - \log \mathbb{P}_{v_j^{(i),\epsilon}}\left((x_\tau, x'_\tau, \mathbf{1}_\tau^{(i)})_{\tau \in |\mathcal{Q}_t^v|}\right) \\
&\quad + \log \mathbb{P}_{v_j^{(i),\epsilon}}\left((x_\tau, x'_\tau, \mathbf{1}_\tau^{(i)})_{\tau \in |\mathcal{Q}_t^v|}\right) - \log \mathbb{P}_{v^{(i)}}\left((x_\tau, x'_\tau, \mathbf{1}_\tau^{(i)})_{\tau \in |\mathcal{Q}_t^v|}\right), \\
&\leq C_L^v \epsilon |\mathcal{Q}_t^v| + \sqrt{32|\mathcal{Q}_t^v|L_v^2 \log \frac{\pi^2 |\mathcal{Q}_t^v|^2 \mathcal{N}(\mathcal{B}^v, \epsilon, \|\cdot\|_\infty)}{6\delta}}.
\end{aligned}
\tag{49}
$$

Under the isotropic norm bound assumption 2.6, this easily extends to $n$ utilities,

$$
\begin{aligned}
&\log \mathbb{P}_{\hat{v}}\left((x_\tau, x'_\tau, \mathbf{1}_\tau^v)_{\tau \in |\mathcal{Q}_t^v|}\right) - \log \mathbb{P}_v\left((x_\tau, x'_\tau, \mathbf{1}_\tau^v)_{\tau \in |\mathcal{Q}_t^v|}\right) \\
&\leq nC_L^v \epsilon |\mathcal{Q}_t^v| + \sqrt{32|\mathcal{Q}_t^v|n^2 L_v^2 \log \frac{\pi^2 |\mathcal{Q}_t^v|^2 \mathcal{N}(\mathcal{B}^v, \epsilon, \|\cdot\|_\infty)}{6\delta}}.
\end{aligned}
\tag{50}
$$

Similarly, the same applies to $u$ under Assumption 2.6,

$$
\begin{aligned}
&\log \mathbb{P}_{\hat{u}}\left((x_\tau, x'_\tau, \mathbf{1}_\tau^u)_{\tau \in |\mathcal{Q}_t^u|}\right) - \log \mathbb{P}_u\left((x_\tau, x'_\tau, \mathbf{1}_\tau^u)_{\tau \in |\mathcal{Q}_t^u|}\right) \\
&\leq nC_L^u \epsilon |\mathcal{Q}_t^u| + \sqrt{32|\mathcal{Q}_t^u|n^2 L_u^2 \log \frac{\pi^2 |\mathcal{Q}_t^u|^2 \mathcal{N}(\mathcal{B}^v, \epsilon, \|\cdot\|_\infty)}{6\delta}}.
\end{aligned}
\tag{51}
$$

Therefore, the joint likelihood $\mathcal{L}_t(\tilde{u}, \tilde{A}, \tilde{v})$ is bounded by:

$$
\begin{aligned}
&\mathcal{L}_t(\tilde{u}, \tilde{A}, \tilde{v}) \\
&= \log \mathbb{P}_{\hat{v}}\left((x_\tau, x'_\tau, \mathbf{1}_\tau^v)_{\tau \in |\mathcal{Q}_t^v|}\right) - \log \mathbb{P}_v\left((x_\tau, x'_\tau, \mathbf{1}_\tau^v)_{\tau \in |\mathcal{Q}_t^v|}\right) \\
&\quad + \log \mathbb{P}_{\hat{u}}\left((x_\tau, x'_\tau, \mathbf{1}_\tau^u)_{\tau \in |\mathcal{Q}_t^u|}\right) - \log \mathbb{P}_u\left((x_\tau, x'_\tau, \mathbf{1}_\tau^u)_{\tau \in |\mathcal{Q}_t^u|}\right), \\
&\leq nC_L^v \epsilon |\mathcal{Q}_t^v| + \sqrt{32|\mathcal{Q}_t^v|n^2 L_v^2 \log \frac{\pi^2 |\mathcal{Q}_t^v|^2 \mathcal{N}(\mathcal{B}^v, \epsilon, \|\cdot\|_\infty)}{6\delta}} \\
&\quad + nC_L^u \epsilon |\mathcal{Q}_t^u| + \sqrt{32|\mathcal{Q}_t^u|n^2 L_u^2 \log \frac{\pi^2 |\mathcal{Q}_t^u|^2 \mathcal{N}(\mathcal{B}^v, \epsilon, \|\cdot\|_\infty)}{6\delta}}.
\end{aligned}
\tag{52}
$$

By range preservation lemma B.2, $L_v = L_u$, thereby $C_L^v = C_L^u$. For brevity, we introduce the notations $C_\epsilon := \frac{\pi^2 \mathcal{N}(\mathcal{B}^v, \epsilon, \|\cdot\|_\infty)}{6\delta}$, $|\mathcal{Q}_t^{uv}| := |\mathcal{Q}_t^v| + |\mathcal{Q}_t^u|$.

Therefore,

$$
\begin{aligned}
&\mathcal{L}_t(\tilde{u}, \tilde{A}, \tilde{v}) \\
&\leq n\epsilon C_L^v(|\mathcal{Q}_t^u| + |\mathcal{Q}_t^v|) + \sqrt{32n^2 L_v^2}\left(\sqrt{|\mathcal{Q}_t^u| \log C_\epsilon |\mathcal{Q}_t^u|^2} + \sqrt{|\mathcal{Q}_t^v| \log C_\epsilon |\mathcal{Q}_t^v|^2}\right), \\
&\hspace{9.5cm} \text{(Rearranging Eq. (52))} \\
&\leq n\epsilon C_L^v |\mathcal{Q}_t^{uv}| + \sqrt{32n^2 L_v^2}\left(\sqrt{|\mathcal{Q}_t^u| \log C_\epsilon |\mathcal{Q}_t^{uv}|^2} + \sqrt{|\mathcal{Q}_t^v| \log C_\epsilon |\mathcal{Q}_t^{uv}|^2}\right), \\
&\hspace{9.5cm} (|\mathcal{Q}_t^u|, |\mathcal{Q}_t^v| < |\mathcal{Q}_t^{uv}|) \\
&= n\epsilon C_L^v |\mathcal{Q}_t^{uv}| + \sqrt{32n^2 L_v^2}(\sqrt{|\mathcal{Q}_t^u|} + \sqrt{|\mathcal{Q}_t^v|})\sqrt{\log C_\epsilon |\mathcal{Q}_t^{uv}|^2}, \hspace{1.5cm} \text{(factor out)} \\
&\leq n\epsilon C_L^v |\mathcal{Q}_t^{uv}| + \sqrt{64n^2 L_v^2 |\mathcal{Q}_t^{uv}| \log C_\epsilon |\mathcal{Q}_t^{uv}|^2}, \hspace{3.4cm} \text{(Cauchy-Schwarz)} \\
&= n\epsilon C_L^v |\mathcal{Q}_t^{uv}| + \sqrt{64n^2 L_v^2 |\mathcal{Q}_t^{uv}| \log \frac{\pi^2 |\mathcal{Q}_t^{uv}|^2 \mathcal{N}(\mathcal{B}^v, \epsilon, \|\cdot\|_\infty)}{6\delta}}, \hspace{1.4cm} \text{(unpack } C_\epsilon) \\
&:= \beta_1(\epsilon, \delta, n, |\mathcal{Q}_t^{uv}|). \hspace{8.3cm} \text{(define } \beta)
\end{aligned}
$$

The last inequality was derived from Cauchy-Schwarz inequality, $\sqrt{a} + \sqrt{b} \leq \sqrt{2(a + b)}$.

# E  EFFICIENT COMPUTATIONS

## E.1  PROOF OF LEMMA E.1

**Lemma E.1** (**Kernelized formulation**). *MAP and (5) can be recasted into convex optimisation:*

*(Reformulated MLE)*

$$\max_{\substack{U_t\in\mathbb{R}^{nt} \\ V_t\in\mathbb{R}^{nt} \\ \tilde{A}\in\mathbb{R}^{n^2}}} \mathcal{L}_t^{\mathrm{MAP}}(U_t, \tilde{A}, V_t \mid D_t)$$

$$s.t. \quad U_t^\top K_{\mathcal{Q}_t^u}^{-1} U_t \le L_v^2, \tag{53a}$$
$$V_t^\top K_{\mathcal{Q}_t^v}^{-1} V_t \le L_v^2,$$
$$1 - \delta_A \ge A_{ij} \ge \delta_A, \ \forall i,j,$$
$$\sum_j A_{ij} = 1, \ \forall j$$

*(Reformulated acquisition function)*

$$\max_{\substack{U_t\in\mathbb{R}^{nt},\, \boldsymbol{z}\in\mathbb{R}^{n} \\ V_t\in\mathbb{R}^{nt},\tilde{A}\in\mathbb{R}^{n^2}}} \mathcal{A}[\boldsymbol{z}] - \mathcal{A}[\boldsymbol{z}_t]$$

$$s.t. \quad \begin{bmatrix} U_t \\ \boldsymbol{z} \end{bmatrix}^\top K_{\mathcal{Q}_t^u,x}^{-1} \begin{bmatrix} U_t \\ \boldsymbol{z} \end{bmatrix} \le L_v^2,$$
$$V_t^\top K_{\mathcal{Q}_t^v}^{-1} V_t \le L_v^2,$$
$$\mathcal{L}_t^{\mathrm{MAP}}(U_t, \tilde{A}, V_t \mid D_t) \ge \mathcal{L}_t^{\mathrm{MAP}}(\hat{u}_t, \hat{A}_t, \hat{v}_t) - \beta_t(|\mathcal{Q}_t^{uv}|),$$
$$\ell_t(U_t \mid D_{\mathcal{Q}_t^u}) \ge \ell_t(\hat{U}_t \mid D_{\mathcal{Q}_t^u}) - \beta_t^u(|\mathcal{Q}_t^u|),$$
$$\ell_t(V_t \mid D_{\mathcal{Q}_t^v}) \ge \ell_t(\hat{V}_t \mid D_{\mathcal{Q}_t^v}) - \beta_t^v(|\mathcal{Q}_t^v|),$$
$$1 - \delta_A \ge A_{ij} \ge \delta_A, \ \forall i,j,$$
$$\sum_j A_{ij} = 1, \ \forall j$$

$$\tag{53b}$$

*Proof.* We begin by the convexity, then we show the equivalence to MAP, and the acquisition function maximisation problem (5).

### E.1.1  PROOF OF CONVEXITY.

**Reformulated MLE**  The function $\psi_\tau(y, y') := \log(e^y + e^{y'}) - p_\tau y - (1 - p_\tau)y'$, $p_\tau \in \{0,1\}$ is a convex function because $\nabla\psi_\tau(y_\tau, y_\tau') = 0$, thus the Hessian is nonnegative. Then, when assume $z_\tau = \tilde{v}(x_\tau), \eta_\tau = \tilde{u}(x_\tau)$, our negative log likelihood function $-\mathcal{L}_t(z_\tau, z_\tau', \eta_\tau, \eta_\tau')$ is also a convex function with $\nabla\mathcal{L}_t(z_\tau, z_\tau', \eta_\tau, \eta_\tau') = 0$.

For graph convolution part, the function $\psi_\tau'(A) := \log(e^{Au} + e^{Au'}) - p_\tau Au - (1 - p_\tau)Au'$ is also convex with respect to A because its Hessian is nonnegative:

$$\frac{\partial^2}{\partial A^2}\psi_\tau'(A) = (u - u')^2 \frac{e^{Au}e^{Au'}}{(e^{Au} + e^{Au'})^2} \ge 0. \tag{54}$$

Similarly, this function is convex with respect to $u$ and $u'$. We introduce the function $P(u, u') = e^{Au}/e^{Au}+e^{Au'}$

$$\frac{\partial^2}{\partial u^2}\psi_\tau'(u) = A^2 P(u)(1 - P(u)), \tag{55}$$

$$\frac{\partial^2}{\partial u'^2}\psi_\tau'(u) = A^2 P(u')(1 - P(u')), \tag{56}$$

$$\frac{\partial^2}{\partial u \partial u'}\psi_\tau'(u, u') = -A^2 P(u, u')(1 - P(u, u')), \tag{57}$$

Therefore, Hessian matrix $H$ is

$$H = A^2 P(u, u')(1 - P(u, u')) \begin{bmatrix} 1 & -1 \\ -1 & 1 \end{bmatrix} \tag{58}$$

and the Hessian matrix is positive semi-definite because all eigenvalues are non-negative. Thus, this is also convex.

**Reformulated acquisition function**  The GSF is the convex combination of utilities independent of both $\tilde{A}$ and $\boldsymbol{z}$. Thus, the aggregate operation is simply reduced to the linear combination of $\boldsymbol{z}$. Under the convex constraint of optimistic MLE, the linear combination of convex functions with nonzero weights is also a convex function. And the weight function of GSF is nonzero by definition.

### E.1.2 PROOF OF MLE REFORMULATION.

The joint likelihood $\mathcal{L}_t$ in Eq. (42) only depends on the values $(\hat{u}(x_\tau, :), \hat{u}(x'_\tau, :), \hat{v}(x_\tau, :), \hat{v}(x'_\tau, :)) = (u_\tau, u'_\tau, v_\tau, v'_\tau)$, where $u_\tau = (u_\tau^{(i)})_{i \in [n]}$. As such, we only need to optimise over $(u_\tau, u'_\tau, v_\tau, v'_\tau)$ subject to that they are functions in $\mathcal{H}_{k_i}$ with norm less or equal to $L_v$. Furthermore, Algorithm 1 sets $x'_\tau = x_{\tau-1}$, thereby $u'_\tau = u_{\tau-1}, v'_\tau = v_{\tau-1}$. Hence, we can further reduce the optimisation variables to $(u_\tau, v_\tau)$.

Here, we only assume $\tilde{u}(\cdot, i) \in \mathcal{H}_{k_i}$, then the norm bound constraints are only subject to $\tilde{u}$. Note that our kernel is vector-valued, so we use the following notation to describe:

$$U_t^\top K_{\mathcal{Q}_t^u}^{-1} U_t := \left( (U_t^{(i)})^\top (K_{\mathcal{Q}_t^u}^{(i)})^{-1} U_t^{(i)} \right)_{i=1}^n \tag{59}$$

where $K_{\mathcal{Q}_t^u}^{(i)} := (k_v(x_{\tau_1}, x_{\tau_2})_{\tau_1, \tau_2 \in \mathcal{Q}_t^u}$. The same applies to $V_t$ and corresponding kernel. As such, the constraint in Prob. (53a) consists of $n$ kernel bound constraints. Each constraint is direct application of representor theorem (Schölkopf et al., 2001).

### E.1.3 PROOF OF ACQUISITION FUNCTION REFORMULATION.

Prob (5) can be formally written as

$$\max_{\tilde{u}} \quad \mathcal{A}[\tilde{u}(x, :)] - \mathcal{A}[\tilde{u}(x_t, :)],$$
$$\text{s.t.} \quad \tilde{u}, \tilde{A}, \tilde{v} \in \mathcal{B}^{u, A, v}, \tag{60}$$
$$\mathcal{L}_t(\tilde{u}, \tilde{A}, \tilde{v}) \geq \mathcal{L}_t(\hat{u}, \hat{A}, \hat{v}) - \beta_t^{u, A, v}.$$

This has an infinite-dimensional function variable, thereby being intractable. Similar to the MLE reformulation, we can recast to finite, tractable optimization problem.

**Simplest setting.** First, we will start with the simplest case where $|\mathcal{Q}_t^u| = T$, as such

$$\max_{\tilde{u}} \quad \mathcal{A}[\tilde{u}(x, :)] - \mathcal{A}[\tilde{u}(x_t, :)],$$
$$\text{s.t.} \quad \tilde{u} \in \mathcal{B}^u, \tag{61}$$
$$\mathcal{L}_t(\tilde{u}) \geq \mathcal{L}_t(\hat{u}) - \beta_t^u.$$

And we show the equivalence to the following kernelized formulation.

$$\max_{Z_{0:t} \in \mathbb{R}^{n(t+1)}, \ \boldsymbol{z} \in \mathbb{R}^n} \quad \mathcal{A}[\boldsymbol{z}] - \mathcal{A}[\boldsymbol{z}_t]$$
$$\text{s.t.} \quad \begin{bmatrix} Z_{0:t} \\ \boldsymbol{z} \end{bmatrix}^\top K_{\mathcal{Q}_t^u, x}^{-1} \begin{bmatrix} Z_{0:t} \\ \boldsymbol{z} \end{bmatrix} \leq L_u^2, \tag{62}$$
$$Z_{0:t}^\top K_{0:t}^{-1} Z_{0:t} \leq L_v^2,$$
$$\ell_t(Z_{0:t}) \geq \mathcal{L}_t(\hat{u}_t) - \beta_t^u.$$

Let $\tilde{u}$ be any feasible solution of the above innter optimisaton problem, and $\tilde{z} = \tilde{u}(x, :)$ and $\tilde{Z}_{0:t} = (\tilde{u}(x_\tau, :))_{\tau=0}^t$ be the corresponding utility value. Consider the minimum-norm interpolation problem,

$$\min_{s \in \mathcal{B}^v} \quad \|s\|^2$$
$$\text{s.t.} \quad s(x_\tau, :) = \tilde{z}_\tau, \forall \tau \in \{0\} \cup [t], \tag{63}$$
$$s(x) = \tilde{z}.$$

By representer theorem, this problem admits an optimal solution with the form $\alpha^\top k_{0:t, x}$, where $k_{0:t, x} := \{k(w, \cdot)\}_{w \in \{x_0, \cdots, x_t, x\}}$. Thus, Prob. (63) can be reduced to

$$\min_{\alpha \in \mathbb{R}^{t+2}} \quad \alpha^\top K_{0:t, x} \alpha$$
$$\text{s.t.} \quad K_{0:t, x} \alpha = \begin{bmatrix} \tilde{Z}_{0:t} \\ \tilde{z} \end{bmatrix}. \tag{64}$$

Then the optimal solution of this Prob. (64) is

$$\alpha^\top K_{0:t,x}\alpha = (K_{0:t,x}\alpha)^\top K_{0:t,x}^{-1}K_{0:t,x}\alpha,$$

$$= \left[\begin{array}{c}\tilde{Z}_{0:t}\\\tilde{z}\end{array}\right]^T K_{0:t,x}^{-1}\left[\begin{array}{c}\tilde{Z}_{0:t}\\\tilde{z}\end{array}\right]. \tag{65}$$

Since $\tilde{u}$ is an interpolant by construction of $(\tilde{Z}_{0:t}, \tilde{z})$. We have

$$\left[\begin{array}{c}\tilde{Z}_{0:t}\\\tilde{z}\end{array}\right]^T K_{0:t,x}^{-1}\left[\begin{array}{c}\tilde{Z}_{0:t}\\\tilde{z}\end{array}\right] \leq \|u\|^2 \leq L_u^2, \tag{66}$$

yielding the first constraint. As the LL function only depends on $(\tilde{Z}_{0:t})$, it holds that

$$\mathcal{L}(\tilde{Z}_{0:t}, \mid D_{\mathcal{Q}_t^u}) = \mathcal{L}_t(\tilde{v}) \geq \mathcal{L}_t(\hat{u}_t) - \beta_t^u, \tag{67}$$

and the objective satisfy

$$\mathcal{A}[\boldsymbol{z}] - \mathcal{A}[\boldsymbol{z}_t] = \mathcal{A}[\tilde{u}(x,:)] - \mathcal{A}[\tilde{u}(x_t,:)]. \tag{68}$$

Therefore, a set $(\tilde{Z}_{0:t}, \tilde{z})$ is a feasible solution for Prob. (62), with the same objective as $\tilde{v}$ for the the infinite dimensional Prob. (61).

Next, we show that for any feasible solution for Prob. (62), we can find a corresponding feasible solution of Prob. (61) with the same objective value. Let $(Z_{0:t}, z)$ be a feasible solution of Prob. (62). We construct

$$\tilde{u}_z = \left[\begin{array}{c}\tilde{Z}_{0:t}\\\tilde{z}\end{array}\right]^T K_{0:t,x}^{-1}k_{0:t,x}(\cdot), \tag{69}$$

Hence,

$$\|\tilde{u}_z\|^2 = \left[\begin{array}{c}\tilde{Z}_{0:t}\\\tilde{z}\end{array}\right]^T K_{0:t,x}^{-1}\left[\begin{array}{c}\tilde{Z}_{0:t}\\\tilde{z}\end{array}\right] \leq L_u^2, \tag{70}$$

and it can be checked that $\tilde{u}_z(x_\tau) = \boldsymbol{z}_\tau, \forall \tau \in \{0\} \cup [t]$ and $\tilde{u}_z(x) = \boldsymbol{z}$. So $\mathcal{L}_t(\tilde{u}_z) = \mathcal{L}(\tilde{Z}_{0:t} \mid D_t) \geq \mathcal{L}(\hat{v}) - \beta_t^u$. And the objectives satisfy $\mathcal{A}[\tilde{u}_z(x)] - \mathcal{A}[\tilde{u}_z(x_t)] = \mathcal{A}[\boldsymbol{z}] - \mathcal{A}[\boldsymbol{z}_t]$. So it is proved that for any feasible solution of Prob. (62), we can find a corresponding feasible solution of Prob. (61) with the same objective value.

**Original setting** The $V_t$ constraint is the same with the previous MLE reformulation. Also, the optimistic MLE bound needs to modify to $\mathcal{L}_t(\hat{u}, \hat{A}, \hat{v})$ as it involves $A$ and $v$ estimate. Then, we can show the equivalence of this Prob. (60) with the constraint in Prob. (53b). □

### E.2 PREDICTIVE CONFIDENCE BOUND

Using the same idea, we can obtain the predictive confidence bounds. Here, $x$ is given as the prediction point, then the upper confidence bound $\overline{u}(x, i)$ and lower confidence bound $\underline{u}(x, i)$ become

$$\overline{u}(x, i) := \max_{\substack{U_t \in \mathbb{R}^{t+1}\ V_t \in \mathbb{R}^t, \boldsymbol{z} \in \mathbb{R}\\ \tilde{A} \in \mathbb{R}^{n^2},}} \boldsymbol{z}$$

$$\text{s.t.} \quad \left[\begin{array}{c}U_t\\\boldsymbol{z}\end{array}\right]^\top K_{\mathcal{Q}_t^u,x}^{-1}\left[\begin{array}{c}U_t\\\boldsymbol{z}\end{array}\right] \leq L_v^2,$$

$$V_t^\top K_{\mathcal{Q}_t^v}^{-1}V_t \leq L_v^2,$$

$$\mathcal{L}_t^{\mathrm{MAP}}(U_t, \tilde{A}, V_t \mid D_t) \geq \mathcal{L}_t^{\mathrm{MAP}}(\hat{u}_t, \hat{A}_t, \hat{v}_t) - \beta_t(|\mathcal{Q}_t^{uv}|), \tag{71}$$

$$\ell_t(U_t \mid D_{\mathcal{Q}_t^u}) \geq \ell_t(\hat{U}_t \mid D_{\mathcal{Q}_t^u}) - \beta_t^u(|\mathcal{Q}_t^u|),$$

$$\ell_t(V_t \mid D_{\mathcal{Q}_t^v}) \geq \ell_t(\hat{V}_t \mid D_{\mathcal{Q}_t^v}) - \beta_t^v(|\mathcal{Q}_t^v|),$$

$$1 - \delta_A \geq A_{ij} \geq \delta_A, \ \forall i, j,$$

$$\sum_j A_{ij} = 1, \ \forall j$$

$$\underline{u}(x, i) := \min_{\substack{U_t \in \mathbb{R}^{t+1}\ V_t \in \mathbb{R}^t, \boldsymbol{z} \in \mathbb{R} \\ \tilde{A} \in \mathbb{R}^{n^2},}} \boldsymbol{z}$$

$$\text{s.t.} \quad \begin{bmatrix} U_t \\ \boldsymbol{z} \end{bmatrix}^\top K_{\mathcal{Q}_t^u, x}^{-1} \begin{bmatrix} U_t \\ \boldsymbol{z} \end{bmatrix} \le L_v^2,$$

$$V_t^\top K_{\mathcal{Q}_t^v}^{-1} V_t \le L_v^2,$$

$$\mathcal{L}_t^{\mathrm{MAP}}(U_t, \tilde{A}, V_t \mid D_t) \ge \mathcal{L}_t^{\mathrm{MAP}}(\hat{u}_t, \hat{A}_t, \hat{v}_t) - \beta_t(|\mathcal{Q}_t^{uv}|),$$

$$\ell_t(U_t \mid D_{\mathcal{Q}_t^u}) \ge \ell_t(\hat{U}_t \mid D_{\mathcal{Q}_t^u}) - \beta_t^u(|\mathcal{Q}_t^u|),$$

$$\ell_t(V_t \mid D_{\mathcal{Q}_t^v}) \ge \ell_t(\hat{V}_t \mid D_{\mathcal{Q}_t^v}) - \beta_t^v(|\mathcal{Q}_t^v|),$$

$$1 - \delta_A \ge A_{ij} \ge \delta_A, \ \forall i, j,$$

$$\sum_j A_{ij} = 1, \ \forall j \tag{72}$$

### E.3 PROJECTION WEIGHT FUNCTION

We decompose the projection weight function to the following:

$$w_t^u(x_t, x_t') := \|\overline{\delta}_t^u(x_t, x_t', :) - \underline{\delta}_t^u(x_t, x_t', :)\|, \tag{73}$$

where

$$\overline{\delta}_t^u(x_t, x_t', i) = \max_{\substack{U_t \in \mathbb{R}^{t+1}\ V_t \in \mathbb{R}^t, \boldsymbol{z} \in \mathbb{R} \\ \tilde{A} \in \mathbb{R}^{n^2},}} \boldsymbol{z} - \max_{t \in \mathcal{Q}_t^u} U_t$$

$$\text{s.t.} \quad \begin{bmatrix} U_t \\ \boldsymbol{z} \end{bmatrix}^\top K_{\mathcal{Q}_t^u, x}^{-1} \begin{bmatrix} U_t \\ \boldsymbol{z} \end{bmatrix} \le L_v^2,$$

$$V_t^\top K_{\mathcal{Q}_t^v}^{-1} V_t \le L_v^2,$$

$$\mathcal{L}_t^{\mathrm{MAP}}(U_t, \tilde{A}, V_t \mid D_t) \ge \mathcal{L}_t^{\mathrm{MAP}}(\hat{u}_t, \hat{A}_t, \hat{v}_t) - \beta_t(|\mathcal{Q}_t^{uv}|),$$

$$\ell_t(U_t \mid D_{\mathcal{Q}_t^u}) \ge \ell_t(\hat{U}_t \mid D_{\mathcal{Q}_t^u}) - \beta_t^u(|\mathcal{Q}_t^u|),$$

$$\ell_t(V_t \mid D_{\mathcal{Q}_t^v}) \ge \ell_t(\hat{V}_t \mid D_{\mathcal{Q}_t^v}) - \beta_t^v(|\mathcal{Q}_t^v|),$$

$$1 - \delta_A \ge A_{ij} \ge \delta_A, \ \forall i, j,$$

$$\sum_j A_{ij} = 1, \ \forall j \tag{74}$$

$$\underline{\delta}_t^u(x_t, x_t', i) = \min_{\substack{U_t \in \mathbb{R}^{t+1}\ V_t \in \mathbb{R}^t, \boldsymbol{z} \in \mathbb{R} \\ \tilde{A} \in \mathbb{R}^{n^2},}} \boldsymbol{z} - \max_{t \in \mathcal{Q}_t^u} U_t$$

$$\text{s.t.} \quad \begin{bmatrix} U_t \\ \boldsymbol{z} \end{bmatrix}^\top K_{\mathcal{Q}_t^u, x}^{-1} \begin{bmatrix} U_t \\ \boldsymbol{z} \end{bmatrix} \le L_v^2,$$

$$V_t^\top K_{\mathcal{Q}_t^v}^{-1} V_t \le L_v^2,$$

$$\mathcal{L}_t^{\mathrm{MAP}}(U_t, \tilde{A}, V_t \mid D_t) \ge \mathcal{L}_t^{\mathrm{MAP}}(\hat{u}_t, \hat{A}_t, \hat{v}_t) - \beta_t(|\mathcal{Q}_t^{uv}|),$$

$$\ell_t(U_t \mid D_{\mathcal{Q}_t^u}) \ge \ell_t(\hat{U}_t \mid D_{\mathcal{Q}_t^u}) - \beta_t^u(|\mathcal{Q}_t^u|),$$

$$\ell_t(V_t \mid D_{\mathcal{Q}_t^v}) \ge \ell_t(\hat{V}_t \mid D_{\mathcal{Q}_t^v}) - \beta_t^v(|\mathcal{Q}_t^v|),$$

$$1 - \delta_A \ge A_{ij} \ge \delta_A, \ \forall i, j,$$

$$\sum_j A_{ij} = 1, \ \forall j \tag{75}$$

# F    PROOF OF THEOREM 3.9

## F.1    PRELIMINARIES

We begin by introducing the known proofs.

### F.1.1    KNOWN RESULTS

We introduce useful theorems from literature.

**Theorem F.1** (Theorem 3.6 in Xu et al. (2024b)). *For any estimate $\tilde{v}_{t+1} \in \mathcal{B}_{t+1}^v$ measurable with respect to the filtration $\mathcal{F}_t$, we have, with probability at least $1 - \delta, \forall t \geq 1, (x, x') \in \mathcal{X} \times \mathcal{X}$,*

$$|(\tilde{v}_{t+1}(x) - \tilde{v}_{t+1}(x')) - (v(x) - v(x'))| \leq 2 \left(2L_v + \lambda^{-1/2} \sqrt{\beta(\epsilon, \delta/2, |\mathcal{Q}_t^v|)}\right) \sigma_{t+1}^{vv'}(x, x'). \quad (76)$$

*where*

$$\beta(\epsilon, \delta/2, |\mathcal{Q}_t^v|) = \mathcal{O}\left(\sqrt{|\mathcal{Q}_t^v| \log \frac{|\mathcal{Q}_t^v| \mathcal{N}(\mathcal{B}^v, \epsilon, \|\cdot\|_\infty)}{\delta}} + \epsilon |\mathcal{Q}_t^v| + \epsilon^2 |\mathcal{Q}_t^v|\right), \quad (77)$$

$$\left(\sigma_{t+1}^{vv'}(x, x')\right)^2 = k^{vv'}(\omega, \omega) - k^{vv'}(\omega_{1:t}, \omega)^\top \left(K_{t-1}^{vv'} + \lambda I\right) k^{vv'}(\omega_{1:t}, \omega), \quad (78)$$

$$k^{vv'}((x, x'), (y, y')) := k(x, y) + k(x', y'), \quad (79)$$

$$\omega := (x, x'), \quad (80)$$

$$\omega_{1:t-1} := ((x_\tau, x'_\tau))_{\tau=1}^{t-1}, \quad (81)$$

$$K_{t-1}^{vv'} := \left(k^{vv'}((x_{\tau_1}, x'_{\tau_1}), (x_{\tau_2}, x'_{\tau_2}))\right)_{\tau_1 \in [t-1], \tau_2 \in [t-1]}, \quad (82)$$

*and $\lambda$ is a positive regularization constant.*

### F.1.2    SUPPORTING RESULTS

We introduce the supporting lemmas for the main proof.

**Theorem F.2.** *With probability at least $1 - \delta$, for $\tilde{v}_t \in \mathcal{B}_t^v$ that is measurable with respect to the filtration $\mathcal{F}_t$,*

$$\sum_{t \in \mathcal{Q}_t^v} |\tilde{v}_t(x_t) - \tilde{v}_t(x'_t) - (v(x_t) - v(x'_t))| \leq \sum_{t \in \mathcal{Q}_t^v} w_t^v(x_t, x'_t) = \mathcal{O}\left(\sqrt{\beta_T \gamma_T^{vv'} |\mathcal{Q}_t^v|}\right) \quad (83)$$

*where*

$$\beta_T := \beta(1/T, \delta, |\mathcal{Q}_t^v|) \quad (84)$$

$$= \mathcal{O}\left(\sqrt{|\mathcal{Q}_t^v| \log \frac{|\mathcal{Q}_t^v| \mathcal{N}(\mathcal{B}^v, 1/T, \|\cdot\|_\infty)}{\delta}}\right), \quad (85)$$

$$\gamma_T^{vv'} := \max_{\Omega \subset \mathcal{X} \times \mathcal{X}; |\Omega| = |\mathcal{Q}_t^v|} \frac{1}{2} \log \left|I + \lambda^{-1} K_\Omega^{vv'}\right|, \quad (86)$$

$$K_\Omega^{vv'} := \left(k^{vv'}((x, x'), (y, y'))\right)_{(x, x'), (y, y') \in \Omega}. \quad (87)$$

*Proof.* The first inequality follows by the definition.

$$\sum_{t \in \mathcal{Q}_T^v} w_t^v(x_t, x_t') \tag{88}$$

$$= \sum_{t \in \mathcal{Q}_T^v} \sup_{\tilde{v} \in \mathcal{B}_t^v} |(\tilde{v}(x_t) - \tilde{v}(x_t')) - (v(x_t) - v(x_t'))| \tag{89}$$

$$\leq \sum_{t \in \mathcal{Q}_T^v} 2 \left( 2L_v + \lambda^{-1/2} \sqrt{\beta(\epsilon, \delta/2, |\mathcal{Q}_t^v|)} \right) \sigma_{t+1}^{vv'}(x, x') \tag{Thm. F.1}$$

$$\leq \left( 2L_v + \lambda^{-1/2} \sqrt{\beta(\epsilon, \delta/2, |\mathcal{Q}_T^v|)} \right) \sum_{t \in \mathcal{Q}_T^v} \sigma_{t+1}^{vv'}(x, x') \tag{Monotonicity of $\beta$ in $t$}$$

$$\leq \mathcal{O} \left( \sqrt{\beta_T \gamma_T^{vv'} |\mathcal{Q}_T^v|} \right), \tag{Lem. 4 in Chowdhury \& Gopalan (2017)}$$

$\square$

The old proof is below for reference. We will remove it upon acceptance.

**Lemma F.3 (Weighted utility difference bound (This will be removed)).** $\forall u \in \mathcal{B}^u, x, x' \in \mathcal{X}, 0 < w_i < 1, \overline{w} := \max_i w_i, \underline{w} := \min_i w_i, i \in \{1, 2\}$, *the weighted utility difference is bounded by*

$$\underline{w}(u(x) - u(x')) \leq w_1 u(x) - w_2 u(x') \leq \overline{w}(u(x) - u(x')), \tag{90}$$

*Proof.* We consider the following inequality:

$$w_1 u(x) - w_2 u(x') \leq \tilde{L}(u(x) - u(x')). \tag{91}$$

Given bound from Lemma B.1, $u(x) \in [-L_u, L_u]$, we consider the following extreme cases;

When $u(x) = L_u$ and $u(x') = -L_u$, Eq. (91) becomes:

$$w_1 L_u + w_2 L_u \leq \tilde{L}(L_u + L_u), \tag{92}$$

$$L_u(w_1 + w_2) \leq 2\tilde{L}L_u, \tag{93}$$

$$w_1 + w_2 \leq 2\tilde{L}. \tag{94}$$

Inversely, when $u(x) = -L_u$ and $u(x') = L_u$, then

$$-w_1 L_u - w_2 L_u \leq \tilde{L}(-L_u - L_u), \tag{95}$$

$$-L_u(w_1 + w_2) \leq -2\tilde{L}L_u, \tag{96}$$

$$w_1 + w_2 \geq 2\tilde{L}. \tag{97}$$

To satisfy both cases, we find

$$w_1 + w_2 = 2\tilde{L}, \tag{98}$$

By definition, we have

$$2\underline{w} \leq w_1 + w_2 \leq 2\overline{w}, \tag{99}$$

Therefore

$$\underline{w} \leq \tilde{L} \leq \overline{w}. \tag{100}$$

$\square$

**Lemma F.4 (Instantaneous regret).** $\forall u \in \mathcal{B}^u, x, x' \in \mathcal{X}, w_i \in \boldsymbol{w}, \overline{w} := \max_{i \in V} w_i, \underline{w} := \min_{i \in V} w_i$

$$\mathcal{A}[\tilde{u}_t(x^*, :)] - \mathcal{A}[\tilde{u}_t(x_t, :)]$$
$$\leq L_{\mathcal{A}} |\tilde{v}_t(x_t) - \tilde{v}_t(x_t') - (v(x_t) - v(x_t'))| \tag{101}$$
$$\leq L_{\mathcal{A}} w_t^u(x_t, x_{t-1})$$

*where $L_{\mathcal{A}} := \sqrt{n} \|\boldsymbol{w}\|$.*

*Proof.* By the acquisition function maximization, we have

$$\mathcal{A}[u(x^\star,:)] - \mathcal{A}[u(x_t,:)] \tag{102}$$

$$=\mathcal{A}[u(x^\star,:)] - \mathcal{A}[u(x_{t-1},:)] + \mathcal{A}[u(x_{t-1},:)] - \mathcal{A}[u(x_t,:)] \quad \text{(cancel out)}$$

$$\leq\mathcal{A}[\tilde{u}_t(x_t,:)] - \mathcal{A}[\tilde{u}_t(x_{t-1},:)] - (\mathcal{A}[u(x_t,:)] - \mathcal{A}[u(x_{t-1},:)]), \quad \text{(Optimality of Line 4 in Alg. 1)}$$

Here, the aggregation function $\mathcal{A}$ is the GSF in Eq. (2). The sorting function $\phi(\cdot)$ rearranges elements based on the current function estimate $\tilde{u}_t(x_{t-1},:)$. To clarify, we denote $\phi_{\tilde{u}_t}(\mathbf{y})$ as the sorting function that rearranges $\mathbf{y}$ based on $\tilde{u}_t(x_t,:)$. More formally,

$$\begin{cases} \phi_{\tilde{u}_t}(\mathbf{y}) & := \mathbf{y}[\mathrm{rank}(\tilde{u}_t(x_t,:))], \\ \phi_{\tilde{u}_{t-1}}(\mathbf{y}) & := \mathbf{y}[\mathrm{rank}(\tilde{u}_t(x_{t-1},:))], \\ \phi_{u_t}(\mathbf{y}) & := \mathbf{y}[\mathrm{rank}(u(x_t,:))], \\ \phi_{u_{t-1}}(\mathbf{y}) & := \mathbf{y}[\mathrm{rank}(u(x_{t-1},:))], \end{cases} \tag{103}$$

where $\mathrm{rank}(\cdot)$ is a function that rearranges the indices based on the input in ascending order, $\mathbf{y}[\cdot]$ is a rearranged vector of $\mathbf{y}$ based on the indices input. We use a slight abuse of notation to avoid unnecessary $x$ for representing the utility vectors. Then, our regret becomes

$$\mathbf{w}^\top \phi_{\tilde{u}_t}[\tilde{u}_t(x_t,:)] - \mathbf{w}^\top \phi_{\tilde{u}_{t-1}}[\tilde{u}_t(x_{t-1},:)] - (\mathbf{w}^\top \phi_{u_t}[u(x_t,:)] - \mathbf{w}^\top \phi_{u_{t-1}}[u(x_{t-1},:)]). \tag{104}$$

Here, Weymark (1981) proved that Hardy-Littlewood-Polya inequality is satisfied for GSF in Lemma 1. This implies that a mismatch between the sorting basis and the actual functions leads to the following inequality, e.g.,

$$\mathbf{w}^\top \underbrace{\phi_{\tilde{u}_t}[\tilde{u}_t(x_t,:)]}_{\tilde{u}_t=\tilde{u}_t} \leq \mathbf{w}^\top \underbrace{\phi_{\tilde{u}_{t-1}}[\tilde{u}_t(x_t,:)]}_{\tilde{u}_{t-1}\neq\tilde{u}_t}, \tag{105}$$

Therefore, we can further upper bound the Eq. (104):

$$\mathbf{w}^\top \phi_{\tilde{u}_t}[\tilde{u}_t(x_t,:)] - \mathbf{w}^\top \phi_{\tilde{u}_{t-1}}[\tilde{u}_t(x_{t-1},:)] - (\mathbf{w}^\top \phi_{u_t}[u(x_t,:)] - \mathbf{w}^\top \phi_{u_{t-1}}[u(x_{t-1},:)]), \tag{106}$$

$$\leq\mathbf{w}^\top \underbrace{\phi_{\tilde{u}_{t-1}}[\tilde{u}_t(x_t,:)]}_{\tilde{u}_{t-1}\neq\tilde{u}_t} - \mathbf{w}^\top \underbrace{\phi_{\tilde{u}_{t-1}}[\tilde{u}_t(x_{t-1},:)]}_{\tilde{u}_{t-1}=\tilde{u}_{t-1}} - (\mathbf{w}^\top \underbrace{\phi_{u_t}[u(x_t,:)]}_{u_t=u_t} - \mathbf{w}^\top \underbrace{\phi_{u_t}[u(x_{t-1},:)]}_{u_t\neq u_{t-1}}),$$

$$\text{(Lemma 1 in Weymark (1981))}$$

$$=\mathbf{w}^\top \phi_{\tilde{u}_{t-1}}[\tilde{u}_t(x_t,:) - \tilde{u}_t(x_{t-1},:)] - \mathbf{w}^\top \phi_{u_t}[u(x_t,:) - u(x_{t-1},:)]. \tag{107}$$

The last combination is assured by the additivity of GSF (Theorem 4 in Weymark (1981)).

Here, recall our acquisition function maximization is defined as the upper bound of the difference between utilities $\max_{\tilde{u}_t\in\mathcal{B}_t^u} \tilde{u}_t(x_t,:) - \tilde{u}_t(x_t,:)$. And Lemma 3.6 proves the true function is within this confidence interval. To maximize, the first and second terms become the upper and lower bounds within the confidence set, respectively. Thus we have

$$\tilde{u}_t(x_t,:) \geq \tilde{u}_t(x_{t-1},:), \tag{108}$$

$$\tilde{u}_t(x_t,:) \geq u(x_t,:), \tag{109}$$

$$u(x_{t-1},:) \geq \tilde{u}_t(x_{t-1},:), \tag{110}$$

$$u(x_t,:) \geq \tilde{u}_t(x_{t-1},:), \tag{111}$$

Here, GSF is the Schur-concave function (Theorem 1 in Weymark (1981)). Using this property and the last inequality, we have

$$\mathbf{w}^\top \phi_{u_t}[\tilde{u}_t(x_t,:) - \tilde{u}_t(x_{t-1},:)] \geq \mathbf{w}^\top \phi_{\tilde{u}_{t-1}}[\tilde{u}_t(x_t,:) - \tilde{u}_t(x_{t-1},:)]. \tag{112}$$

By using this inequality back to Eq. (107),

$$\mathbf{w}^\top \phi_{\tilde{u}_{t-1}}[\tilde{u}_t(x_t,:) - \tilde{u}_t(x_{t-1},:)] - \mathbf{w}^\top \phi_{u_t}[u(x_t,:) - u(x_{t-1},:)], \quad \text{(Eq. (107))}$$

$$\leq\mathbf{w}^\top \phi_{u_t}[\tilde{u}_t(x_t,:) - \tilde{u}_t(x_{t-1},:)] - \mathbf{w}^\top \phi_{u_t}[u(x_t,:) - u(x_{t-1},:)], \quad \text{(Eq. (112))}$$

$$\leq\mathbf{w}^\top \left(\phi_{u_t}[\tilde{u}_t(x_t,:) - \tilde{u}_t(x_{t-1},:) - u(x_t,:) - u(x_{t-1},:)]\right), \quad \text{(Additivity of GSF)}$$

$$\leq\|\mathbf{w}\|\|\phi_{u_t}[\tilde{u}_t(x_t,:) - \tilde{u}_t(x_{t-1},:) - u(x_t,:) - u(x_{t-1},:)]\|, \quad \text{(Cauchy-Schwarz)}$$

$$\leq\|\mathbf{w}\|\|\tilde{u}_t(x_t,:) - \tilde{u}_t(x_{t-1},:) - u(x_t,:) - u(x_{t-1},:)\|,$$

$$\text{(Euclidean norm is permutation-invariant)}$$

$$\leq\sqrt{n}\|\mathbf{w}\||\tilde{u}_t(x_t) - \tilde{u}_t(x_{t-1}) - u(x_t) - u(x_{t-1})|, \quad \text{(Cauchy-Schwarz for } n \text{ agents)}$$

$$=L_\mathcal{A}|\tilde{u}_t(x_t) - \tilde{u}_t(x_{t-1}) - u(x_t) - u(x_{t-1})|, \quad \text{(Define } L_\mathcal{A})$$

$$\leq L_\mathcal{A} w_t^u(x_t, x_{t-1}), \quad \text{(Definition of } w_t^u(x_t, x_{t-1}) \text{ (supremum))}$$

$\square$

## F.2 MAIN PROOF

### F.2.1 CASE-I: GIVEN AND INVERTIBLE MATRIX $A$

**Confidence set.** The ground-truth matrix $A$ is known and invertible. In this case, we do not need to learn $A$, so we can restrict the confidence set,

$$\mathcal{B}_t^v = \{\tilde{v} \mid \ell_t(\tilde{v} \mid D_{\mathcal{Q}_t^v}) \geq \ell_t^{\mathrm{MLE}} - \beta_t^v\}$$
$$\mathcal{B}_t^u = \{\tilde{u} \mid \ell_t(\tilde{u} \mid D_{\mathcal{Q}_t^u}) \geq \ell_t^{\mathrm{MLE}} - \beta_t^u\}$$
$$\mathcal{B}_t^{v,A,u} = \{(\tilde{v}, \tilde{A}, \tilde{u}) \mid \ell_t(\tilde{A}, \tilde{u}, \tilde{v} \mid D_{\mathcal{Q}_t^u}, D_{\mathcal{Q}_t^v}) \geq \ell_t^{\mathrm{MLE}} - \beta_t^{v,A,u}, \tilde{v} = \tilde{A}\tilde{u}, \tilde{A} = A, \tilde{v} \in \mathcal{B}_t^v, \tilde{u} \in \mathcal{B}_t^u\}.$$
$$(113)$$

**Instantaneous regret.**

$$\mathcal{A}[u(x^\star, :)] - \mathcal{A}[u(x_t), :]$$
$$\leq L_{\mathcal{A}} |\tilde{u}_t(x_t, :) - \tilde{u}_t(x_{t-1}, :) - (u(x_t, :) - u(x_{t-1}, :))|, \qquad \text{(Lemma F.4)}$$
$$\leq L_{\mathcal{A}} \|A^{-1}\| |\tilde{v}_t(x_t, :) - \tilde{v}_t(x_{t-1}, :) - (v(x_t, :) - v(x_{t-1}, :))|. \qquad \text{(Cauchy-Schwarz)}$$

**Cumulative regret.** By using Theorem F.2, our cumulative regret is:

$$R_T = \sum_{t \in [T]} \mathcal{A}[u(x^\star, :)] - \mathcal{A}[u(x_t), :]$$

$$\leq \sum_{t \in [T]} L_{\mathcal{A}} \|A^{-1}\| |\tilde{v}_t(x_t, :) - \tilde{v}_t(x_{t-1}, :) - (v(x_t, :) - v(x_{t-1}, :))|,$$

$$\leq \mathcal{O}\left(n L_{\mathcal{A}} \sqrt{\beta_T \gamma_T^{vv'} T}\right) \qquad \text{(Theorem F.2 and Lemma 2.7)}$$

**Cumulative queries.** Obviously, we do not even need to query the ground truth $u$. Thus,

$$|\mathcal{Q}_T^u| = 0. \qquad (114)$$

### F.2.2 CASE-II: UNKNOWN BUT IDENTIFIABLE $A$

**Confidence set.** Altough the matrix $A$ can be non-invertibible, the linear relationship can constrain the confidence set, as such,

$$\mathcal{B}_t^v = \{\tilde{v} \mid \ell_t(\tilde{v} \mid D_{\mathcal{Q}_t^v}) \geq \ell_t^{\mathrm{MLE}} - \beta_t^v\}$$
$$\mathcal{B}_t^u = \{\tilde{u} \mid \ell_t(\tilde{u} \mid D_{\mathcal{Q}_t^u}) \geq \ell_t^{\mathrm{MLE}} - \beta_t^u\}$$
$$\mathcal{B}_t^{v,A,u} = \{(\tilde{v}, \tilde{A}, \tilde{u}) \mid \ell_t(\tilde{A}, \tilde{u}, \tilde{v} \mid D_{\mathcal{Q}_t^u}, D_{\mathcal{Q}_t^v}) \geq \ell_t^{\mathrm{MLE}} - \beta_t^{v,A,u}, \tilde{v} = \tilde{A}\tilde{u}, \tilde{v} \in \mathcal{B}_t^v, \tilde{u} \in \mathcal{B}_t^u\}.$$
$$(115)$$

**Instantaneous regret.** The result is exactly the same with Lemma F.4.

**Cumulative regret.** Consider the following stopping criterion,

$$w_t^u(x_t, x_{t-1}) \geq \max\left\{\frac{1}{t^q}, w_t^v(x_t, x_{t-1})\right\},$$

Then, we have

$$\sum_{t \in [T]} \left( \mathcal{A}[u(x^\star, :)] - \mathcal{A}[u(x_t, :)] \right) \tag{116}$$

$$\leq L_{\mathcal{A}} \sum_{\tau \in [T]} w_t^u(x_\tau, x_{\tau-1}), \tag{Lemma F.4}$$

$$\leq L_{\mathcal{A}} \sum_{\tau \in \mathcal{Q}_T^u} w_\tau^u(x_\tau, x_{\tau-1}) + L_{\mathcal{A}} \sum_{\tau \in [T] \backslash \mathcal{Q}_T^u} \max \left\{ \frac{1}{t^q}, w_\tau^v(x_\tau, x_{\tau-1}) \right\}, \tag{stopping criterion}$$

$$\leq L_{\mathcal{A}} \sum_{\tau \in \mathcal{Q}_T^u} w_\tau^u(x_\tau, x_{\tau-1}) + L_{\mathcal{A}} \sum_{\tau \in [T] \backslash \mathcal{Q}_T^u} w_\tau^v(x_\tau, x_{\tau-1}) + L_{\mathcal{A}} \sum_{\tau \in [T] \backslash \mathcal{Q}_T^u} \frac{1}{\tau^q}, \tag{117}$$

$$\leq \mathcal{O} \left( L_{\mathcal{A}} \sqrt{\beta_T^u \gamma_T^{uu'} |\mathcal{Q}_T^u|} + L_{\mathcal{A}} \sqrt{\beta_T^v \gamma_T^{vv'} (T - |\mathcal{Q}_T^u|)} + L_{\mathcal{A}} (T - |\mathcal{Q}_T^u|)^{1-q} \right) \tag{Theorem F.2}$$

This is the tightest bound. For visibility and interpretability, we simplify

$$R_T \leq \mathcal{O} \left( L_{\mathcal{A}} T^{1-q} + L_{\mathcal{A}} \sqrt{\beta_T^u \gamma_T^{uu'} T} + L_{\mathcal{A}} \sqrt{\beta_T^v \gamma_T^{vv'} T} \right), \tag{$|\mathcal{Q}_T^u| < T$}$$

$$\leq \mathcal{O} \left( L_{\mathcal{A}} T^{1-q} + L_{\mathcal{A}} \sqrt{(\beta_T^u \gamma_T^{uu'} + \beta_T^v \gamma_T^{vv'}) T} \right), \tag{factor out}$$

**Cumulative queries.**

$$|\mathcal{Q}_T^u| = \sum_{t \in \mathcal{Q}_T^u} 1,$$

$$\leq T^q \sum_{t \in \mathcal{Q}_T^u} \frac{1}{t^q}, \tag{118}$$

$$\leq T^q \sum_{t \in \mathcal{Q}_T^u} w_t^u(x_t, x_{t-1}), \tag{stopping criterion}$$

$$= \mathcal{O} \left( T^q \sqrt{\beta_T^u \gamma_T^{uu'} |\mathcal{Q}_T^u|} \right) \tag{Theorem F.2}$$

Here, by setting $\epsilon = \frac{1}{T}$, we have

$$\beta_T^u = \mathcal{O} \left( \sqrt{|\mathcal{Q}_T^u| \log \frac{T \mathcal{N}(\mathcal{B}^u, 1/T, \|\cdot\|_\infty)}{\delta}} \right). \tag{119}$$

Hence,

$$|\mathcal{Q}_T^u| \leq \mathcal{O} \left( T^q |\mathcal{Q}_T^u|^{3/4} \sqrt{\gamma_T^{uu'}} \left( \log \frac{T \mathcal{N}(\mathcal{B}^u, 1/T, \|\cdot\|_\infty)}{\delta} \right)^{1/4} \right), \tag{120}$$

$$|\mathcal{Q}_T^u|^{1/4} \leq \mathcal{O} \left( T^q \sqrt{\gamma_T^{uu'}} \left( \log \frac{T \mathcal{N}(\mathcal{B}^u, 1/T, \|\cdot\|_\infty)}{\delta} \right)^{1/4} \right), \tag{121}$$

$$|\mathcal{Q}_T^u| \leq \mathcal{O} \left( T^{4q} (\gamma_T^{uu'})^2 \log \frac{T \mathcal{N}(\mathcal{B}^u, 1/T, \|\cdot\|_\infty)}{\delta} \right), \tag{122}$$

Here, we consider the optimal $q$. For upper bound, at least we want $|\mathcal{Q}_T^u| \leq T$, otherwise we have to query $u$ every iteration. Based on this, we have

$$\mathcal{O} \left( T^{4q} L_k \right) \leq T, \tag{123}$$

$$\mathcal{O} \left( T^{4q} \right) \leq T, \tag{124}$$

$$4q \leq 1, \tag{125}$$

$$q \leq \frac{1}{4}, \tag{126}$$

That is to say, by picking $q \leq \frac{1}{4}$, we can get a sublinear regret bound for the cumulative queries of $u$.

### F.2.3 CASE-III: UNIDENTIFIABLE $A$

**Confidence set.** The matrix $A$ and public $u$ information is not useful anymore. Thus, we just only query private $u$.

$$\mathcal{B}_t^u = \{\tilde{u} \mid \ell_t(\tilde{u} \mid D_{\mathcal{Q}_t^u}) \geq \ell_t^{\mathrm{MLE}} - \beta_t^u\} \tag{127}$$

**Instantaneous regret.**

$$\begin{aligned}
&\mathcal{A}[u(x^\star,:)] - \mathcal{A}[u(x_t),:] \\
&\leq L_{\mathcal{A}} |\tilde{u}_t(x_t,:) - \tilde{u}_t(x_{t-1},:) - (u(x_t,:) - u(x_{t-1},:))|,
\end{aligned} \qquad \text{(Lemma F.4)}$$

**Cumulative regret.** By using Theorem F.2, our cumulative regret is:

$$\begin{aligned}
R_T &= \sum_{t \in [T]} \mathcal{A}[u(x^\star,:)] - \mathcal{A}[u(x_t),:] \\
&\leq \sum_{t \in [T]} L_{\mathcal{A}} |\tilde{u}_t(x_t,:) - \tilde{u}_t(x_{t-1},:) - (u(x_t,:) - u(x_{t-1},:))|, \\
&\leq \mathcal{O}\left(L_{\mathcal{A}} \sqrt{\beta_T \gamma_T^{uu'} T}\right) \qquad \text{(Theorem F.2 and Lemma 2.7)}
\end{aligned}$$

**Cumulative queries.** Obviously, we end up querying the ground truth $u$ all the time. Thus,

$$|\mathcal{Q}_T^u| = T. \tag{128}$$

### F.3 PROOF OF THE KERNEL-SPECIFIC BOUNDS IN TABLE 1

To focus on kernel specific term only, we reduce the constants

$$R_T \leq \mathcal{O}\left(T^{1-\frac{q}{4}} + \sqrt{\beta_T \gamma_T^{vv'} T}\right), \tag{129}$$

$$|\mathcal{Q}_t^u| = T^q \left(\gamma_T^{vv'}\right)^2 \log \mathcal{N}(\mathcal{B}^v, T^{-1}, \|\cdot\|_\infty), \tag{130}$$

Recall

$$\beta_T = \mathcal{O}\left(\sqrt{T \log \frac{T \mathcal{N}(\mathcal{B}^v, 1/T, \|\cdot\|_\infty)}{\delta}}\right).$$

For kernel specific bound, we have,

**Linear kernel**

$$\log \mathcal{N}(\mathcal{B}^v, T^{-1}, \|\cdot\|_\infty) = \mathcal{O}\left(\log \frac{1}{\epsilon}\right) = \mathcal{O}(\log T).$$

The corresponding $k^{vv'}((x,x'),(y,y')) = x^\top y + {x'}^\top y' = \langle (x,x'),(y,y')\rangle$, which is also linear. Thus, by Theorem. 5 in Srinivas et al. (2012),

$$\gamma_T^{vv'} = \mathcal{O}(\log T).$$

Hence,

$$R_T \leq \mathcal{O}\left(T^{1-\frac{q}{4}} + T^{3/4}(\log T)^{3/4}\right), \tag{131}$$

$$|\mathcal{Q}_t^u| \leq \mathcal{O}\left(T^q (\log T)^3\right). \tag{132}$$

**Squared exponential kernel**

$$\log \mathcal{N}(\mathcal{B}^v, T^{-1}, \|\cdot\|_\infty) = \mathcal{O}\left((\log \frac{1}{\epsilon})^{d+1}\right) = \mathcal{O}\left((\log T)^{d+1}\right).$$

(Example 4, Zhou (2002)). By Thm. 4 in Kandasamy et al. (2015), we have,

$$\gamma_T^{vv'} = \mathcal{O}((\log T)^{d+1}).$$

Hence,

$$R_T \leq \mathcal{O}\left(T^{1-\frac{q}{4}} + T^{3/4}(\log T)^{3/4(d+1)}\right), \tag{133}$$

$$|\mathcal{Q}^u|_t \leq \mathcal{O}\left(T^q(\log T)^{3(d+1)}\right). \tag{134}$$

**Mátern kernel**

$$\log \mathcal{N}(\mathcal{B}^v, T^{-1}, \|\cdot\|_\infty) = \mathcal{O}\left((\frac{1}{\epsilon})^{d/\nu} \log \frac{1}{\epsilon}\right) = \mathcal{O}\left(T^{d/\nu} \log T\right).$$

(by Thm. 5.1 and Thm. 5.3 in Xu et al. (2024a)). By Thm. 4 in Kandasamy et al. (2015), we have,

$$\gamma_T^{vv'} = \mathcal{O}\left(T^{\frac{d(d+1)}{2\nu+d(d+1)}} \log T\right).$$

where $\nu > \frac{d(d+3+\sqrt{d^2+14d+17})}{4}$.

Hence,

$$R_T \leq \mathcal{O}\left(T^{1-\frac{q}{4}} + T^{\frac{d}{4\nu}+\frac{d(d+1)}{4\nu+2d(d+1)}}(\log T)^{3/4}\right), \tag{135}$$

$$|\mathcal{Q}^u|_t \leq \mathcal{O}\left(T^{q+\frac{d}{v}+\frac{2d(d+1)}{2\nu+d(d+1)}}(\log T)^3\right). \tag{136}$$

## G PROOF OF THEOREM 3.12

We first introduce the supporting results, then we prove Theorem G.

### G.1 SUPPORTING RESULTS

**Lemma G.1** (**Strongly convex MAP estimation**). *The log posterior defined in Eq. (44) is strongly convex with respect to $A$.*

*Proof.* By Eq. (44), the (unnormalised) negative log posterior can be written as:

$$\mathcal{L}_t^{\text{MAP}} := -\mathcal{L}_t(\hat{u}, \hat{A}, \hat{v}) - \log p(A) - \log p(u) - \log p(v) \tag{137}$$

Here, we assume the priors for $u$ and $v$ are the same uniform distribution $\mathcal{U}(u; -L_v, L_v)$, where the range is the same due to the range preservation Lemma 2.7. Then, its log prior becomes $-\log p(u) = -\log p(v) = \log(2L_v)$, and these are constant, thereby negligible in terms of the optimisation. Similarly, the normalising constant (also known as Bayesian evidence, marginal likelihood) is also constant, thereby negligible.

Original log likelihood function was convex yet not strongly convex with respect to $A$ because the Hessian matrix is positive semi-definite instead of positive definite. By adding the negative log prior term as regularliser, Eq. (137) becomes strongly convex.

As the sum of strongly convex functions is strongly convex, we will show the row-wise $A_i$ is strongly convex. First, we unpack the Eq. (137) for graph $A$ related parts. By symmetric structure for $u$ and $v$, we only extract for $u$ at $t = 1$ step for brevity,

$$\mathcal{L}^{\text{MAP}} = \log\left(e^{A_i^\top u(x,:)} + e^{A_i^\top u(x',:)}\right) - \mathbf{1}^u A_i^\top u(x,:) - (1 - \mathbf{1}^u)A_i^\top u(x,:)$$
$$- \sum_{i \in V}(\kappa_i - 1)\log A_i + \xi \sum_{i,j \in V} A_{ij}, \tag{138}$$

Then, we introduce the function $P = e^{A_i^\top u(x,:)} / e^{A_i^\top u(x,:)} + e^{A_i^\top u(x',:)}$. Then, the Hessian matrix $H$ is,

$$H_{jk} = \frac{\partial^2 \mathcal{L}^{\text{MAP}}}{\partial A_{ij} \partial A_{ik}}, \tag{139}$$

$$= P(1-P)(u(x,j) - u(x',j))(u(x,k) - u(x',k)) + \delta_{jk}\frac{\kappa_i}{A_i^2} + 2\xi \tag{140}$$

where $\delta_{jk}$ is the Kronecker delta. The first term is a positive semi-definite matrix as show in Lemma E.1. The second term is a diagonal matrix, which is positive definite because $\alpha_i - 1 > 0$ and $A_i > 0$. The sum of a positive definite matrix and positive semi-definite matrix is positive definite. Therefore, Hessian is positive definite.

The smallest eigenvalue of Hessian $\lambda_{\min}(H)$ is at least as large as the smallest eigenvalue of the positive definite matrix, namely the second term, as such:

$$\lambda_{\min}(H) \geq \frac{\kappa_i - 1}{\delta_A^2} + 2\xi > 0. \tag{141}$$

As we confirmed, all elements are positive, thereby the minimum eigenvalues of Hessian is strictly positive. Therefore, our MAP loss function is stringly convex with respect to $A$. $\square$

## G.2 MAIN PROOF

*Proof.* We first prove the graph estimation error, then we show the utility estimation error convergence.

### G.2.1 GRAPH IDENTIFICATION ERROR

Firstly, we define the minimum excess risk (MER; Xu & Raginsky (2022)):

$$\text{MER} := \inf_{\tilde{A}} \mathbb{E}_{D_{\mathcal{Q}_t^v}}[\mathcal{L}_{\text{MAP}}(\tilde{A})] - \mathcal{L}_{\text{MAP}}(A), \tag{142}$$

Here, $\mathcal{L}_{\text{MAP}}$ is 'omniscient' loss function when true hyperparamter $A$ is given. Even if $A$ estimation is perfect, our log likelihood estimate should have some error according to the randomness of the data generating process. Thus, the second term express the fundemental limit of Bayesian learning, and we can interpret this as aleatoric uncertainty. The first term, on the other hand, represents the empirical estimate from the observed data, i.e., $\mathbb{E}_{D_{\mathcal{Q}_t^u}}[\mathcal{L}_{\text{MAP}}(\tilde{A})] = \mathcal{L}_t^{\text{MAP}}(A)$. Thus, MER represents reducible risk by gathering more data, thus we can interpret MER as epistemic uncertainty.

Xu & Raginsky (2022) theoretically analyses the connection between MER and the information-theoretic quantity, particularly conditional mutual information, then derived the following asymptotic convergence rate,

$$\text{MER} \leq \mathcal{O}\left(\frac{n^2}{2|\mathcal{Q}_t^u|}\right), \tag{143}$$

for the logistic regression cases with binary feedback in Theorem 5, which is the same with our case.

By Definition A.3 of strong convexity,

$$\mathcal{L}_{\text{MAP}}(\hat{A}) \geq \mathcal{L}_{\text{MAP}}(A) + \nabla\mathcal{L}_{\text{MAP}}(A)^\top(\hat{A} - A) + \frac{m}{2}\|\hat{A} - A\|^2 \tag{144}$$

Since $A$ is the ground truth matrix, the optimality assures its gradient is zero. Thus, the inequality can simplify

$$\mathcal{L}_{\text{MAP}}(\hat{A}) \geq \mathcal{L}_{\text{MAP}}(A) + \frac{m}{2}\|\hat{A} - A\|^2, \tag{145}$$

$$\mathcal{L}_{\text{MAP}}(\hat{A}) - \mathcal{L}_{\text{MAP}}(A) \geq \frac{m}{2}\|\hat{A} - A\|^2, \tag{146}$$

This left-hand side is exactly the MER since $\mathcal{L}_{\text{MAP}}(A)$ is irreducible risk as $A$ is a ground-truth parameter, and $\hat{A} := \arg\min_{\tilde{A}\in\mathcal{A}^{u,v,B}} \mathcal{L}_t^{\text{MAP}}(\tilde{A} \mid D_{|\mathcal{Q}_t^u|})$ is reducible risk. Then, we have

$$\text{MER} \geq \frac{m}{2}\|\hat{A} - A\|^2, \tag{147}$$

By Eq. (143), we have

$$\|\hat{A} - A\|^2 \leq \mathcal{O}\left(\frac{mn^2}{2|\mathcal{Q}_t^u|}\right), \qquad \text{(Eqs. (143) and (147))}$$

$$\|\hat{A} - A\| \leq \mathcal{O}\left(\frac{n\sqrt{m}}{\sqrt{2|\mathcal{Q}_t^u|}}\right), \qquad \text{(square root)}$$

Next, we analyse the upper bound of strong convexity constant $m$. Recall Lemma G.1 that the strong convexity constant is only dependent on the prior term, thereby

$$m \leq \max_i \left(\frac{\kappa_i - 1}{A_{ij}^2}\right) + 2\xi, \tag{148}$$

$$\leq \frac{\overline{\kappa} - 1}{\delta_A^2} + 2\xi, \tag{149}$$

where $\overline{\kappa} = \max_{i,j \in V} \kappa_{i,j}$. As $\overline{\kappa}, \xi$ are user-defined parameters, we set

$$\frac{\overline{\kappa} - 1}{\delta_A^2} + 2\xi = \frac{1}{n^2}, \tag{150}$$

$$\overline{\kappa} = 1 + \frac{\delta_A^2}{n^2} - 2\xi\delta_A^2, \tag{151}$$

where $\kappa_i > 1, \xi < \frac{1}{2n^2}$ for the positivity. Then $\sqrt{m} \leq \frac{1}{n^2}$, leading to

$$\|\hat{A} - A\| \leq \mathcal{O}\left(\frac{1}{\sqrt{2|\mathcal{Q}_t^u|}}\right), \tag{152}$$

The equal constraint Eq. (150) shows that $\kappa_i$ almost equal to 1 for all $i, j$, we need 'flat' prior for graph $A$ estimate.

### G.2.2 POINTWISE UTILITY ESTIMATION ERROR

By definition of $w_t^u(x_t, x_t')$, which is the supremum of the pointwise error, we can upper bound

$$|\tilde{u}_t(x_t) - \tilde{u}_t(x_t') - (u(x_t) - u(x_t'))| \leq w_t^u(x_t, x_t'), \tag{153}$$

Here, we introduce the following notation:

$$W(T) := \sum_{\tau \in \mathcal{Q}_t^u} w(\tau) := w_t^u(x_t, x_{t-1}), \tag{154}$$

By Theorem F.1, $w(t)$ is submodular with respect to $t$, and $W(T)$ is monotonic with respect to $T$ because it is cumulative sum of positive values. Then by Theorem F.2, we have

$$W(T) \leq \mathcal{O}\left(\sqrt{\beta_T^u \gamma_T^{uu'} |\mathcal{Q}_T^u|}\right), \tag{155}$$

$$\leq \mathcal{O}\left(|\mathcal{Q}_T^u|^{3/4} \left(\gamma_T^{uu'}\right)^{1/2} \left(|\mathcal{Q}_t^u| \log \frac{T\mathcal{N}(\mathcal{B}^v, 1/T, \|\cdot\|_\infty)}{\delta}\right)^{1/4}\right), \tag{156}$$

$$= \mathcal{O}\left(|\mathcal{Q}_T^u|^{3/4} L_{k,\mathcal{Q}_t^u}^{1/4}\right), \tag{157}$$

where $L_{k,\mathcal{Q}_t^u} := \left(\gamma_T^{uu'}\right)^2 |\mathcal{Q}_t^u| \log \frac{T\mathcal{N}(\mathcal{B}^v, 1/T, \|\cdot\|_\infty)}{\delta}$ is the kernel-dependent term.

By submodularity and monotonicity, we have

$$w(t) \leq \frac{W(t)}{t} \tag{158}$$

for large $t$, i.e., $t \gg 1$, because

$$w(t) \geq w(T), \qquad\qquad\qquad \text{(submodularity, } t \leq T)$$

$$\sum_{\tau=1}^{T} w(\tau) \geq \sum_{\tau=1}^{T} w(T), \qquad\quad \text{(submodular inequality holds } \tau \leq T \text{ for all } \tau)$$

$$W(T) \geq Tw(T), \qquad\qquad\qquad\qquad (W(T) \text{ definition})$$

$$w(T) \leq \frac{W(T)}{T}. \tag{159}$$

Here, we assume the running horizons $Q, T \gg 1$. Then we have

$$w(|\mathcal{Q}_T^u|) \leq \mathcal{O}\left(\frac{|\mathcal{Q}_T^u|^{3/4} L_{k,Q}^{1/4}}{Q}\right) = \mathcal{O}\left(|\mathcal{Q}_T^u|^{-1/4} L_{k,\mathcal{Q}_t^u}^{1/4}\right). \tag{160}$$

$\square$

# H  EXTENSIONS

## H.1  GAUSSIAN PROCESS MODEL APPROACH

**Assumption H.1** (**Direct feedback**). *At step $t$, if query point $x_t$ is evaluated, we get a noisy evaluation of $u^{(i)}$, $u_t^{(i)} = u^{(i)}(x_t) + \xi_t$, where $\xi_t$ is i.i.d. $\sigma^{(i)}$-sub-Gaussian noise with fixed $\sigma^{(i)} > 0$.*

**Modelling.**   Considering the data-generation process under Defn. 2.5, we employ zero-mean multi-task GP (MTGP) regression model (Bonilla et al., 2007). For simplicity, define $u : \mathcal{X} \times E \to \mathbb{R}$ as our utility function taking location query $x$ and agent index as arguments, i.e. $u(x, i)$ is the utility for query $x$ and agent $i$. We place a prior over $u$ as $u \sim \mathcal{GP}(0, k_X \otimes k_E)^4$, which is distributed as,

$$u(x, :) \sim \mathcal{N}(0, k_X(x, x) \times K_E),$$

where $K_E$ is the kernel across agents and $k_X$ is the kernel across options $x$. Under the bandwagon model, with a graph $G$ (and its adjacency matrix $A$), we have,

$$v(x, :) = Au(x, :) \sim \mathcal{N}(0, k_X(x, x) \times AK_E A^\top), \tag{161}$$

due to the linearity of the graph convolution, we naturally get $v$ being itself an induced MTGP

$$v \sim \mathcal{GP}(0, k_X \times Ak_E A^\top), \text{ such that } \mathrm{Cov}(v(x, i), v(x', j)) = k_X(x, x') \times (AK_E A^\top)_{ij} \tag{162}$$

This holds true for the inverse case, where $B := (A + \lambda I)^{-1}$, resulting in $u(x, :) \approx Bv(x, :)$. Here, $\lambda$ is a regularization term and we typically set a fixed small positive value (e.g., 1e-4).

**Estimate social graph.**   Interestingly, Eq. (H.1) tells us that the graph adjacent matrix $A$ is merely the kernel hyperaparameter for $v$. Thus, similarly to optimise other kernel hyperparameters, we can estimate the graph $A$ (or equivalently, $B$) through maximum likelihood estimation (MLE) of log marginal likelihood (LML), with a slight modification:

$$\log \mathbb{P}(U_{\mathcal{Q}_t^u} \mid D_{\mathcal{Q}_t^v}, X_{\mathcal{Q}_t^u}, B) := \frac{1}{n} \sum_{i=1}^{n} \log \mathcal{N}\left(U_{\mathcal{Q}_t^u}^{(i)}; m_{\mathcal{Q}_t^v}^{u^{(i)}}(X_{\mathcal{Q}_t^u}), C_{\mathcal{Q}_t^v}^{u^{(i)}}(X_{\mathcal{Q}_t^u}, X_{\mathcal{Q}_t^u})\right), \tag{163}$$

where $D_{\mathcal{Q}_t^u} := (X_{\mathcal{Q}_t^u}, U_{\mathcal{Q}_t^u})$ is the $u$ observations, and $m_{\mathcal{Q}_t^v}^{u^{(i)}}$ and $C_{\mathcal{Q}_t^v}^{u^{(i)}}$ represent the predictive mean and covariance of $u(x, i)$ using the GP conditioned on $D_{\mathcal{Q}_t^v} := (X_{\mathcal{Q}_t^v}, V_{\mathcal{Q}_t^v})$ through Eq. (H.1).

$$m_{\mathcal{Q}_t^v}^{u^{(i)}}(x) = \left[k_X(x, X_{\mathcal{Q}_t^v}) \otimes k_E'^{(i)}\right]^\top \Sigma^{-1}(BV_{\mathcal{Q}_t^v}),$$

$$C_{\mathcal{Q}_t^v}^{u^{(i)}}(x, x') = k_X(x, x') \times k_E'^{(i)} - \left[k_X(x, X_{\mathcal{Q}_t^v}) \otimes k_E'^{(i)}\right]^\top \Sigma^{-1} \left[k_X(x, X_{\mathcal{Q}_t^v}) \otimes k_E'^{(i)}\right],$$

---

$^4\otimes$ denotes Kronecker product.

where $\Sigma := k_X(X_{\mathcal{Q}_t^v}, X_{\mathcal{Q}_t^v}) \otimes k_E^{\prime(i)} + D_{\sigma^2} \otimes I$, $k_E^{\prime(i)} := \left(BK_EB^\top\right)^{(i)}$ is the $i$-th column of the matrix $BK_EB^\top$, $D_{\sigma^2}$ is the diagonal matrix whose $(i,i)$-th element is the $v^{(i)}$ noise variance.

Note that the GP is conditioned on $v$, not $u$. $D_{\mathcal{Q}_t^u}$ is used as 'test dataset' to estimate $B$. This offers $|\mathcal{Q}_t^v| \neq |\mathcal{Q}_t^u|$, where typical MTGP requires $X_{\mathcal{Q}_t^v} = X_{\mathcal{Q}_t^u}$. Thus, this formulation allows us to separate the predictive contribution on $\mathcal{Q}_t^v$ and $\mathcal{Q}_t^u$, allowing *decoupled* query for $u$ and $v$.

### H.1.1 INFERENCE FROM DIRECT UTILITY FEEDBACK

To estimate the posterior predictive distribution of $u$ including the uncertainty of $B$ estimate requires extensive MCMC approximation. For the computational efficiency and closed-form propagation, we adopt the Laplace approximation:

$$\mathbb{P}(B \mid D_{\mathcal{Q}_t^u}, D_{\mathcal{Q}_t^v}) \approx \mathcal{N}(B; \hat{B}_{\mathcal{Q}_t}, \Lambda_{\mathcal{Q}_t}^{-1}),$$

$$\hat{B}_{\mathcal{Q}_t} = \arg\max_B \log \mathbb{P}(U_{\mathcal{Q}_t^u} \mid D_{\mathcal{Q}_t^v}, X_{\mathcal{Q}_t^u}, B), \quad \Lambda_{\mathcal{Q}_t} = -\nabla_B \nabla_B \log \mathbb{P}(U_{\mathcal{Q}_t^u} \mid D_{\mathcal{Q}_t^v}, X_{\mathcal{Q}_t^u}, B) \mid_{B=\hat{B}}.$$

The Hessian for covariance can be conveniently estimated via auto-differentiation. Now all of our variables are Gaussian, offering the closed-form uncertainty propagation:

**Corollary H.2 (Uncertainty propagation).** *Given $B \sim \mathcal{N}(B; \hat{B}_{\mathcal{Q}_t}, \Lambda_{\mathcal{Q}_t}^{-1})$ and $v(x,i) \mid D_{\mathcal{Q}_t^v} \sim \mathcal{GP}(m_{\mathcal{Q}_t^v}^{u^{(i)}}, C_{\mathcal{Q}_t^v}^{u^{(i)}})$, the posterior predictive distribution of $u$ becomes the closed-form:*

$$\mathbb{P}(u(x,i) \mid D_{\mathcal{Q}_t^u}, D_{\mathcal{Q}_t^v}) \approx \mathcal{N}\left(u(x,i); \mu_{\mathcal{Q}_t}^{u^{(i)}}(x), \sigma_{\mathcal{Q}_t}^{u^{(i)}}(x)\right)$$

$$\mu_{\mathcal{Q}_t}^{u^{(i)}}(x) = \hat{B}_{Q_t}(i,:)m_{Q_t^v}^v(x), \quad \left(\sigma_{\mathcal{Q}_t}^{u^{(i)}}(x)\right)^2 = \sum_{j=1}^n \left(\prod_{j=1}^n \bar{b}_{ij}^2 \bar{v}^{(j)^2}(x) - \prod_{j=1}^n \hat{b}_{ij}^2 m_{Q_t^v}^{v^{(j)^2}}(x)\right).$$

We obtain the closed-form UCB $\bar{u}(x,i) := m_{\mathcal{Q}_t}^{u^{(i)}}(x) + \beta_t^{1/2}\sigma_{\mathcal{Q}_t}^{u^{(i)}}(x)$ for the acqusition function, and $\Psi(B) := \mathcal{A}'[\text{diag}(\Lambda_{\mathcal{Q}_t}^{-1})]$ for stopping criterion.

*Proof.* Given graph convolution operation in Eq. 161, we can decompose the $i$-th truthful utility $u(x,i)$ as such:

$$u(x,i) = B(i,:)v(x,:) = \sum_{j=1}^n b_{ij}v(x,j), \tag{164}$$

where $b_{ij}$ is the element of $B$ at $i$-th row and $j$-th column. Here, $b_{ij}$ is independent of $v$, thus this linear operation is the product of two independent random variables. The expectation and variance of such case is known (Goodman, 1960), as such:

$$\mu_{\mathcal{Q}_t}^{u^{(i)}} = \mathbb{E}\left[\sum_{j=1}^n b_{ij}v(x,j)\right], \tag{Eq. 164}$$

$$= \sum_{j=1}^n \mathbb{E}[b_{ij}v(x,j)], \tag{linearity of expectation}$$

$$= \sum_{j=1}^n \mathbb{E}[b_{ij}]\mathbb{E}[v(x,j)], \tag{independence}$$

$$= \sum_{j=1}^n \hat{b}_{ij}m_{Q_t^v}^{v^{(j)}}(x), \tag{predictive mean of $b_{ij}$ and $u$}$$

$$= \hat{B}_{Q_t}(i,:)m_{Q_t^v}^v(x). \tag{inner product}$$

$$\sigma_{\mathcal{Q}_t}^{u^{(i)}{}^2}(x) = \mathbb{V}\left[\sum_{j=1}^n b_{ij}v(x,j)\right], \tag{Eq. 164}$$

$$= \sum_{j=1}^n \mathbb{V}\left[b_{ij}v(x,j)\right], \qquad \text{(independence)}$$

$$= \sum_{j=1}^n \left(\prod_{j=1}^n (\mathbb{V}[b_{ij}] + \mathbb{E}[b_{ij}]^2)(\mathbb{V}[v(x,j)] + \mathbb{E}[v(x,j)]^2) - \prod_{j=1}^n \mathbb{E}[b_{ij}]^2\mathbb{E}[v(x,j)]^2\right) \qquad \text{(independence)}$$

$$= \sum_{j=1}^n \left(\prod_{j=1}^n (\sigma_{b_{ij}}^2 + \hat{b}_{ij}^2)(C_{Q_t^v}^{v^{(j)}}(x) + m_{Q_t^v}^{v^{(j)}{}^2}(x)) - \prod_{j=1}^n \left(\hat{b}_{ij}m_{Q_t^v}^{v^{(j)}}(x)\right)^2\right) \qquad \text{(posterior mean)}$$

$$= \sum_{j=1}^n \left(\prod_{j=1}^n \bar{b}_{ij}^2\bar{v}^{(j)}{}^2(x) - \prod_{j=1}^n \hat{b}_{ij}^2 m_{Q_t^v}^{v^{(j)}{}^2}(x)\right), \qquad \text{(notation change)}$$

where $\bar{b}_{ij}^2 := \sigma_{b_{ij}}^2 + \hat{b}_{ij}^2$ and $\bar{v}^{(j)}{}^2(x) := C_{Q_t^v}^{v^{(j)}}(x) + m_{Q_t^v}^{v^{(j)}{}^2}(x)$. $\qquad\square$

## H.2 POSITIVE SOCIAL INFLUENCE

When consider the our objective is now changed to:

$$x^\star = \arg\max_{x\in\mathcal{X}} \mathcal{A}[v(x,:)]. \tag{165}$$

Note that now we consider $v(x,:)$ as the truthful utilities, and more costly to query than $u(x,:)$. We can think individual utility $u(x,:)$ can be based on misunderstanding, but the meeting can mitigate this confusion, then we get truthful $v(x,:)$ thanks to the social interaction. This can be seen in real-world, highlighting our OpenReview discussion is exactly the same, where each review is based on individual utility, yet the discussion can mitigate this misunderstanding thus the utility after discussion $v(x,:)$ can be regarded as truthful. Then, our problem becomes easier than before; we can cheaply observe $u(x,:)$, and $v(x,:) = Au(x,:)$, meaning that we do not need to consider the invertibility issues. Moreover, importantly, the impossibility theorem also can be mitigated. That means, of course $\mathcal{A}[u(x,:)] \neq \mathcal{A}[v(x,:)]$ is still valid, yet we can say the following Pareto Front containment Theorem:

**Theorem H.3 (Pareto Front Containment).** *For any social graph G the Pareto front corresponding to non-truthful utilities is a subset of the original Pareto front of truthful utilities.*

*Proof.* Before we define the Pareto fronts of truthful and non-truthful utilities, we define two preferences $\succeq_u$ and $\succeq_v$ as

$$x \succeq_u x' \quad \text{iff} \quad u(x,i) \geq u(x',i) \quad \forall i \tag{166}$$
$$x \succeq_v x' \quad \text{iff} \quad v(x,i) \geq v(x',i) \quad \forall i \tag{167}$$

Consider the Pareto front of truthful utilities as $P_u := \{x | \forall x' \in \mathcal{X} \, x' \not\succ_u x\}$ and analogously for non-truthful utilities as $P_v := \{x | \forall \, x' \in \mathcal{X} \, x' \not\succ_v x\}$. We want to show that $P_v \subseteq P_u$. Before we show that, we prove that the following holds for any unknown social influence graph $G$

$$\forall \, x, x' \in \mathcal{X} \, x \succeq_u x' \implies x \succeq_v x'$$

Given the transition matrix of graph $G$ as $A$, we know that $a_{ij} \geq 0$ and $\sum_j a_{ij} = 1$. Consider

$$x \succeq_u x'$$
$$\forall i \; u(x, i) \geq u(x', i) \quad \text{(Definition 167)}$$
$$\sum_i a_{ij} u(x, i) \geq \sum_i a_{ij} u(x', i) \quad (a_{ij} \geq 0)$$
$$A_j^T u(x, :) \geq A_j^T u(x', :) \quad (A_i := j^{\text{th}} \text{ row of } A)$$
$$v(x, j) \geq v(x', j) \quad \forall j$$
$$x \succeq_v x'$$

Since, the Pareto set is defined as $P_v = \{x | \forall \, x' \in \mathcal{X} x' \nsucc_v x\}$, we can rewrite the condition of $x$ not being strictly dominated by $x'$ as either $x$ weakly domainates $x'$ or is incomparable

$$x' \nsucc_v x = (x \succeq_v x') \vee [(x \nsucceq_v x') \wedge (x' \nsucceq_v x)]$$
$$= [(x \succeq_v x') \vee (x \nsucceq_v x')] \wedge [(x \succeq_v x') \vee (x' \nsucceq_v x)]$$
$$= (x \succeq_v x') \vee (x' \nsucceq_v x)$$

Then $P_v = \{x | \forall \, x' \in \mathcal{X} (x \succeq_v x') \vee (x' \nsucceq_v x)\}$ and similarly, $P_u = \{x | \forall \, x' \in \mathcal{X} (x \succeq_u x') \vee (x' \nsucceq_u x)\}$. To show that $P_v \subseteq P_u$, consider $x \in P_v$, then $x$ needs to satisfy $(x \succeq_v x') \wedge (x' \nsucceq_v x)$ $\forall x' \in \mathcal{X}$. We divide the condition into two cases,

**Case 1:** $\forall x' \in \mathcal{X} \; x' \nsucceq_v x$

Since $x \succeq_u x' \implies x \succeq_v x'$, considering the contrapositive we have $x' \nsucceq_v x \implies x \nsucceq_u x'$. Then $x \in P_u$.

**Case 2:** $\forall x' \in \mathcal{X} \; x \succeq_v x'$

This case is rarer as it implies that $P_v = \{x\}$. We can re-write the case as, $\forall x' \in \mathcal{X}$

$$\forall j \quad v(x, j) \geq v(x', j) \quad \text{(Definition 167)}$$
$$\forall j \quad A_j^T u(x, :) \geq A_j^T u(x', :)$$

By fixing any $j$ we can say that $x = \arg\max_{x \in \mathcal{X}} A_j^T u(x, :)$. Therefore $x \in P_u$. $\qquad \square$

Therefore, we can use private votes to explore the Pareto front $\mathcal{X}^\star$, then the consensus is contained $x^\star \in \mathcal{X}^\star$. This can naturally lead to the combination of multi-objective BO, where we use private vote $u$ to estimate the Pareto Front $\mathcal{X}^\star$, then we search the consensus $x^\star$ within the estimated Pareto Front set.

Let $\text{vol}(\mathcal{X})$ be the volume of the domain. Then, we have $\text{vol}(\mathcal{X}^\star) \leq \text{vol}(\mathcal{X})$ since $\mathcal{X}^\star \subseteq \mathcal{X}$. As shown in Kandasamy et al. (2016), the maximum information gain (MIG) depends on the domain volume. Let $\Psi_t(\mathcal{X})$ be the MIG at $t$-th iteration over the domain $\mathcal{X}$. For instance, with the squared exponential kernel, MIG can be expressed as $\Psi_t(\mathcal{X}) \propto \text{vol}(\mathcal{X}) \log(t)^{d+1}$. This means that the regret of our algorithm should improve by a factor of $\Psi_t(\mathcal{X}^g)/\Psi_t(\mathcal{X}) = \text{vol}(\mathcal{X}^g)/\text{vol}(\mathcal{X})$. Thus, in this case, we can provably better regret convergence bound when individual votes are available. This is actually what our OpenReview system does.

## I    HYPERPARAMETERS

We summarized the comprehensive list of hyperparameters used in this work and their settings in Table 4. Most of these are standard in typical GP-UCB approaches. The newly introduced hyperparameters are primarily tunable in a data-driven manner, and we provided a sensitivity analysis in the experiment section for those that are not.

### I.1    UPDATE KERNEL HYPERPARAMETERS

By Assumption 2.6, there exists a large enough constant $L_v$ that upper bounds the norm of the ground-truth latent black-box utility function $u, v$. However, a tight estimate of this upper bound

may be unknown to us in practice, while the execution of our algorithm explicitly relies on knowing a bound $L_v$ (in Prob. (5), $L_v$ is a key parameter).

So it is necessary to estimate the norm bound $L_v$ using the online data. Suppose our guess is $\hat{L}$. It is possible that $\hat{L}$ is even smaller than the ground-truth function norm $\|v\|$. To detect this underestimate, we observe that, with the correct setting of $L_v$ such that $L_v \geq \|v\|$, we have that by Lemma 3.6 and the definition of MAP estimate,

$$\mathcal{L}_t^{\mathrm{MAP}}(\hat{u}_{t|\hat{L}}, \hat{A}_{t|\hat{L}}, \hat{v}_{t|\hat{L}}) \geq \mathcal{L}_t^{\mathrm{MAP}}(u, A, v) \geq \mathcal{L}_t^{\mathrm{MAP}}(\hat{u}_{t|\hat{L}}, \hat{A}_{t|\hat{L}}, \hat{v}_{t|\hat{L}}) - \beta_{t|\hat{L}},$$

where $\hat{u}_{t|\hat{L}}$ is the MAP estimate function with function norm bound $\hat{L}$ and $\beta_t$ is the corresponding parameter as defined in Lemma 3.6 with norm bound $\hat{L}$. We also have $2\hat{L}$ is a valid upper bound on $\|v\|$ and thus,

$$\mathcal{L}_t^{\mathrm{MAP}}(\hat{u}_{t|2\hat{L}}, \hat{A}_{t|2\hat{L}}, \hat{v}_{t|2\hat{L}}) \geq \mathcal{L}_t^{\mathrm{MAP}}(u, A, v) \geq \mathcal{L}_t^{\mathrm{MAP}}(\hat{u}_{t|2\hat{L}}, \hat{A}_{t|2\hat{L}}, \hat{v}_{t|2\hat{L}}) - \beta_{t|2\hat{L}},$$

Therefore,

$$\mathcal{L}_t^{\mathrm{MAP}}(\hat{u}_{t|\hat{L}}, \hat{A}_{t|\hat{L}}, \hat{v}_{t|\hat{L}}) \geq \mathcal{L}_t^{\mathrm{MAP}}(u, A, v) \geq \mathcal{L}_t^{\mathrm{MAP}}(\hat{u}_{t|2\hat{L}}, \hat{A}_{t|2\hat{L}}, \hat{v}_{t|2\hat{L}}) - \beta_{t|2\hat{L}},$$

That is to say, $\mathcal{L}_t^{\mathrm{MAP}}(\hat{u}_{t|\hat{L}}, \hat{A}_{t|\hat{L}}, \hat{v}_{t|\hat{L}})$ needs to be greater than or equal to $\mathcal{L}_t^{\mathrm{MAP}}(\hat{u}_{t|2\hat{L}}, \hat{A}_{t|2\hat{L}}, \hat{v}_{t|2\hat{L}}) - \beta_{t|2\hat{L}}$ when $\hat{L}$ is a valid upper bound on $\|v\|$.

Therefore, we can use the heuristic: every time we find that

$$\mathcal{L}_t^{\mathrm{MAP}}(\hat{u}_{t|\hat{L}}, \hat{A}_{t|\hat{L}}, \hat{v}_{t|\hat{L}}) < \mathcal{L}_t^{\mathrm{MAP}}(\hat{u}_{t|2\hat{L}}, \hat{A}_{t|2\hat{L}}, \hat{v}_{t|2\hat{L}}) - \beta_{t|2\hat{L}},$$

we double the upper bound guess $\hat{L}$.

### I.2 OPTIMIZE THE KERNEL HYPERPARAMTERS

Unlike the GP, our likelihood model does not have the analytical form of marginal likelihood. Thus, we adopt the leave-one-out cross-validation (LOO-CV) as the optimization loss (See Section 5.3 in Williams & Rasmussen (2006)). That is, we leave one out from the observed dataset and compute the negative log posterior of the left one dataset, and averaging all samples. We optimize the kernel hyperparameters by minimizing this LOO-CV.

## J EXPERIMENTS

### J.1 TOY EXAMPLE 1

The instrinsic utilities of the influencer ($u^{(1)}$) and the follower ($u^{(2)}$) are defined as follows:

$$u^{(1)} := 0.3\mathcal{N}(x; 0.35, 0.05) + 1.2\mathcal{N}(x; 0.45, 0.18) + 0.8\mathcal{N}(x; 0.75, 0.1), \tag{168}$$

$$u^{(2)} := 0.5\mathcal{N}(x; 0.25, 0.1) + 0.8\mathcal{N}(x; 0.65, 0.15) + 0.4\mathcal{N}(x; 0.85, 0.05), \tag{169}$$

The social-influence graph is defined as $A := \begin{pmatrix} 0.9 & 0.1 \\ 0.6 & 0.4 \end{pmatrix}$, and the aggregation function is utilitarian $\mathcal{A} := \sum_{i=1}^{2} \frac{1}{2} u^{(i)}$.

### J.2 REAL-WORLD TASKS

#### J.2.1 THERMAL COMFORT

We compute the utility function using the PMV (predicted mean vote), which is the estimation from the large collection of human preference dataset that predicts the values from $[-3, 3]$, where $-3$ means very cold, and $3$ means very hot, and $0$ means comfortable at the current condition. We take the minus absolute PMV, with $0$ is the maximum for each agent's utility. The input values are two dimensional continuous values $x \in \mathcal{R}^2$, where the first dimension denotes the temperature $[15, 35]$ with degrees Celcius, and the second dimension is the air velocity $[0.3, 1.5]$.

The agents has different conditions of the garments and activity conditions, resulting in the relative air velocity difference. The conditions are as follows:

**Activity**

(agent 1)  Seated, heavy limb movement
(agent 2)  House cleaning
(agent 3)  Writing

**Garments**

(agent 1)  'Executive chair', 'Thick trousers', 'Long-sleeve long gown', "Boots", 'Ankle socks'
(agent 2)  "Thin trousers", "T-shirt", "Shoes or sandals"
(agent 3)  'Standard office chair', 'Long sleeve shirt (thin)', 'Long-sleeve dress shirt', 'Slippers'

Then PyThermalComfort (Tartarini & Schiavon, 2020) computes the corresponding metabolic generation and thermal insulation based on ASHRAE industrial standard. The social-influence graph is defined as $A := \begin{pmatrix} 0.8 & 0.1 & 0.1 \\ 0.6 & 0.1 & 0.3 \\ 0.4 & 0.3 & 0.3 \end{pmatrix}$, and the aggregation function is egalitarian $\rho = 0.1$.

J.2.2   TEAMOPT

We have modified the setting from Wan et al. (2023); Adachi et al. (2024a). Each team is represented by graphs with 8 members from 11 candidates. Such teams are positioned on the node of the supergraph, of which edge is the similarity between teams defined as the Jaccord index. There are two additional information on nodes; skills and inter-member compatibility. Both are represented as continuous values, generated from row-wise Dirichlet distribution, resulting in $n \times n$ square matrices. There are four agents who has the different utility function.

**Utility functions**

(agent 1)  Skill set diversity: this is measured by the entropy of the skill set matrix, assuming the optimal team is when each member is specialised in one skill, and the whole skill distribution is close to uniform.
(agent 2)  mean compatibility: averaging the compatibility matrix.
(agent 3)  minimax compatibility: take minimum element of the compatibility matrix.
(agent 4)  flat compatibility: take minimum variance of the compatibility matrix.

The social-influence graph is defined as $A := \begin{pmatrix} 0.7 & 0.1 & 0.1 & 0.1 \\ 0.4 & 0.4 & 0.1 & 0.1 \\ 0.3 & 0.2 & 0.3 & 0.2 \\ 0.3 & 0.2 & 0.2 & 0.3 \end{pmatrix}$, and the aggregation function is egalitarian $\rho = 0.1$.

J.2.3   TRIPADVISER:

We used the TripAdvisor New Zealand Hotel dataset (Rahman, 2023) which consists of three dimensional data, price, number of review, and review rank. We denote $p(x)$ as price, $r(x)$ as number of review, and $R(x)$ as the review rank. There are three agents who has the different utility function.

**Utility functions**

(agent 1)  luxury: 0.5 p(x) + 0.5 R(x)
(agent 2)  budget: -0.5 p(x) + r(x) + 0.5 r(x)
(agent 3)  review: R(x)

The social-influence graph is defined as $A := \begin{pmatrix} 0.5 & 0.3 & 0.2 \\ 0.1 & 0.8 & 0.1 \\ 0.2 & 0.1 & 0.7 \end{pmatrix}$, and the aggregation function is egalitarian $\rho = 0.5$.

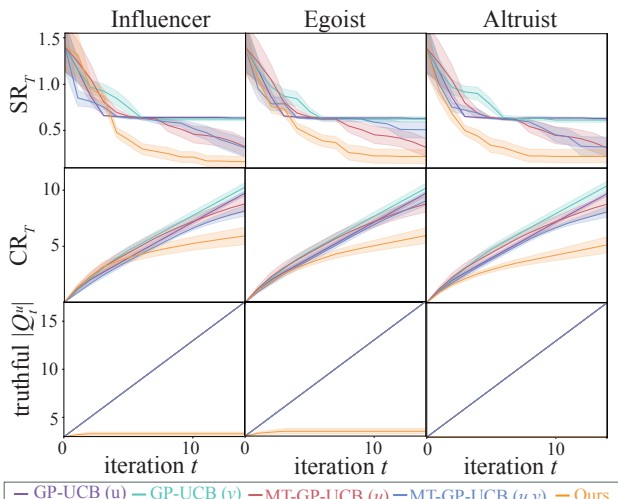

Figure 6: Simple regret, cumulative regret, and cumulative queries on different social influence graph.

### J.2.4 ENERGYTRADING:

We used METER dataset, UK-wide energy demand dataset (Grünewald & Diakonova, 2020; Grünewald & Diakonova, 2019). Due to the privacy reason, we cannot identify which smart meter id corresponds to the geography in the UK. Therefore, we place the all meter dataset into the hypothetical two-dimensional space, which placed based on the similarity between the time series data. Intuitively, the energy demand should have some geographical relationship, for instance Scotland is cooler than England, thereby the heating energy is more necessary. As such estimated two dimensional space locations are further transformed into continuous space, by interpolating by GP model. We refer to this GP as oracle GP. We also assume three firms use GPs as their demand prediction model. We refer to these individual GPs as internal models. As we use GP, the maximum information gain can be reasonably approximated by the predictive variance (Srinivas et al., 2010). Then, we allocate different number of datasets to each internal models, resulting in different information gain functions. We use this predictive variance of each internal model as the utility functions. A large corporation is assumed to have the largest and diverse data, and a niche startup has handful of data but is very niche where the large corporation does not have. A joint venture is the mixture of niche data. These data distribution creates the different maximizers of utility functions.

The social-influence graph is defined as $A := \begin{pmatrix} 0.8 & 0.1 & 0.1 \\ 0.6 & 0.1 & 0.3 \\ 0.4 & 0.3 & 0.3 \end{pmatrix}$, and the aggregation function is egalitarian $\rho = 0.5$.

### J.3 GAUSSIAN PROCESS BASED MODEL

Now we tested our GP-based algorithm with popular baselines: (a) GP-UCB (Srinivas et al., 2010): This baseline solves Prob. (1) as a single-objective BO, conditioning a GP on aggregated votes, $y_t := \mathcal{A}[u(x_t, :)]$. We compare scenarios where the GP is conditioned on truthful utilities $(u)$ versus non-truthful ones $(v)$. (b) Multi-task (MT) GP-UCB (Kandasamy et al., 2016): This baseline models utilities using a simple MTGP without incorporating the graph structure, employing our acquisition function from Prob. (5) for querying. We compare cases where the GP is conditioned only on truthful utilities $(u)$ versus both truthful and non-truthful utilities $(u, v)$. If the non-truthful utilities $(v)$ provide useful low-fidelity information, the convergence rate for $(u, v)$ should improve relative to $(u)$ alone; otherwise, it might remain the same or deteriorate.

**Social-influence graph.** To examine the effect of the social-influence graph $A$, we varied $A$ using the same funcitons and graph in Figure 3. Figure 6 highlights the robustness and efficacy of our algorithm. Our approach consistently outperforms the baselines in both simple and cumulative regret. Notably, the cumulative number of truthful queries $|\mathcal{Q}_T^u|$ grows logarithmically, requiring

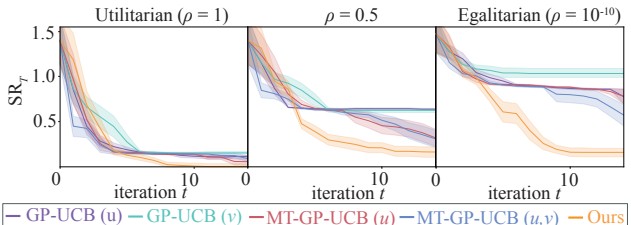

Figure 7: Simple regret, cumulative regret, and cumulative queries on different aggregation function.

only a few queries overall, while the baselines require a linear growth in $|\mathcal{Q}_T^u|$. This demonstrates our algorithm's sample efficiency for expensive $u$ queries. A closer look reveals that GP-UCB often gets stuck in local maxima. This is a known limitation of GP-UCB under model misspecification (Berkenkamp et al., 2019), which necessitates additional exploration through an increased $\beta$ parameter (Bogunovic & Krause, 2021), requiring more iterations for $\beta$ to grow. In contrast, our convolutional kernel GP captures the correlations in corrupted $v$, providing better extrapolation and avoiding misspecification issues compared to the vanilla GP model. Contrastingly, a naïve combination of MT-GP-UCB $(u, v)$ fails to accelerate in most cases because standard correlation learning in MTGP is not the convolution learning. It often performs worse than using only truthful data $(u)$, as Mikkola et al. (2023) explains that incorporating unreliable information can actually decelerate convergence.

**Aggregate function.** We further tested the effect of the aggregation function by varying $\rho \in [1, 0.5, 10^{-10}]$ in Eq. (2), keeping the influencer-follower matrix $A$ fixed. Smaller $\rho$ values lead to more pronounced differences in our model. This makes sense, as $\rho$ controls the degree to which minority preferences, such as those of the follower in this case, are prioritized, thereby accentuating model misspecification issues as $\rho$ decreases.

## J.4 COMPUTATION TIME

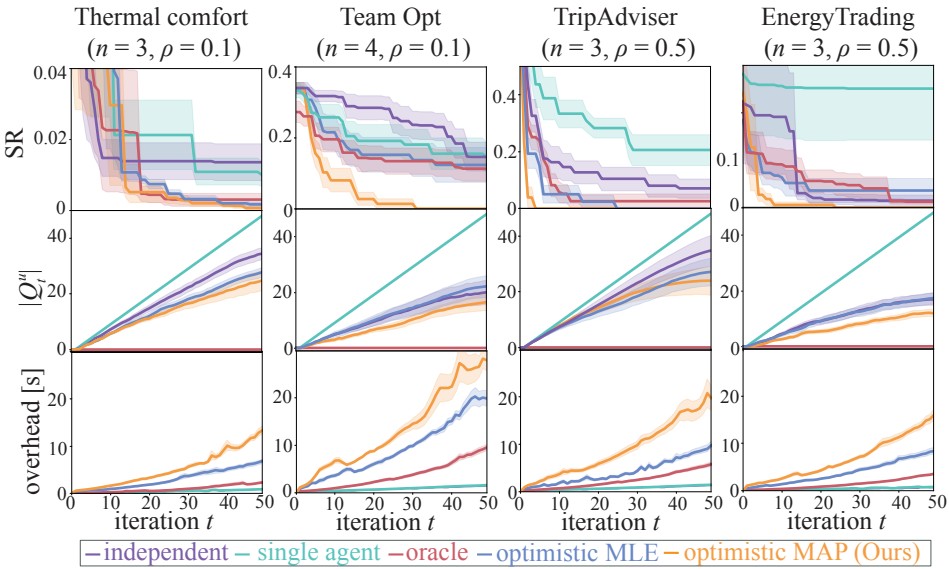

Figure 8: Computation time on real-world tasks.

We report the computation time on Figure 8. As we can see, although our algorithm is the slowest, each query only takes within 30 seconds at iteration $t = 50$. The complexity is $\mathcal{O}(n(t + n))$, and the computation time scales linearly as shown in the figure. Taking tens of seconds in multi-agentic scenario is common as it is inherently expensive computation.

