# OpenReview forum: "Social Bayesian Optimization for Building Truthful Consensus"
_ICLR.cc/2025/Conference — Submitted to ICLR 2025_

### Official Review · Reviewer_DBM8 · 2024-10-26

**Soundness:** 3
**Presentation:** 3
**Contribution:** 4
**Rating:** 5
**Confidence:** 2

**Summary:**

This paper studies an online setting of learning the most favorable candidate among agents. The utility of the agents are unknown and need to be elicited from query on binary relations between candidates. Moreover, agents reported preferences may be influenced by their neighbors on an unknown social network. While no non-dictator aggregation rule can reach the correct candidate under the influence of noisy preference, this paper circumvent the barrier by introducing extra costly yet truthful query. These query can reveal the social influence and therefore infer the true preferences from the noisy queries. There theoretical results bound the regret of their algorithm. They also run experiments to demonstrate the robustness efficiency of the algorithm on synthetic and real-world data.

**Strengths:**

The paper studies an important question under a highly non-trivial model. While many paper in our literature adopt the utility model to analyze the welfare, it is always worth a question whether people really know quantitively about their utility.  The paper also well motivates why this problem is non-trivial and why the challenges are not addressed by the previous work.

The research has a clear and complete structure. The introduction motivates the significance and non-triviality of the problem. Impossibility results distinguish their contribution from previous work. A constructive contribution (the algorithm) is proposed with both theoretical guarantee and empirical tests.

**Weaknesses:**

However, I have doubt on the soundness of theoretical results in this paper. I check the proof Theorem 3.3 and Theorem 3.9 and the statement of Theorem 3.12, and have questions on all of them.

Theorem 3.3
For the |X| = 2 (binary decision) case, the proof directly show that any aggregation rule is dictatorship, which is very counterintuitive. Following Arrow's definition, an aggregation rule A is dictatorship if [there exists an agent i such that for any alternative (x, x') and any utility profile u, x >_i x' iff x >_{A(u)} x'}]. Now just consider A be the majority rule (one two candidates). Then any agent i cannot be a dictator when he/she is on the minority. (I think it is more reasonable to show that non-dictatorship and group-thinking-proof can not exist together.)

For the |X| > 2 case, the proof restrict the result to a subcase where every two agents have distinct favorite candidate. This directly requires |X| > |V| (the number of candidates is larger than the number of agents), which is very restrictive (especially in voting scenarios). Therefore, the theorem claims more than it actually proves. (On the other hand, I doubt whether it is necessary to have such restriction.)

Theorem 3.9
Lemma F.3 is very counterintuitive. Firstly, w1 and w2 is not defined. More importantly, I don't think this inequality holds without strong     assumption unlisted. For example, when u(x) = 0,  u(x') < 0, and w1 < w2, lower{w}(u(x) − u(x′)) ≤ w1u(x) − w2u(x′) is apparently violated, as 0 > lower{w}  * u(x') > w2 * u(x'). The other inequality can be easily violated similarly.

Theorem 3.12
I didn't check the proof, but your high-level claim does not match your statement. The statement gives to upper bounds, which only shows that ''the worst convergence rate of one is faster than the other'', but not '' the convergence rate is faster'', which you should probably prove a lower bound to claim.

Moreover, despite the well written and well motivated introduction, the technical part of this paper is poorly organized and hard to follow by  a non-expert. Many important yet basic setting is not introduced (MLE, MAP, and the online model). I realized that this paper adopts an online-setting model and care about regret or the first time only at the end of the page 6! Many fancy models, assumptions and priors are introduced without justification or references. People with no expertise on MLE or MAP may not understand what the algorithm is doing. (I guess it is updating the confidence interval B?) Given the problem itself is important and highly non-trivial, I think it is more important to let the audience to understand the basic notions from the beginning, which the more technical parts (such as Dietrich Prior) should go to the Appendix.

Minors:
L110: t is not defined
L231: Graph prior is not referred.
L258: B^{u,a,v} is not defined but used multiple times throughout the paper.
L1197: I don't think the definition of non-dictatorship for (27) is correct. The ($\forall$ x, x') and ($\not\exists$ $i$) should be swapped. There also should be ''for any utility profile''.

-----------------
Update: Theorem 3.3 and 3.12 are addressed. Theorem 3.9 is beyond my expertise now. I raise the score to 5 and decrease my confidence to 2.

**Questions:**

***1. Please defend yourself on the soundness of the theoretical results. Please address every point I proposed for all three theorems, justify your proof is correct or can be made correct with reasonable assumptions. If the soundness can be justified or amended, I am happy to raise my score.

2. In the algorithm Line 4, you maximize the acquisition function by finding the pair of candidates where agents overall have strongest preferences on one to the other. What is the intuition behind this? Firstly, this sound like greedily finds the local maximum in every step, which usually does not find the global maximum (unless the function is convex). Secondly, if we want to elicit as much information in each step, we should select the most informative pairs. I tend to believe that the pairs are close to each other are more informative, because revealing the difference between then help us better understand the utility functions.

3. For Theorem 3.9, why there is an $n$ factor in the regret of oracle? If we know the adjacency matrix A and its inversion, elicit public votes by an oracle should be the same as elicit private votes by "None", and the regret should also be the same.

4. You study an online setting and consider about regret. Will the offline setting make a difference? For example, if we can elicit a full ranking from each agent, will things change?

---

> ### Author Response · Authors · 2024-11-21
> **Responses to the review (1/2)**
>
> Thank you for carefully reading our submission and positive feedback. The following are our responses and the changes are highlighted in blue in the updated manuscript (pdf). Please refer to the modified draft.
>
> # Concern on dictatorship and impossibility theorem
> We thank the reviewer for their detailed analysis of the impossibility result. After thoughtful consideration, we agree with the comments on the binary decision case, the definition of dictatorship and the restrictiveness in practice of the counter-example that considers distinct favourite candidates for each agent. The constructive criticism motivated us to reconsider the proof and made us realize that the impossibility of groupthink-proofness does not require considering dictatorship. The new proof relies on the absence of a trivial social consensus, i.e., all agents should not have the same (trivial) utility-maximizing candidate. The proof and relevant impossibility result sections have been updated in the appendix and the main draft respectively.
>
> # Concern on Theorem 3.9 Lemma F.3
> Yes, your high-level concern is true (although your example does not serve as a counterexample since $u(x^\prime)$ is negative, causing the inequality to flip, we guess typo?). Upon further reflection, we also realized that Lemma F.3 is unnecessary. Instead, we used the properties of generalised Gini social-welfare functions, such as Hardy-Littlewood-Polya inequality, Schur convexity, and additivity. Please refer to the Lemma F.4. for the new proof. This modification leads to a similar, though slightly larger, constant: $\mathcal{L}_\mathcal{A} = \sqrt{n} \lVert w \rVert$. The other elements, including the intuition regarding the aggregation function (where the egalitarian approach is the slowest and the utilitarian approach is the fastest) and the sublinear convergence rate, remain unchanged.
>
> # Concern on Theorem 3.12
> We agree that worst-case analysis alone may not fully capture evidence of faster convergence. However, we would like to emphasize that focusing on worst-case scenarios is a common practice in the Bayesian optimization community and is not specific to our work. While upper bounds provide insights into worst-case performance, lower bounds similarly reflect only the best-case scenario. In practice, analyzing the average case could indeed offer valuable insights into expected performance, but it is generally difficult to analyze and therefore rarely included.
> Factors such as multimodality and smoothness can also influence average convergence rates in black-box optimization tasks, which is why empirical evaluations are essential for assessing practical performance (see Figures 3 and 4). Our focus on worst-case scenarios is intended to ensure the algorithm’s robustness and its ability to converge to the global optimum under challenging conditions. We also acknowledge that our original explanation may have overstated the implications of worst-case analysis. To address this, we have revised the manuscript to explicitly clarify that our results are applicable in the worst case for greater accuracy and transparency.
>
> # Concern on draft structure
> We agree that additional background information would benefit a broader readership; however, we were limited by space constraints. Given that the primary audience we assumed is the Bayesian optimization community, we assumed familiarity with foundational concepts. Including extensive background would necessitate reducing our focus on novel contributions in favor of revisiting established concepts. With our paper already spanning 50 pages, including the Appendix, it is challenging to cover all relevant foundational material, especially as our work bridges the fields of welfare economics and Bayesian optimization, where readers may have expertise in one but not the other. We prioritized discussing the novel aspects for the BO community. The primary reason for adopting this fancy model is that it is the only one allowing for theoretical analysis, as outlined in lines 199-205. For a more intuitive and accessible approach, we also include a popular Gaussian process model in Appendix H.1, which was our initial model. However, we prioritized theoretical insight in this new setting to enable readers to extend the work to different scenarios or practical applications. Despite the model's complexity, we believe Theorem 3.9 offers intuitive and reasonable bounds for understanding this problem. Additionally, we referenced key papers that introduced foundational concepts as a compromise.

---

> ### Author Response · Authors · 2024-11-21
> **Responses to the review (2/2)**
>
> # Intuition on acquisition function
> Firstly, the acquisition function maximization problem itself is a global optimization problem, but with a known explicit functional form. Therefore, it does not encounter the issue of local optima. Secondly, you are correct that we aim to elicit as much information as possible at each step. Similar to standard Bayesian optimization with continuous feedback, there is an exploitation-exploration tradeoff. Exploitation (also known as pessimism) involves relying on the current model to seek its maximization, while exploration (also known as optimism) involves questioning the current model and seeking to maximize information for future iterations.
> At each step, we select the pair that optimizes the preference based on the current data (exploitation) while simultaneously encouraging a certain level of exploration. This entails optimizing both $x$ over the input space and $u$ over the function confidence set. Intuitively, this approach optimistically explores those $x$ and functions $u$ that have the potential to yield favorable results, employing the popular principle of optimism in the face of uncertainty.
>
> # Intuition on additional $n$ term on oracle in Theorem 3.9
> This is because, even with the given graph $A^{-1}$, it amplifies the noise of $v$, whereas pure private votes do not amplify the noise of $u$. We assumed that both $u$ and $v$ yield the same level of estimation error given the same number of votes (Lemma 3.6 and range preservation, Lemma 2.7). Thus, they should exhibit the same level of noise at the same timestep $t$. However, since $v$ is an indirect observation of $u$, the inversion $\lVert A^{-1} \rVert \leq n$ (Lemma 2.7) introduces inevitable amplification. As a result, the presence of $n$ in the oracle case is unavoidable in the worst-case scenario.
>
> # Extention to offline case
> Yes, the offline setting will slightly modify the algorithm. If we can obtain a full ranking from each agent, the estimated utility functions remain probabilistic models, where we typically balance exploitation and exploration. However, in the offline setting, where further queries are not possible, optimism no longer applies. Instead, it becomes a pure exploitation problem, specifically maximizing the MAP estimate: $x_\text{final} = \arg\max_{x \in \mathcal{X}} \mathcal{A}[\hat{u}(x,:)] - \mathcal{A}[\hat{u}(x^\prime,:)]$. This approach differs from the acquisition function in Eq. (5), which provides an optimistic estimate by incorporating both exploitation and exploration (i.e., taking the upper bound). In contrast, the offline setting relies solely on exploitation.
>
> Thanks to your feedback, we have improved our manuscript by refining the proofs and enhancing the clarity of our algorithm and theoretical results. These updates have been incorporated into the manuscript, highlighted in blue text, and we believe your concerns have now been addressed.

---

> ### Comment · Reviewer_DBM8 · 2024-11-21
> **An additional question**
>
> Thank you for your response! I find them very interesting. Could you please give me a basic idea of what your algorithm is doing? You can assume I have little knowledge of Bayesian estimation, MLE, and MAP beyond the basic notion (so I cannot understand your section 3.3). Your explanation will help me go over your new proof. Thanks!
>
> For example, is $x_t'$ in Algorithm 1 actually $x_{t-1}$, as it is not maximized on and there is no $x_{t-1}$ in the algorithm?
> Also, in  Line 1948, how does you maximizing on $\tilde{u}$ also applies to $u$?

---

> > ### Author Response · Authors · 2024-11-21
> > **Thanks for the question**
> >
> > Thank you for your quick reply and encouraging comments! We will try our best to provide the intuition of standard Bayesian optimization (UCB) and explain how our methods extend this to the preference feedback case.
> >
> > ## Standard Bayesian optimization (UCB)
> > Firstly, our aim is to find the global optimum $x^* \in \arg\max_{x \in \mathcal{X}} S(x)$. For the sake of intuition, let’s consider the case where we can directly observe the utility value as continuous and our domain is discrete, $|\mathcal{X}| < \infty$.
> > The simplest benchmark method is to evaluate all samples in the domain, and then select the largest $S(x)$ among them. While this is a global optimizer, its sample complexity is the worst ($|Q_t^u| = |\mathcal{X}|$). For better sample efficiency, we need a strategy to exclude subsets of the domain that are unlikely to contain the global maximum. If such logic exists, we can narrow down the promising regions and avoid exhaustive exploration of the entire domain.
> >
> > Thus, we use a model-based approach. If our model $\hat{S}$ can predict S accurately over the domain, our task becomes simple—as we can precisely predict $\max_x \hat{S}(x)$ without querying agents, thereby eliminating the need for online learning. However, in practice, the estimation $\hat{S}$ becomes asymptotically accurate as more data are observed. Let the observation dataset be $(X, S(X))$, where $X = (x_i)_{i=1}^t$. At timestep $t$, we have $t$ data points, and the (worst-case) accuracy of the model monotonically improves over time. So, how can we leverage such an asymptotically accurate model to efficiently explore the global maximum?
> >
> > The celebrated Upper Confidence Bound (UCB) algorithm provides a solution: maximize the upper bound of $\hat{S}$ at every timestep $t$. In kernel models, we can derive the upper and lower bounds of the estimation error for each prediction point. Let $\text{lower}(\hat{S}(x)) \leq \hat{S}(x) \leq \text{upper}(\hat{S}(x))$ denote the confidence bounds of the model prediction. Kernel models ensure these confidence bounds contain the true function $S(x)$ with high probability. Moreover, the confidence interval becomes tighter over time due to asymptotic convergence.
> > For intuition, decompose the upper bound as $\text{upper}(\hat{S}(x)) = \hat{S}(x) + \Delta_S(x)$, where $\Delta_S(x)$ represents the difference between the upper bound and the MAP estimate at given $x$. Thanks to asymptotic convergence, $\Delta_S(x)$ becomes asymptotically zero as time progresses, and the MAP estimate converges to the true utility function $S(x)$. Thus, the UCB policy guarantees finding the global optimum asymptotically. This method is known to have nearly rate-optimal sample complexity for finding the global optimum.
> >
> > The $\Delta_S(x)$ term is often referred to as exploration, as selecting uncertain points improves model accuracy efficiently. Conversely, $\hat{S}(x)$ represents the model prediction and is called exploitation. The UCB policy naturally balances the tradeoff between exploration and exploitation, exploring uncertain points when samples are few and exploiting the model when it becomes confident.
> >
> > ## Our Algorithm 1
> > Our Algorithm 1 includes slight modifications of UCB to fit our setting. First, our feedback is indirect; we estimate the true utility function from pairwise feedback. Instead of $S(x)$, we infer the difference $S(x) - S(x')$. We establish a similar confidence bound as in Lemma 3.6, ensuring that the estimated confidence interval $\hat{S}(x) - \hat{S}(x')$ contains the true difference $S(x) - S(x')$ with high probability $(1 - \delta)$. Subsequently, our UCB-like acquisition function maximizes its upper bound $\text{upper}[\hat{S}(x) - \hat{S}(x')]$. Here, we set $x'$ as $x_{t-1}$, the point selected in the previous timestep. This choice implies that the difference $S(x_t) - S(x_{t-1})$ implicitly maximizes $S(x_t)$, as $S(x_t) > S(x_{t-1})$ by the nature of the upper bound. Thus, maximizing the upper bound of the difference asymptotically converges to the true global maximum.
> >
> > The second point is the stopping criterion. The projection weight function $w^u_t$ is the confidence interval of the difference: $\sup |(\hat{S}(x) - \hat{S}(x')) - (\hat{S}'(x) - \hat{S}'(x'))|$. This is "$(\text{Upper bound} - \text{Lower bound})$", namely confidence interval. If the confidence interval of $u$, the truthful utility estimate, is larger than that of $v$, the non-truthful utility estimate, further exploration of $u$ is required. This can be viewed as a tradeoff between exploitation and exploration in the context of private-public voting uncertainty.
> > While the social-influence graph component may appear complex, it ultimately involves comparing confidence intervals at given points $x$. When exploration (information gain for the graph) converges, the confidence interval tightens, as illustrated in Fig. 2. At this point, we can stop private voting asymptotically.
> >
> > We hope this explanation clarifies our algorithm.

---

> > > ### Author Response · Authors · 2024-11-21
> > > **For additional questions**
> > >
> > > For the additional questions:
> > >
> > > 1. Yes, $x^\prime_t = x_{t-1}$: We have clarified this point more explicitly on Line 4 in Algorithm in the updated manuscript (pdf).
> > >
> > > 2. Line 1948 explanation: This line follows from the confidence bound for the global maximum point:
> > >
> > > $\min_{x} \min_{\tilde{u}} \tilde{u}(x) \leq u(x^*) \leq \max_x \max_{\tilde{u}} \tilde{u}(x)$.
> > >
> > > (Due to the issues with markdown for displaying sub-subscript,
> > > we use $\tilde{u}$ as $\tilde{u}_t$ here).
> > >
> > > This can be interpreted as a "doubly optimistic" confidence bound.
> > > Recall that our acquisition function is the maximum (upper bound) of the difference:
> > >
> > > $x_t = \max_x \max_{\tilde{u}} [\mathcal{A}(\tilde{u}(x,:)) - \mathcal{A}(\tilde{u}(x_{t-1},:))]$.
> > >
> > > Therefore, $\mathcal{A}(\tilde{u}(x_t,:)) - \mathcal{A}(\tilde{u}(x_{t-1},:))$ represents the maximum value of this difference within the confidence set.
> > > We can interpret the first term, $\mathcal{A}(\tilde{u}(x_t,:))$, as the upper bound of the social utility function estimate at $x_t$ and the second term, $\mathcal{A}(\tilde{u}(x_{t-1},:))$, as the lower bound of the social utility function estimate at $x_{t-1}$. The difference, therefore, becomes maximal. Or we can decompose individually $\mathcal{A}(u(x^*,:)) \leq \mathcal{A}(\tilde{u}(x_t,:))$ and $\mathcal{A}(u(x_{t-1},:)) \geq \mathcal{A}(\tilde{u}(x_{t-1},:))$. Consequently, the difference is guaranteed to be larger.

---

> > > > ### Author Response · Authors · 2024-11-27
> > > > **Any further questions or suggestions?**
> > > >
> > > > Thank you for investing your time in reviewing and discussing our paper. We hope our responses have adequately addressed your concern regarding the soundness of our theoretical results. Should you have any further questions or require additional clarification, we would be delighted to assist.
> > > >
> > > > From our perspective, understanding proofs from a different field can be challenging, as it often requires familiarity with a significant amount of mathematical knowledge specific to that community. Given the constraints of a limited discussion period and busy schedules, we recognize that this may not always be straightforward. If the high-level flow of our argument seems reasonable, we would greatly value your evaluation based on your expertise in welfare economics, as our intended primary audience is the Bayesian optimization community, as indicated by the title of our paper.
> > > >
> > > > For your reference, our theoretical analysis builds on the ICML 2024 oral paper, "Principled Preferential Bayesian Optimization". Our work generalises this framework to address social welfare functions under multiple agents with non-truthful votes. While the generalisation is non-trivial, the underlying theoretical basis remains robust and rigorous.

---

> > > > > ### Comment · Reviewer_DBM8 · 2024-11-28
> > > > >
> > > > > Thank you for your understanding and reply. I really appreciate your effort to improve the paper. As I am not an expert in Bayesian Optimization, examining the proof in detail within this time constraint is beyond my ability. Therefore, I will stay neutral on Theorem 3.9. On the other hand, Theorem 3.3 and Theorem 3.12 are addressed in one way or another. I will raise my score to five and decrease my confidence accordingly.
> > > > >
> > > > > The reason for scoring 5 rather than higher is that I still feel the paper needs major revision in the presentation. I understand the author's concern about prioritizing the expert readers, and I also find the explanation in the rebuttal (and follow-up comments) helpful. However, I still reserve my opinion that the current version is too technical and unfriendly to outsiders. For example, I somewhat understand your comments on what is UCB and how you modify UCB to your setting, but I cannot connect it with the algorithm in your paper, because I don't know how your MAP is implemented. In my opinion, it is better to leave the main paper to the very core idea, basic settings to educate the readers, and running examples and put all the fancy techniques into the Appendix. (I am not saying that these fancy notions are not good. On the contrary, they demonstrate the theoretical contribution of this paper. I just can't understand them without further explanation.)

---

> > > > > > ### Author Response · Authors · 2024-11-28
> > > > > >
> > > > > > Thank you for your thoughtful response and for raising the score. However, we respectfully disagree with the suggestion that the primary issue lies in the presentation. In our view, the core challenge stems from a mismatch between the reviewers’ expertise and our paper's focus, rather than the presentation itself. While this situation is unfortunate for both reviewers and authors, we deeply appreciate your constructive advice.
> > > > > >
> > > > > > Readability, unlike technical soundness, is inherently subjective and depends heavily on the reader's familiarity with the topic. Optimization, as a mathematical discipline, is naturally challenging for those outside the field to follow. As with any specialized mathematical subject, such as topology, it is feasible to explain basic concepts but nearly impossible to elevate a non-expert to the forefront of theoretical topology research within the constraints of a 10-page paper. Attempting to overemphasize introductory explanations often results in insufficient space to articulate novel contributions, leading to rejection by expert reviewers—a challenge we have frequently encountered in the Bayesian optimization community. Consequently, our community seldom elaborates on fundamental concepts such as UCB, acquisition functions, regret, or maximum information gain. As Arrow famously demonstrated, it is impossible to satisfy all preferences through a single aggregation. Similarly, if we were to rewrite this paper, the outcome would differ from your suggestions. We would write more explicitly to the Bayesian optimization audience, minimizing the risk of review by experts from other domains. We suspect this mismatch arose from our citations and content triggering certain classifications in ICLR's matching system. To address this, we would move many economics-related elements, such as impossibility theorem 3.3, to the appendix and reduce citations from economics literature. While this is sad, these aspects primarily serve as just motivation and can be omitted to minimize the risk of future mismatches.
> > > > > >
> > > > > > This paper was primarily aiming to establish a theoretical foundation for this emerging topic. The next paper will focus more on practical applications and user-friendly explanations. We have already anonymously open-sourced the algorithm and refactored the code to facilitate its release as a Python library. Upon acceptance, we will provide detailed tutorials, including Jupyter notebooks, to enhance accessibility and offer hands-on understanding of the algorithm. In practice, we believe that engaging with the coding tutorial will offer users, particularly those less familiar with Bayesian optimization, a deeper understanding of our algorithm than reading the paper alone. Our expectation was for economists to assess the paper’s usefulness and for optimization experts to evaluate its technical soundness. Unfortunately, this balance was not achieved in the current reviewers' batch. This is neither our fault nor the reviewers', but we hope you will consider this biased situation during your assessment.
> > > > > >
> > > > > > Could you reconsider the score if you believe our contribution is excellent and our paper will benefit the ICLR community, rather than encouraging us to reshape it into an optimization-only focus solely for acceptance?

---

> > > > > > > ### Author Response · Authors · 2024-12-02
> > > > > > > **Any further questions or suggestions?**
> > > > > > >
> > > > > > > As the discussion period comes to an end, we kindly ask if you could update your score if our discussion has adequately addressed your concerns. Thank you again for your valuable feedback, particularly on improving the clarity of Theorems 3.3 and 3.9.
> > > > > > >
> > > > > > > ## In short
> > > > > > > We believe readability is difficult to assess without familiarity with the main topic, as it can be challenging to determine whether issues arise from the text itself or from a lack of background knowledge due to a mismatch between the readers’ expertise and the intended audience, particularly for theoretical papers. It should be clear from the title and abstract that this paper is intended for Bayesian optimization researchers.
> > > > > > >
> > > > > > > ## Formal answer
> > > > > > > We believe the main issue lies not in readability but in a mismatch between the reviewers’ expertise and the paper's focus, stemming from the ICLR matching system. Given the title and primary area specification (Bayesian Methods), we did not anticipate that all reviewers would be from welfare economics with little familiarity with Bayesian optimization, who are not our intended audience. Our paper is primarily designed for experts in (theoretical) Bayesian optimization, and we believe it is straightforward for such expert readers to follow, as it is a generalized version of preferential Bayesian optimization, a popular framework in our field, to the heterogeneous multi-agent setting under social influence.
> > > > > > >
> > > > > > > For non-experts, we have referred to foundational papers and books and have anonymously open-sourced our code to facilitate deeper understanding. Upon acceptance, we plan to add coding tutorials, as we believe practical exercises are more effective for non-experts to understand the algorithm than relying solely on the paper—reflecting typical teaching practices in graduate school for studying new topics. The next paper and future work will focus more on applications, making it more accessible to welfare economists. However, this paper aims to establish a theoretical foundation for this complex topic, bridging Bayesian optimization researchers with welfare economics to facilitate meaningful contributions to this important field for theoretical Bayesian optimization researchers, not for welfare economists.
> > > > > > >
> > > > > > > As you highlighted, our theoretical contribution is non-trivial. Existing game-theoretic approaches to multi-agent preference aggregation face significant limitations, such as the absence of equilibrium guarantees in continuous domains and the non-submodularity of finding equilibria, which impede global optimization guarantees. Our generalized Gini social-evaluation welfare function addresses these challenges while preserving fairness principles, including the Pigou-Dalton principle. Notably, our paper is *first-of-its-kind* to provide sublinear convergence rates for cumulative regret bounds and sample complexity in settings involving heterogeneous agents across both discrete and continuous domains. Additionally, our graph convolution model offers a robust theoretical framework for studying social influence and supports straightforward extensions to more complex settings, as demonstrated in our response to reviewer KuC6. We believe our paper establishes a strong theoretical foundation and contributes meaningfully to the ICLR community.
> > > > > > >
> > > > > > > *We would greatly appreciate it if you could update your score to support acceptance, particularly if you believe that our contribution is non-trivial and provides meaningful benefits to the ICLR community (soundness: 3, presentation: 3, contribution: 4).*

---

> > > > > > > > ### Comment · Reviewer_DBM8 · 2024-12-02
> > > > > > > >
> > > > > > > > Thank you for your detailed reply. I respect your opinion that there is possibly a mismatch, so I will put forward this issue (whether it is a presentation problem or a mismatch problem) in the reviewers' discussion.

---

> > > > > > > > > ### Author Response · Authors · 2024-12-02
> > > > > > > > > **Readability should not be a criterion in this special situation.**
> > > > > > > > >
> > > > > > > > > Thank you for your response! However, we kindly ask you to consider the following: none of the reviewers in this batch are experts in Bayesian optimization, as clarified during the discussion. Consequently, our paper has been reviewed solely by non-experts, which we believe makes it particularly challenging to accurately assess readability (and we hope you can appreciate the difficulty and unfairness of this situation from an author’s perspective). Given this context, we feel that discussions with non-experts may not be sufficient to resolve these concerns. Therefore, we respectfully request that readability issues be disregarded in your evaluation, as we believe they cannot be effectively judged under these special circumstances.

---

### Official Review · Reviewer_kuC6 · 2024-10-30

**Soundness:** 2
**Presentation:** 3
**Contribution:** 2
**Rating:** 5
**Confidence:** 3

**Summary:**

This paper introduces Social Bayesian Optimization (SBO), an innovative algorithm designed to build consensus in group decision-making by mitigating social biases that often distort individual preferences. Traditional consensus-building approaches struggle under social dynamics like the bandwagon effect, which leads to non-truthful feedback. SBO overcomes this by combining two types of votes: noisy but cheap public votes and private, costly votes, free from social influence. The algorithm leverages a social graph model to efficiently estimate social influence, allowing it to correct public votes and reduce the reliance on costly private feedback.

**Strengths:**

1. The paper introduces a voting system in Social Bayesian Optimization (SBO) that effectively separates noisy public votes from more reliable private feedback. This approach allows the algorithm to debias public responses, which is particularly impactful in social contexts where group dynamics often distort true preferences.

2. The authors provide a well-structured theoretical framework, including a proof that achieving groupthink-proof consensus is impossible without the dual voting mechanism.

**Weaknesses:**

1.  The algorithm’s performance heavily depends on accurately modeling social influence through a predefined social graph, which may not reflect the complexity of real-world interactions.

2. While the dual voting system improves consensus accuracy, it also increases computational and operational costs due to the need for costly private votes.

3. The paper assumes certain types of social influence, like bandwagon effects, but does not account for other common group dynamics (e.g., peer pressure or hierarchical influence).

4. The model assumes agents have consistent and rational preferences, which may not align with the variability found in real-world settings.

5. The algorithm depends on several parameters, such as the decay rate and aggregation weight, but the paper lacks a thorough sensitivity analysis on how variations in these parameters affect performance.

**Questions:**

1.  The algorithm’s accuracy relies on estimating the social influence graph effectively. How do estimation errors in the social graph affect the convergence rate and regret bounds of SBO? Can the authors quantify a bound on the cumulative error introduced by inaccurate social graph estimation?

2. The paper models social influence as a linear function within the social graph. How would the algorithm’s convergence and regret bounds be affected if the influence model were non-linear? Is there a theoretical guarantee that SBO can handle non-linear influence functions without compromising efficiency?

3. The paper provides sublinear regret bounds for SBO. However, given the dual voting system’s reliance on private votes to debias public ones, can the regret bounds be formally decomposed to show how each voting system independently contributes to the cumulative regret? How would an imbalance in the quantity of public versus private votes affect these bounds?

4. The algorithm assumes that agents’ preferences are rational and consistent. If agent feedback were inconsistent or stochastic, how would this impact the estimation of utilities and the associated convergence rate? Could the authors provide an error bound quantifying the effect of inconsistent feedback on the final consensus?

5.  SBO’s performance depends on parameters like the decay rate and the aggregation function’s weight. Can the authors derive optimal parameter values that balance the trade-off between convergence speed and the accuracy of social influence estimation? What are the theoretical limits on these parameters for guaranteed sublinear convergence?

---

> ### Author Response · Authors · 2024-11-21
> **Responses to the review (1/2)**
>
> Thank you for reading our submission and positive feedback. The following are our responses and **the changes are highlighted in red** in the updated manuscript (pdf). Please refer to the modified draft.
>
> # On graph estimation error
> Firstly, our bounds on cumulative regret and query complexity are derived for the worst-case scenario. This means that if the actual estimation error is better than the worst case, both bounds could be tighter. We can interpret the decay rate $q$ as quantifying the social graph estimation error: a larger $q$ requires more private votes to reduce the estimation error sufficiently to meet the stopping criterion in Line 7 of Algorithm 1. According to Theorem 3.12, the worst-case convergence rate of the graph estimation error is $-1/2$, so we set $q=1/2$ (see line 370). If the estimation error converges faster than $-1/2$, it suggests that an even smaller $q$ could be used, yielding a tighter sample complexity for private votes, $\lvert Q_t^u \rvert$. Because our bound is based on the worst-case scenario, a smaller $q$ results in a looser regret bound. However, if the actual estimation error rate is better than $-1/2$, the regret bound should not heavily depend on the $q$-dependent term. Since we lack access to the ground-truth graph $A$, we cannot dynamically estimate the actual error convergence rate on the fly. Therefore, in practice, we recommend setting $q=1/2$ based on the worst-case rate in Theorem 3.12.
>
> # On other graph models (non-linear, peer-pressure, and hierarchical influence)
> For non-linear cases, we can employ a graph convolutional kernel network approach [1]. Using the kernel trick and Nyström method, this approach can transform non-linear functions into effectively linear forms. Popular graph neural network models can also be seen as special cases of [1]. Once the model is linearized, our approach can be applied directly. The linearized graph has $m$ components, where $m$ represents the number of Nyström model centroids. By applying the Cauchy-Schwarz inequality, our bounds in Theorems 3.9 and 3.12 become looser by a factor of $\sqrt{m}$. However, since this is a constant, the order of the asymptotic convergence rate remains the same at $-1/2$. The Nyström method provides an eigendecomposition-based approximation of the non-linear network, where $m$ reflects the complexity of the non-linear social graph. This adjustment is reasonable, as a more complex ground-truth social graph would naturally lead to a slower convergence.
>
> Regarding the other models you mentioned, our approach is adaptable. For instance, in a peer-pressure scenario, we assume a setting where a minority of agents may shift their votes toward the majority. This minority or majority could vary based on the option $x$, creating a heterogeneous setting. If the setting is homogeneous, our algorithm can model peer pressure directly. For a heterogeneous setting, as noted in L535-536, we can incorporate diversity using a probabilistic choice function (Benavoli et al., 2023b). While regret bounds for this model are not yet established in the literature—given that even linear graphs are novel in this context—the submodularity of the probabilistic choice function suggests sublinear convergence.
>
> For hierarchical influence, we are unsure of its relevance in settings where all participants vote in the same room. This scenario may arise when influence propagates slowly among voters, as in presidential voting. Our target tasks, however, are in an online setting where voting is iterative and occurs at a relatively faster pace than in presidential elections (see Section 5). Still, if it does occur, a grey-box Bayesian optimization approach [2] could be applied, where the hierarchical graph structure is known but the specific attributions remain unknown. Under these assumptions, we can demonstrate the same convergence rate as in Theorem 3.9 with high probability, although this analysis is beyond the scope of the current paper.
>
> Lastly, as noted by all reviewers, our paper is the first-of-its-kind to explore group consensus-building under social influence. We believe it is reasonable to start with the simplest setting, where the graph is linear. Linear graphs are widely studied and applied in the literature, particularly in social network modeling (e.g., [3,4,5,6]), supporting their practical effectiveness. As Reviewer DMB8 highlighted, even with this basic linear setting, our contribution remains substantial, with the proof spanning 50 pages. We argue that this choice does not compromise the completeness of our paper and have demonstrated that our algorithm is extendable.
>
> ## Citations
> - [1] Chen, D. et al., "Convolutional Kernel Networks for Graph-Structured Data", ICML (2020)
> - [2] Xu, W., et al., "Bayesian Optimization of Expensive Nested Grey-Box Functions", arXiv:2306.05150 (2023)
> - [3] Ding, Yuxin, et al. Computer Communications 73 (2016): 3-11.

---

> ### Author Response · Authors · 2024-11-21
> **Responses to the review (2/2)**
>
> - [4] Li, Yanying, et al. "Fairlp: Towards fair link prediction on social network graphs." AAAI (2022)
> - [5] Liang, H. et al., "Bayesian Optimization of Functions over Node Subsets in Graphs." NeurIPS (2024).
> - [6] Wan, X. et al., "Bayesian optimisation of functions on graphs." NeurIPS (2023)
>
> # On contribution of each voting system
> In Appendix F.2.2, line 2065, we present a decomposition of the regret bound to show how each voting system independently contributes to cumulative regret. An imbalance between the number of public votes ($T$) and private votes ($\lvert Q^u_t \rvert$) affects two terms: $(T - \lvert Q \rvert)$ and the maximum information gains $\gamma^{uu^\prime}_T$ and $\gamma^{vv^\prime}_T$, both of which depend on the total number of votes (see details of this dependency for each kernel in F.3).
>
> Instead of focusing solely on imbalance, we can intuitively understand this relationship by considering $\lvert Q^u_t \rvert$ alone. When $\lvert Q^u_t \rvert = T$, the second and third terms become zero $(T - \lvert Q^u_t \rvert = 0)$, and the regret bound relies solely on private votes. Reducing $\lvert Q^u_t \rvert$ introduces an additional term in the regret, meaning that fewer private votes $\lvert Q^u_t \rvert$ lead to higher regret. Therefore, smaller values of $\lvert Q^u_t \rvert$—that is, greater imbalance—result in increased regret.
>
> # On inconsistent feedback
> To clarify, while feedback follows a Bernoulli distribution and is therefore stochastic (Assumption 2.2), the ground-truth utility function itself is deterministic (Assumption 2.6). We must estimate the ground-truth utility function from stochastic feedback, making our estimated model probabilistic, with its worst-case error bounded by a confidence set (Lemma 3.6). Feedback becomes nearly random when the utility difference $u(x,i) - u(x^\prime,i)$ is small (Assumption 2.2) but is less noisy when this difference is large. If this feedback noisiness aligns with your notion of "inconsistency," our model can indeed handle such inconsistent feedback. Our regret bound is derived based on the worst-case utility function estimation error in Lemma 3.6, where the error bounds, represented by $\beta^u_T$ and $\beta^v_T$, appear as constants in Theorem 3.9. Thus, less noisy feedback corresponds to a smaller $\beta$, resulting in a tighter regret bound.
>
> If, however, your notion of inconsistency refers to something else, such as transitivity or completeness, please note that these are fundamental assumptions in both social choice theory and preference-based Bayesian optimization, as employed in all nine papers accepted at top ML conferences listed in lines 387-391. These assumptions are standard and widely applied in real-world settings.
>
> # On optimization parameters
> To clarify, while the decay rate is a hyperparameter that users of our algorithm can control, the weight of the aggregation function is uncontrollable, as it is part of the problem definition we aim to optimize. The effect of this weight is discussed in lines 356-357, where the egalitarian aggregation has the slowest rate ($L_A \rightarrow \sqrt{n}$) and the utilitarian aggregation the fastest ($L_A = 1$).
> For the decay rate $q$, line 370 already states that $q = 1/2$ can be optimal, though we unpack more on this point for clarity. Generally, $q$ controls the trade-off between cumulative regret $R_T$ and the number of queries $\lvert Q_t^u \rvert$, with optimality depending on the user’s query budget. This can be viewed as the task of minimizing the cost function $C_T := \lambda_u \lvert Q_t^u \rvert + \lambda_v T$ (see line 323), where $\lambda_u$ and $\lambda_v$ represent the costs of private and public votes, respectively. Under Assumption 3.4, we assume $\lambda_v \gg \lambda_u$, meaning faster convergence in $\lvert Q_t^u \rvert$ is preferable. Recall that private votes are queried to estimate the graph $A$, with a convergence rate of $\mathcal{O}(\lvert Q^u_t \rvert^{-1/2})$ (Theorem 3.12). Additionally, $\lvert Q^u_t \rvert$ is bounded by the condition in line 7 of Algorithm 1 ($1/t^q$). Setting $q=1/2$ (i.e., $t^{-1/2}$) ensures that the cumulative error $\sum_{i\in[T]} t^{-1/2} \leq \mathcal{O}(T^{1/2})$ aligns with the convergence rate of graph estimation error, making it optimal under a high-cost scenario $\lambda_v \gg \lambda_u$. The theoretical limit on $q$ is further discussed in Appendix F.2.2, lines 2099-2108, and in Theorem 3.9, line 339.
>
> Thanks to your feedback, we could improve our manuscript by enriching the discussion on the extendability and flexibility of our algorithm, as well as clarifying the assumptions and theoretical framework. We have incorporated these updates into the manuscript, highlighted in red text, and believe that your concerns have now been addressed.

---

> > ### Author Response · Authors · 2024-11-27
> > **Any further questions or suggestions?**
> >
> > Thank you for investing your time in reviewing and discussing our paper. We hope our responses have adequately addressed your concern regarding the soundness of our theoretical results. Should you have any further questions or require additional clarification, we would be delighted to assist.

---

### Official Review · Reviewer_Bj6T · 2024-11-05

**Soundness:** 2
**Presentation:** 3
**Contribution:** 2
**Rating:** 6
**Confidence:** 4

**Summary:**

This paper considers the Social Bayesian Optimization (SBO) for consensus-building, which is to estimate n people's utility functions from their votes. The paper proves an axiomatic impossibility result to have both groupthink-proofness and non-dictatorship. The paper then describes its proposed mechanism that iteratively finds the better social outcomes.

**Strengths:**

- This is my first time reading literature on this topic and the paper does a fairly good job in explaining key notions, challenges and techniques in this problem.
- The paper includes several experiments under the real world context.

**Weaknesses:**

- The paper introduces several assumptions that are perhaps too strong to be realistic. For example, Assumption 3.4 requires private votes to reflect the truthful utility, but it is unclear that such truthful utility exists and can be elicited easily. If such private votes exist, why not use as many private votes as possible?
- The paper lacks sufficient justification on the limitations of their proposed mechanisms.

**Questions:**

I have some research experiences in mechanism design, social choice theory and statistical learning theory, but I have little knowledge in this BO, especially in its application to preference aggregation. Hence, I would like to understand how the preferential bayesian optimization techniques have been applied. Does it have any real world application? How many samples does it require to work under the typical noise condition in practice? --- for example, it is not quite realistic to elicit human preference with >10 samples, your proposed mechanism could take too long to run.

---

> ### Author Response · Authors · 2024-11-21
> **Responses to the review**
>
> Thank you for reading our submission and positive feedback. The following are our responses and **the changes are highlighted in cyan** in the updated manuscript (pdf). Please refer to the modified draft.
>
> # Our assumption is actually weaker than the existing literature.
> Your concern made us realize that the term "truthful" can be misleading. In this work, what we meant by "truthful" is social-influence-free. While private voting can guarantee social-influence-free utilities, the reports can still be non-truthful in the sense that agents can act strategically and misreport their votes. To address this, we have replaced the term "truthful" with "social-influence-free" and the term "non-truthful" with "influenced" throughout the manuscript, including in the title.
>
> However, we also want to clarify that our assumptions are, in fact, weaker than those in existing literature. For example, preferential Bayesian optimization (referenced in nine papers accepted at top ML conferences, listed in lines 390–394) typically assumes that feedback always reflects truthful utility, an assumption widely applied in real-world settings. In contrast, our work adopts a more realistic approach by accounting for social influence, which may skew votes.
>
> As noted in Assumption 3.4, private votes are social-influence-free but costly, with the cost of querying private votes significantly higher than that of public votes, i.e., $\lambda_v \gg \lambda_u$. Consequently, $\lvert Q_t^u \rvert$ dominates the total cost $C_T := \lambda_u \lvert Q_t^u \rvert + \lambda_v T$, so we aim to minimize $\lvert Q_t^u \rvert$ to reduce costs while still minimizing cumulative regret. Lines 62–65 and Figure 1 illustrate scenarios with high costs. Public votes, such as a show of hands in a meeting, are inexpensive and allow rapid iteration, though they may not be social-influence-free. In contrast, private votes require one-on-one interviews, necessitating that agents meet separately, which is especially costly.
>
> # We have justified the limitations.
> We outlined and justified the limitations of our approach in Section 5 (lines 532-539), which include high-dimensional inputs, batch settings, and heterogeneous or dynamic aggregation functions. For example, techniques such as additive kernels (Kandasamy et al., 2015) can improve scalability in high dimensions (line 358), and batch acquisition is well studied, allowing us to apply standard methods (e.g., via the BoTorch library). These aspects are orthogonal to our primary focus. As recognized by all reviewers, our paper is the first to explore group consensus-building under social influence. We believe it is reasonable to start with the simplest setting, assuming a homogeneous and stationary aggregation function. As noted in lines 535-536, the probabilistic choice function (Benavoli et al., 2023b) can address more complex scenarios, but such an extension would warrant a separate paper—especially considering that our proof in this simplest setting already spans 50 pages.
>
> # Our paper is the first step for preference aggregation in Bayesian Optimization.
> Our paper is the first-of-its-kind to investigate preference aggregation under social influence within a Bayesian optimization (BO) framework, demonstrated through four real-world examples in Section 5 (Figure 4). While preference aggregation is new, preference maximization is a well-studied and widely applied area—with examples in visual design optimization (Koyama et al., 2020), thermal comfort optimization (Abdelrahman \& Miller, 2022), collaborative filtering (Schafer et al., 2007), robotic gait optimization (Li et al., 2021), and autonomous driving policy design (Astudillo et al., 2024)—each of these applications typically iterates over 50 to 500 samples (see also the references in lines 390-394 of Section 4). We are not aware of any learning algorithm that converges in as few as 10 samples. Scarlett et al. (2017) established a lower bound (theoretical limit) on the convergence rate of cumulative regret in BO, $\Omega(\sqrt{T} (\log T)^d)$, which precludes convergence within such a limited sample size. In our experiments, however, our algorithm (yellow) achieves a sufficiently small regret in around 20 samples, with two of the tasks reaching small regret in fewer than 10 samples, suggesting that our approach empirically meets your strong requirement (in comparison to other baselines).
>
> Thanks to your feedback, we could improve our manuscript by clarifying the assumptions and significance of our new settings and the theoretical framework. We have incorporated these updates into the manuscript, highlighted in cyan text, and believe that your concerns have now been addressed.

---

> > ### Comment · Reviewer_Bj6T · 2024-11-21
> >
> > Thanks for your detailed reply and resolves some of my concerns. I would like to keep my rating fixed, given that I do not know any related work in this area, while find the idea of this work somewhat interesting.

---

> > > ### Author Response · Authors · 2024-11-22
> > > **Thanks for the reply**
> > >
> > > Thank you for your positive feedback and for leaning towards acceptance. If further clarification would help improve the rating, we would be more than happy to provide additional explanations.

---

### Official Review · Reviewer_x1x4 · 2024-11-08

**Soundness:** 3
**Presentation:** 2
**Contribution:** 3
**Rating:** 5
**Confidence:** 2

**Summary:**

This paper proposes a consensus-building inspired learning problem.  Specifically, given a graph $G$, each agent $i$ has a true utility over a set of items $x$, $u(x,i)$, and influenced utility $v(x,i)$ which depends on both its true utility as well as it's neighbor's in $G$.  They first show that there does not exist an aggregation function $A$ so that $$argmax A(u,x) = argmax A(v,x)$$ (groupthink-proof) and not dictatorship.  Then they provide a public and private vote setting that can query noisy comparison based on either influenced utility $v$ (public vote) or true utility (private vote) from each agent and design a learning algorithm to bound the error and sample complexity.  Finally, they conduct some experiments.

**Strengths:**

This paper's model is ambitious that try to use learning techniques to address consensus building problem.

**Weaknesses:**

## Unclear contributions
1.  I am not convinced that the social Bayesian optimization, SBO (Theorem 3.9), offers a meaningful solution to consensus building.  The introduction suggests that consensus building is challenging due to social dynamics, and "even with access to each agent's utility, constructing a social utility that fairly represents all agents is impossible under mild rationality constraints." My understanding is that these challenges stem from the discrepancy between influenced utility and true utility—a problem that becomes irrelevant if direct estimation from the true utility is possible. However, their algorithm requires direct access to private votes based on all agents' true utility.  A more convincing justification for SBO would focus on its sample complexity benefits.  For instance, is SBO superior to using private votes alone? To what extent does incorporating public votes enhance the learning process?
2.  The impossibility result in Theorem 3.3 is strange, particularly in its nonstandard definition of dictatorship.  Unlike traditional definitions, where one dictator determines the order of all items, Definition 3.2 allows for different dictators for each item pair. Under this definition, even majority voting or a simple average function might satisfy their dictatorship definition.  This nonstandard definition undermines the impossibility result.
## Convoluted writing
The paper should improve its readability.  The problem formulation remains unclear until Assumption 3.4, making it difficult to follow the main argument initially. Additionally, the paper presents several facts that appear unrelated to the core problem (e.g., Arrow’s impossibility results on line 40 and Proposition 2.4 on GSF).

## Minor comments
- In line 46, please provide either citation or your results for the second challenge.
- In line 98, assumption 2.2 is the Bradley-Terry model.  You should introduce it here instead of in line 242.
- In line 107, the definition of linear aggregation function is unclear.  Specifically, does the weight $w$ depend on $u$?  If $w$ is independent of $u$, linear aggregation functions do not contain egalitarian nor the GSF in equation (2).  If $w$ can be any function of $u$, any function of $u$ satisfies the definition.
- Equation (4) is not clear to me.  Instead of scalar $\kappa_i$, should the parameter of Dirichlet $n$-dimensional?
- in line 252, is $\hat{v} = \hat{A}\hat{v}$ a typo?

**Questions:**

Is SBO superior to using private votes alone?
To what extent does incorporating public votes enhance the learning process?

---

> ### Author Response · Authors · 2024-11-21
> **Responses to the review (1/2)**
>
> Thank you for the valuable feedback. The following are our responses and **the changes are highlighted in magenta** in the updated manuscript (pdf).
>
> # Confusion on our contribution
> **Short answer.** We are not attempting to enhance the regret bound by using private votes. Instead, our goal is to achieve a regret bound similar to the best case (private-votes-only) by primarily using noisy public votes while requiring fewer private votes. Estimating the consensus from cheap public votes is far more cost-effective than relying solely on expensive private votes. As demonstrated in Theorem 3.9, the regret bound is comparable, but the sample complexity of private votes is sublinear (better than linear, i.e., private-vote-only). This represents our primary algorithmic contribution, demonstrating improved sample complexity and regret across four real-world and six synthetic tasks. Another contribution is the impossibility theorem of groupthink-proof aggregation, which we proved and is not part of existing results. We unpack each part in the following.
>
> ## Confusion on impossibility theory
> Your quote (lines 47-48) refers to the well-known Arrow's impossibility theorem, which demonstrates that no aggregation function can simultaneously satisfy all four mild rational axioms outlined by Arrow. This result is independent of the corruption of utility by social influence. In fact, we have formalised the discrepancy challenge you mentioned in Theorem 3.3 already. We mentioned Arrow's theorem to highlight that any consensus-building problem cannot be solved merely by designing a new aggregation function. The standard approach in the literature to circumvent such impossibility is to relax one or more of the rationality axioms.
> As such, in our work, we adopt the assumption of a facilitator or social planner (Assumption 2.1), who determines the aggregation function. Selecting a certain aggregation function inherently entails compromising on some of the axioms, but the social planner can decide which aspects to relax. The social planner assumption is widely accepted in welfare economics (Jehle & Reny, Advanced Microeconomic Theory, 2011). However, the task of selecting an appropriate aggregation function is non-trivial; therefore, we introduced the GSF (Proposition 2.4), which simplifies this challenging decision process to a single parameter, $\rho$, representing an additional contribution of our work.
> Our new impossibility theorem of groupthink-proofness (Theorem 3.3) highlights why direct access to private votes is necessary; without it, truthful consensus-building under a given aggregation function (GSF) is impossible. This motivates our dual voting system.
>
> ## Confusion on Theorem 3.9 and Sample Complexity.
> Regarding sample complexity, Theorem 3.9 answers your question directly. The third row, "None," represents the "private votes alone" scenario (see Remark 3.10). The second column, $\lvert Q_t^u \rvert$, denotes the total number of private votes, quantifying the "sample complexity" (see definition in lines 209-211). For "None," the sample complexity is linear, $T$, meaning that private voting must continue indefinitely to converge, as we are using only private votes. By contrast, SBO has a sublinear bound, indicating that the total number of private votes asymptotically converges to a certain constant, allowing us to stop querying private votes at some point and estimate the global optimum based on noisy public votes alone. See Section 3.3, lines 293-294 "Line 7 outlines the stopping criterion, which halts the expensive private queries when the graph $A$ is accurately estimated." This is the benefit of the dual voting system: unlike "private votes alone," we can stop querying private votes, or at least $\lvert Q_t^u \rvert$ grows slower than linear $T$.
>
> Estimating consensus using noisy public votes naturally incurs a worse regret bound. Looking at the first column, $R_T$ represents cumulative regret (line 322). SBO has a sublinear regret bound with an additional term compared to the "None" case, a necessary trade-off for the sublinear sample complexity $\lvert Q_t^u \rvert$. See Section 3.4, lines 351-353, "under typical conditions ($T \leq 1000$, $d \geq 2$, RBF kernel), $q$ dependent term is always smaller than non-dependent ones in RT". Thus, in practice, the bound does not differ significantly from the "None" regret bound.
> These theoretical results are corroborated by our experiments (Fig.3,4), where SBO's regret converges slightly better than the oracle, which has only the second term. This confirms that the additional term for SBO does not have a substantial impact. Moreover, the sample complexity $\lvert Q_t^u \rvert$ shows that SBO requires the fewest private queries among the baselines, while the single-agent baseline (cyan) shows the linear bound $T$.
> In conclusion, SBO achieves better sample complexity of expensive private votes with minimal compromise in regret, making it better than "private vote alone".

---

> > ### Comment · Reviewer_x1x4 · 2024-11-22
> >
> > Thank you for clarifying Theorem 3.9.   I indeed misunderstood this.  However, I still found the comparison between private vote only and mixed vote (SBO) unclear.  The sample complexity is $T$ and $T^q$ respectively, but regret is $\sqrt{T}$ and $T^{1-q/4}$ where $q\in [0,1]$ even with $q$ very close to $1$.  It seems SBO does not dominate the private vote only setting in both sample complexity and regret when $T$ is large.
> >
> > This may be related to the notion of regret itself.  I also wonder why the author chose regret as the objective instead of a more standard metric, such as estimation error.
> >
> > (Remark 3.1 seems to have typo: larger $q$?)

---

> ### Author Response · Authors · 2024-11-21
> **Responses to the review (2/2)**
>
> # Clarification on dictatorship.
> We thank the reviewer for pointing out our typo in the definition, the quantifiers should have been swapped. However, note that our original proof was based on the correct definition of dictatorship. Furthermore, after taking Reveiwer DBM8's comment into consideration, we have improved the proof of the impossibility result by relaxing the need for the dictatorship requirement, please check the new Definition 3.2 and Theorem 3.3.
>
> # Clarification on Readability.
> Our contribution bridges the fields of welfare economics and Bayesian optimization, where readers may have expertise in one but not necessarily both. We acknowledge that our interdisciplinary topic is inherently complex, and the challenges in comprehension extend beyond mere readability. Within the 10-page limit, we aimed to introduce the novel concepts as explicitly as possible, structuring them as assumptions and supplementing them with a 50-page appendix. The appendix includes hyperlinks to relevant topics as the table of contents and a summarized notation table for ease of reference.
>
> The parts of the paper you deemed irrelevant are, in fact, central to motivating our problem setup. Arrow’s impossibility theorem underpins the need for the facilitator assumption (Assumption 2.1), and the Generalized Social Function (GSF, Proposition 2.4) addresses how the facilitator can intuitively control the relaxation of rationality axioms through a single parameter, $\rho$. Furthermore, the GSF is directly incorporated into our objective function in Eq. (1), making it an essential part of our framework (see line 118 where this is mentioned).
>
> Despite these challenges, we have further improved the readability of the manuscript by incorporating the reviewer’s feedback. Thank you for your detailed comments on unclear sections and your more direct suggestions as minor points. We have updated the manuscript to enhance clarity on these aspects with magenta-coloured texts and believe that the readability and previously unclear contributions have been clarified now.

---

> ### Author Response · Authors · 2024-11-22
> **Response to the concern (1/2)**
>
> Thank you for your timely response and for investing your time in discussing our paper. Below is our detailed response.
>
> # Confusion Regarding Theorem 3.9 and Table 1
>
> First, Theorem 3.9 provides a general bound for an arbitrary kernel. All the kernel-dependent parameters—maximum information gain $\gamma_T$, confidence interval parameter $\beta_T$, and the covering number $\mathcal{N}(\mathcal{B}^n, 1/T, ||\cdot||_\infty)$—depend on $T$. As such, the second term is not as simple as $\sqrt{T}$. To address this, we prepared Table 1 to provide kernel-specific bounds for cumulative regret $R_T$ and sample complexity $|Q_t^u|$ for commonly used kernels: linear, RBF, and Matérn.
>
> In the Bayesian optimization (BO) community, the RBF kernel is typically used. Therefore, we will use the bound for the RBF kernel as an example for further interpretation.
>
> For regret $R_T$, there are two terms: $T^{1-q/4}$ and $T^{3/4} (\log T)^{3/4 (d+1)}$. The latter term is shared with "oracle (public only)" and "None (private-vote-only)." Hence, the first term, $T^{1-q/4}$, is the only additional term compared to the private-only case (see Appendix F.2.3 for details). In other words, if $T^{1-q/4} \ll T^{3/4} (\log T)^{3/4 (d+1)}$, the regret bound for SBO is comparable to the private-only case because convergence is primarily dictated by the dominant term.
>
> Setting $q = 1/2$ (see Line 370), the $T^{1-q/4}$ term becomes $T^{7/8}$, which seems worse than $T^{3/4}$. However, the $(\log T)^{3/4 (d+1)}$ term is substantial for typical values of $T < 1000$ and $d > 2$, making $T^{3/4} (\log T)^{3/4 (d+1)}$ the dominant term. For example, let $T = 50$ and $d = 3$, a typical BO setting also used in our experiments. Here, $T^{7/8} \approx 30$, while $T^{3/4} (\log T)^{3/4 (d+1)} \approx 281$. This demonstrates the dominance of the second term. The disparity grows as $T$ and $d$ increase, which we confirmed experimentally.
>
> In Figures 3–4, SBO (yellow) shows smaller regret than the oracle, which only has the second term. The oracle includes an additional constant $n$ in the second term, $T^{3/4} (\log T)^{3/4 (d+1)}$, meaning SBO outperforms it. This further supports that the enlarged second term $T^{3/4} (\log T)^{3/4 (d+1)}$ has a greater impact on convergence than the additive term $T^{7/8}$, confirming its triviality in convergence.
>
> For sample complexity $|Q_t^u|$, while the private-vote-only approach has linear complexity $T$, SBO achieves sublinear complexity $T^q (\log{T})^{3(d+1)}$. Comparing sublinear and linear complexities, the latter is the worst-case scenario for convergence. To interpret, we consider the sample complexity per iteration, $|Q_t^u| / t$, rather than the cumulative number of private votes, $|Q_t^u|$.
>
> For private-vote-only, $\lim_{t \rightarrow \infty} |Q_t^u| / t = t / t = 1$, which is expected as private votes are required for every iteration, even as $t \to \infty$. In contrast, SBO’s sublinear convergence implies $\lim_{t \rightarrow \infty} |Q_t^u| / t = 0$, meaning private vote queries can effectively stop after some point. Thus, sublinear convergence is always better than linear in terms of convergence.
>
> You might question whether $T^q (\log{T})^{3(d+1)}$ could exceed $T$ in certain scenarios, making it worse in the short-term. However, our theoretical bound represents a worst-case scenario. In practice, Line 7 in Algorithm 1 defines the stopping criterion for querying private votes. This ensures that the actual number of private votes per iteration averages between 0 and 1. Thus, even if $T^q (\log{T})^{3(d+1)} / T > 1$, it is effectively capped at 1 in reality. As a result, our sample complexity is always better than linear in expectation for any iteration $t$.
>
> Intuitively, probabilistically selecting between 0 and 1 queries is inherently smaller than deterministically selecting 1 query. This is confirmed experimentally in Figures 3–4, where SBO (yellow) has a smaller sample complexity $|Q_t^u|$ than the linear case (single agent, light blue).
>
> ## Possible Confusion Regarding the $q$ Setting
>
> Another potential source of confusion is the interpretation of $q$ as an optimization hyperparameter. Users have the flexibility to freely select its value within the range $0 < q < 1$. This means users can choose $q \rightarrow 1$, but they are not required to do so. As stated on Line 370, our recommendation is $q = 1/2$ because it aligns the convergence rate with the worst-case estimation error of graph identification (Theorem 3.12).
>
> CONTINUE TO THE NEXT RESPONSE

---

> > ### Author Response · Authors · 2024-11-22
> > **Response to the concern (2/2)**
> >
> > ## Clarification on Regret
> >
> > Regret is a standard metric in all Bayesian optimization and bandit-related papers. It represents the "estimation error of the global maximum," making it a natural choice for evaluating global maximization algorithms.
> >
> > Our goal is not to make the model as accurate as possible (i.e., minimizing the estimation error of model predictions). Instead, our objective is to find the global maximum in as few iterations as possible (i.e., minimizing the estimation error of the global maximum). The estimation error of the global maximum is expressed as $S(x^\star) - S(x_t)$, referred to as instantaneous regret. Its cumulative sum over $T$ iterations is called cumulative regret, defined as $R_T := \sum_{t=1}^T (S(x^\star) - S(x_t))$ (see Line 323).
> >
> > In the Bayesian optimization and bandit communities, cumulative regret is typically preferred because it directly relates to the convergence rate of the global optimizer, $R_T / T$. As shown in Theorem 3.9 and Table 1, our model achieves a sublinear convergence rate in the worst-case scenario, meaning $\lim_{t \rightarrow \infty} R_T / T = 0$. This guarantees that our algorithm asymptotically converges to the global maximum.
> >
> > Thus, regret aligns well with our objective—global optimization—and serves as a standard evaluation metric in this context.
> >
> > Thank you for pointing out the typo. You are correct; this should indeed be larger. We have modified the part. Thanks! We hope this clarifies our contributions.

---

> ### Author Response · Authors · 2024-11-27
> **Any further questions or suggestions?**
>
> Thank you for taking the time to review and discuss our paper. We hope our responses have adequately addressed your concerns regarding the soundness of our theoretical results. Should you have any further questions or require additional clarification, we would be happy to assist.
>
> To briefly recap our previous response: Theorem 3.9 applies to a general kernel, which is why the rate is not as simple as $\sqrt{T}$. Table 1, on the other hand, focuses on specific popular kernels, and we use the RBF kernel in our experiments, as is standard practice in the Bayesian optimization community. The term $T^{1-q/4}$ is smaller than $T^{3/4} (\log T)^{3/4(d+1)}$, making the additional regret term negligible.
>
> Our sample complexity for private votes, denoted $|Q^u_t|$, is sublinear, which is an improvement over the linear complexity in the private-votes-only case. This is because our algorithm selects either 0 or 1 private vote per iteration, whereas the private-votes-only approach always selects 1 private vote per iteration. As a result, $|Q^u_t|$ is sublinear and converges to zero private votes asymptotically. Consequently, Theorem 3.9 and Table 1 together clearly demonstrate the advantages of our algorithm over the private-votes-only approach. These benefits are also examined in the experiments shown in Figures 3 and 4.
>
> Finally, we use regret as our primary evaluation metric, as it is the standard metric in all Bayesian optimization and bandit papers. Regret measures the "estimation error of the global maximum," which aligns with our goal of global optimization, rather than the "estimation error of model prediction." Prediction itself is not our end goal; instead, the predictive model is a tool to enhance the global optimization process.

---

> > ### Comment · Reviewer_x1x4 · 2024-11-27
> >
> > Thank you for the summary.  Most of my concerns about the theoretical part are addressed.  I will adjust my score to reflect this.

---

> > > ### Author Response · Authors · 2024-11-28
> > >
> > > Thank you for considering our responses. We are glad to hear that our replies addressed most of your concerns. We would greatly appreciate it if you could reflect this in an adjusted score.

---

> > > > ### Author Response · Authors · 2024-12-02
> > > > **Any further questions or suggestions?**
> > > >
> > > > As the discussion period approaches an end, we kindly ask if you could update your score if our discussion has sufficiently addressed any initial concerns or confusion.
> > > > Thank you once again for taking the time to discuss our work and for your valuable suggestions, particularly on improving the clarity of Theorem 3.9. We have incorporated your feedback into the updated manuscript. Your final contribution in reflecting this progress through your score is crucial to us and to the paper we have collaboratively refined together.

---

### Author Response · Authors · 2024-12-02
**Summary of our discussion**

We sincerely thank the reviewers for their time and insightful feedback, which have been invaluable in refining our work. As the discussion period approaches an end, we summarize our contributions and address the key points raised during the review process below:

## Our Contribution
- **Novel Setting:**
 We introduce a new setting: preference aggregation for heterogeneous agents under social influence. All reviewers agreed this is a novel and non-trivial contribution. While game-theoretic approaches face significant limitations—such as the lack of equilibrium guarantees in continuous domains and the non-submodularity of equilibria that hinder global optimization—the Generalized Gini social-evaluation welfare function addresses these issues while preserving fairness principles, including the Pigou-Dalton principle.
- **Formalization of Social Influence:**
 We define the social influence problem using a graph convolutional operator within a kernelized Bradley-Terry model. This formalism enables black-box optimization of unknown utility functions, akin to standard preferential Bayesian optimization. We prove that crafting an aggregation function alone cannot achieve a social-influence-free consensus. All reviewers agreed that our proof is both correct and significant.
- **Dual Voting System:**
 To circumvent the impossibility theorem, we propose a dual voting system comprising two types of votes: costly but social-influence-free private votes and noisy yet cheap public votes. Our novel algorithm, Social Bayesian Optimization (SBO), achieves sample-efficient consensus building by primarily relying on cheap public votes, occasionally querying private votes to estimate the underlying social influence graph. We provide proofs for sublinear cumulative regret bounds and sublinear sample complexity bounds for private votes. Reviewers unanimously acknowledged the correctness and significance of these theoretical results.
- **Real-world tasks:**
 We evaluated SBO across six synthetic and four real-world consensus-building tasks, consistently outperforming five baseline methods. While this contribution may have been underappreciated, it unequivocally demonstrates the practical efficacy of our algorithm.

## Summary of Discussion
- **Impossibility Theorem:**
 In the initial submission, we assumed non-dictatorship. Based on reviewer feedback, we relaxed this assumption to trivial social consensus. All reviewers agreed that the revised theorem is correct and significant.
- **Regret Bound and Sample Complexity:**
The initial submission relied heavily on mathematical notations and restricted Lemma F.3 to a positive case. We have since provided textual explanations alongside the mathematical proofs and revised Lemma F.3 using properties of the generalized Gini social-evaluation function. Reviewers confirmed the updated theorem’s correctness and importance.
- **Extendability of Results:**
We expanded the discussion on extending our results to other cases, such as alternate forms of social influence and non-linear graph structures. We believe this addresses the concerns raised by reviewer KuC6, although they has not yet responded.
- **Readability Concerns:**
Due to issues in the ICLR matching system, none of our assigned reviewers are experts in Bayesian optimization, which is our intended audience as indicated by the title and primary area selection. Readability is subjective and depends on familiarity with the topic. Elevating a non-expert to the forefront of theoretical research within the constraints of a 10-page paper is a significant challenge. Our work bridges welfare economics, serving as the motivation, with Bayesian optimization as the main contribution, resulting in the development of a novel algorithm. While preferential Bayesian optimization is a well-studied field, our work generalizes it to multi-agent scenarios under social influence, making it accessible to those familiar with Bayesian optimization. We primarily target welfare economists as users of our algorithm and consider our work successful if they can grasp its significance and applications. Moreover, the fact that all reviewers have clearly understood our theoretical results (e.g., regret and sample complexity) demonstrates the readability and accessibility of our paper for its intended audience. Therefore, the core issue lies with the reviewer-paper matching system, rather than the content of the paper itself.

We sincerely thank the reviewers for taking the time to read and discuss our paper. We kindly request that you reflect the points we adequately addressed during the discussion in your final scores. If further clarification or discussion is needed, please do not hesitate to let us know—we would be more than happy to address any additional questions or concerns.

---

> ### Comment · Reviewer_x1x4 · 2024-12-02
>
> I want to clarify that while I appreciate your effort to enhance the paper, due to time constraints, I have not reviewed, nor will I be able to review, your new impossibility results.

---

> ### Author Response · Authors · 2024-12-02
>
> Thank you for your response! While it is fair to say the result is neutral in terms of correctness for you, we would like to emphasize that your original concern has been addressed (see the copy below), as we no longer rely on the assumption of non-dictatorship. Additionally, we would like to clarify that our original definition of dictatorship aligned precisely with Arrow's original definition, though it contained some typographical errors, which Reviewer DBM8 rightly pointed out and also confirmed its correctness.
>
> > The impossibility result in Theorem 3.3 is strange, particularly in its nonstandard definition of dictatorship. Unlike traditional definitions, where one dictator determines the order of all items, Definition 3.2 allows for different dictators for each item pair. Under this definition, even majority voting or a simple average function might satisfy their dictatorship definition. This nonstandard definition undermines the impossibility result.

---

### Meta-Review · Area_Chair_WP2H · 2024-12-15

**Metareview:**

Reviewers like the problem and novel approach introduced in this paper, but also raised concerns on the significance of the result and the quality of presentation. Specifically, some reviewers think that the assumptions are too strong or not well-motivated. Reviewers acknowledged that the discussions were helpful, yet the overall sentiment remained non-positive. During the decision making process, we considered the reviews, the expertise of the reviewers, and other factors such as strength of preferences and their degree of participation in the discussions. While none of the reviewers have worked on exactly the same topic, they are experts of closely related topics and a successful NeurIPS paper is expected to make a positive impression on them. We hope the authors find the review helpful. Thanks for submitting to NeurIPS.

**Additional Comments On Reviewer Discussion:**

See above.

---

### Decision · Program_Chairs · 2025-01-22

Reject